# Logits are All We Need to Adapt Closed Models

**Gaurush Hiranandani** [1]  **Haolun Wu*** [2 3]  **Subhojyoti Mukherjee†** [4]  **Sanmi Koyejo** [2]

## Abstract

Many commercial Large Language Models (LLMs) are often closed-source, limiting developers to prompt tuning for aligning content generation with specific applications. While these models currently do not provide access to token logits, we argue that if such access were available, it would enable more powerful adaptation techniques beyond prompt engineering. In this paper, we propose a token-level probability reweighting framework that, given access to logits and a small amount of task-specific data, can effectively steer black-box LLMs toward application-specific content generation. Our approach views next-token prediction through the lens of supervised classification. We show that aligning black-box LLMs with task-specific data can be formulated as a label noise correction problem, leading to *Plugin* model – an autoregressive probability reweighting model that operates solely on logits. We provide theoretical justification for why reweighting logits alone is sufficient for task adaptation. Extensive experiments with multiple datasets, LLMs, and reweighting models demonstrate the effectiveness of our method, advocating for broader access to token logits in closed-source models. We provide our code at this https URL.

## 1. Introduction

The rise of Large Language Models (LLMs) has revolutionized generative Artificial Intelligence, yet the most capable models are often closed-source or black-box (Achiam et al., 2023; Bai et al., 2022a). These models generate text based on input prompts but keep their internal weights and training data undisclosed, limiting transparency and customization.

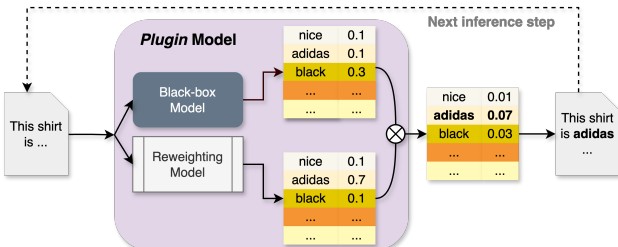

*Figure 1.* Inference phase of the *Plugin* model. The token probabilities are a product of the probabilities from the black-box model and a reweighting model that denotes label transitioning.

Despite these constraints, closed-source LLMs are widely adopted across applications ranging from travel itinerary generation to tax advice, with developers largely relying on prompt optimization to achieve domain-specific outputs.

However, this reliance on prompt engineering is insufficient for specialized tasks, e.g., those requiring brand-specific tone or style. Consider a content writer aiming to generate product descriptions that reflect a brand's unique identity. Black-box LLMs, trained on broad datasets, often fail to meet such nuanced requirements. With access limited to generated tokens, developers resort to zero-shot (Kojima et al., 2022) or few-shot (Song et al., 2023) prompting techniques. However, if model weights were accessible, advanced techniques like Parameter-Efficient Fine-Tuning (PEFT) using LoRA (Hu et al., 2021), QLoRA (Dettmers et al., 2024), prefix tuning (Li & Liang, 2021), or adapters (Hu et al., 2023a) could be employed for fine-tuning. Yet, due to intellectual property concerns and the high costs of development, most commercial LLMs remain closed-source, and even with API-based fine-tuning options, concerns over data privacy discourage developers from sharing proprietary data.

In this paper, we propose a middle ground between general-purpose LLM creators and developers seeking application-specific alignment. We argue that providing access to token logits, in addition to generated text, would enable more effective customization for downstream tasks. Viewing next-token prediction as a classification problem, we draw an analogy between LLMs and supervised classification models. Since decoder-only LLMs are trained to predict the next token given preceding tokens, aligning black-box LLMs to domain-specific data can be reframed as a label noise correction problem in supervised classification. In this analogy, the LLM's broad training data serves as proxy labels, while

---

*Work done while visiting Stanford University. †Work done while PhD Candidate at UW Madison. [1]Typeface [2]Stanford University [3]Mila - Quebec AI Institute [4]Adobe Research. Correspondence to: Gaurush Hiranandani <gaurush@typeface.ai>, Haolun Wu <haolunwu@stanford.edu>.

*Proceedings of the 42nd International Conference on Machine Learning*, Vancouver, Canada. PMLR 267, 2025. Copyright 2025 by the author(s).

application-specific data represents true labels. This can be interpreted as a distribution shift scenario. For example, in "label shift" (Lipton et al., 2018), certain tokens may appear more frequently in application-specific data than in the LLM's original corpus. In "class-dependent or independent label noise" (Patrini et al., 2017), synonymous expressions or stylistic variations in application data may diverge from those seen during model training.

Inspired by the label noise correction method of Patrini et al. (2017), which estimates a transition matrix to correct class-dependent noise, we adapt this idea to black-box LLM alignment. Unlike prior work that modifies the loss and retrains the model, we lack access to the LLM's training data and cannot retrain the model. Instead, we estimate an autoregressive transition matrix from application-specific data and use it to reweight token probabilities at inference.

This autoregressive extension is novel, as it accounts for dependencies on previously generated tokens when adjusting logits for the next token. By adapting label noise correction techniques to autoregressive language modeling, we present a practical method to align black-box LLMs using only logits—without requiring access to model weights or original training data.

Our contributions are summarized as follows:

1. We formulate the problem of adapting black-box LLMs for application-specific content generation as a loss correction approach, requiring only token logits at each generation step. This bridges label noise correction in supervised classification with autoregressive language modeling (Sections 2 and 3).

2. We propose an autoregressive probability reweighting framework, enabling token-level probability adjustment during inference. The resulting *Plugin* model dynamically reweights logits to align generation with task-specific data (Section 4).

3. We provide theoretical guarantees, showing that under mild assumptions, the *Plugin* model consistently aligns probability estimates with the target distribution given sufficient application-specific samples. To our knowledge, this is the first work to establish such consistency in an autoregressive label noise setting (Section 5).

4. We conduct extensive experiments across four language generation datasets and three black-box LLMs. Our results, supported by multiple ablations, demonstrate that the *Plugin* model outperforms baselines in adapting black-box LLMs for domain-specific content generation (Section 7). Based on our results, we advocate for publishing token logits alongside outputs in closed-source LLMs.

## 2. Preliminaries

We begin by establishing the notation. The index set is denoted as $[c] = \{1, \ldots, c\}$ for any positive integer $c$. Vectors are represented in boldface, for example, $\boldsymbol{v}$, while matrices are denoted using uppercase letters, such as $V$. The coordinates of a vector are indicated with subscripts, for instance, $v_j$. The all-ones vector is denoted by $\mathbf{1}$, with its size being clear from the context. The $c$-dimensional simplex is represented as $\Delta^{c-1} \subset [0, 1]^c$. Finally, a sequence $(x_t, x_{t-1}, \ldots, x_1)$ of size $t$ is denoted by $x_{t:1}$.

We assume access to language data for the target task, while the black-box LLM, trained on broad world knowledge, is treated as having learned from a noisy version of this data. We seek to adapt the black-box model to align with the task-specific distribution. To formalize this, we extend the label-noise framework from supervised classification (Patrini et al., 2017) to the decoder-only language modeling.

Decoder-only models are trained using a next-token prediction objective. At each step, this setup resembles a supervised classification problem with $|V|$ classes, where $V$ is the vocabulary of tokens. Formally, the label space at step $t$ is $\mathcal{X}_t = \{\boldsymbol{e}^i : i \in [|V|]\}$, where $\boldsymbol{e}^i$ denotes the $i$-th standard canonical vector in $\mathbb{R}^{|V|}$, i.e., $\boldsymbol{e}^i \in \{0, 1\}^{|V|}, \mathbf{1}^T \boldsymbol{e}^i = 1$. The task at each step $t$ is to predict the next token $\boldsymbol{x}_t$ (denoted as one-hot vector) given a sequence of tokens $\boldsymbol{x}_{t-1:1}$.

One observes examples $(\boldsymbol{x}_t, \boldsymbol{x}_{t-1:1})$ drawn from an unknown distribution $p^*(\boldsymbol{x}_t, \boldsymbol{x}_{t-1:1}) = p^*(\boldsymbol{x}_t|\boldsymbol{x}_{t-1:1})p^*(\boldsymbol{x}_{t-1:1})$ over $V \times V^{[t-1]}$, with expectations denoted by $E^*_{\boldsymbol{x}_t, \boldsymbol{x}_{t-1:1}}$. Cross-entropy loss is typically used for training over the vocabulary tokens. Assuming access to token logits, and thus the softmax outputs, from the black-box LLM, we interpret the softmax output as a vector approximating the class-conditional probabilities $p^*(\boldsymbol{x}_t|\boldsymbol{x}_{t-1:1})$, denoted as $b(\boldsymbol{x}_t|\boldsymbol{x}_{t-1:1}) \in \Delta^{|V|-1}$.

To quantify the discrepancy between the target label $\boldsymbol{x}_t = \boldsymbol{e}^i$ at step $t$ and the model's predicted output, we define a loss function $\ell : |V| \times \Delta^{|V|-1} \to \mathbb{R}$. A common choice in next-token prediction tasks is the cross-entropy loss:

$$\ell(\boldsymbol{e}^i, b(\boldsymbol{x}_t|\boldsymbol{x}_{t-1:1})) = -(\boldsymbol{e}^i)^T \log b(\boldsymbol{x}_t|\boldsymbol{x}_{t-1:1})$$
$$= -\log b(\boldsymbol{x}_t = \boldsymbol{e}^i|\boldsymbol{x}_{t-1:1}). \quad (1)$$

With some abuse of notation, the loss in vector form $\boldsymbol{\ell} : \Delta^{|V|-1} \to \mathbb{R}^{|V|}$, computed on every possible label is $\boldsymbol{\ell}(b(\boldsymbol{x}_t|\boldsymbol{x}_{t-1:1})) = \left(\ell(\boldsymbol{e}^1, b(\boldsymbol{x}_t|\boldsymbol{x}_{t-1:1})), \ldots, \ell(\boldsymbol{e}^{|V|}, b(\boldsymbol{x}_t|\boldsymbol{x}_{t-1:1}))\right)^T$.

## 3. Loss Robustness

We extend label noise modeling to the autoregressive language setting, focusing on asymmetric or class-conditional

noise. At each step $t$, the label $\boldsymbol{x}_t$ in the black-box model's training data is flipped to $\tilde{\boldsymbol{x}}_t \in V$ with probability $p^*(\tilde{\boldsymbol{x}}_t|\boldsymbol{x}_t)$, while preceding tokens $(\boldsymbol{x}_{t-1:1})$ remain unchanged. As a result, the black-box model observes samples from a noisy distribution: $p^*(\tilde{\boldsymbol{x}}_t, \boldsymbol{x}_{t-1:1}) = \sum_{\boldsymbol{x}_t} p^*(\tilde{\boldsymbol{x}}_t|\boldsymbol{x}_t)p^*(\boldsymbol{x}_t|\boldsymbol{x}_{t-1:1})p^*(\boldsymbol{x}_{t-1:1})$.

We define the noise transition matrix $T_t \in [0,1]^{|V| \times |V|}$ at step $t$, where each entry $T_{t_{ij}} = p^*(\tilde{\boldsymbol{x}}_t = \boldsymbol{e}^j|\boldsymbol{x}_t = \boldsymbol{e}^i)$ represents the probability of label flipping. This matrix is row-stochastic but not necessarily symmetric.

To handle asymmetric label noise, we modify the loss $\ell$ for robustness. Initially, assuming a known $T_t$, we apply a loss correction inspired by (Patrini et al., 2017; Sukhbaatar et al., 2015). We then relax this assumption by estimating $T_t$ directly, forming the basis of our *Plugin* model approach.

We observe that a language model trained with no loss correction would result in a predictor for noisy labels $b(\tilde{\boldsymbol{x}}_t|\boldsymbol{x}_{t-1:1})$. We can make explicit the dependence on $T_t$. For example, with cross-entropy we have:

$$\ell(\boldsymbol{e}^i, b(\tilde{\boldsymbol{x}}_t|\boldsymbol{x}_{t-1:1})) = -\log b(\tilde{\boldsymbol{x}}_t = \boldsymbol{e}^i|\boldsymbol{x}_{t-1:1})$$
$$= -\log \sum_{j=1}^{|V|} p^*(\tilde{\boldsymbol{x}}_t = \boldsymbol{e}^i|\boldsymbol{x}_t = \boldsymbol{e}^j)b(\boldsymbol{x}_t = \boldsymbol{e}^j|\boldsymbol{x}_{t-1:1})$$
$$= -\log \sum_{j=1}^{|V|} T_{t_{ji}} b(\boldsymbol{x}_t = \boldsymbol{e}^j|\boldsymbol{x}_{t-1:1}), \qquad (2)$$

or in matrix form

$$\boldsymbol{\ell}(b(\tilde{\boldsymbol{x}}_t|\boldsymbol{x}_{t-1:1})) = -\log T_t^\top b(\boldsymbol{x}_t|\boldsymbol{x}_{t-1:1}). \qquad (3)$$

This loss compares the noisy label $\tilde{\boldsymbol{x}}_t$ to the noisy predictions averaged via the transition matrix $T_t$ at step $t$. Cross-entropy loss, commonly used for next-token prediction, is a *proper composite loss* with the softmax function as its *inverse link function* (Patrini et al., 2017). Consequently, from Theorem 2 of Patrini et al. (2017), the minimizer of the *forwardly-corrected* loss in Equation (3) on noisy data aligns with the minimizer of the true loss on clean data, i.e.,

$$\underset{w}{\arg\min}\, E^*_{\tilde{\boldsymbol{x}}_t, \boldsymbol{x}_{t-1:1}} \left[ \boldsymbol{\ell}(\boldsymbol{x}_t, T_t^\top b(\boldsymbol{x}_t|\boldsymbol{x}_{t-1:1})) \right]$$
$$= \underset{w}{\arg\min}\, E^*_{\boldsymbol{x}_t, \boldsymbol{x}_{t-1:1}} \left[ \boldsymbol{\ell}(\boldsymbol{x}_t, b(\boldsymbol{x}_t|\boldsymbol{x}_{t-1:1})) \right],$$

where $w$ are the language model's weights, implicitly embedded in the softmax output $b$ from the black-box model. This result suggests that if $T_t$ were known, we could transform the softmax output $b(\boldsymbol{x}_t \mid \boldsymbol{x}_{t-1:1})$ using $T_t^T$, use the transformed predictions as final outputs, and retrain the model accordingly. However, since $T_t$ is unknown and training data is inaccessible, estimating $T_t$ from clean data is essential to our approach.

### 3.1. Estimation of Transition Matrix

We assume access to a small amount of target data for the task. Given that the black-box model is expressive enough to approximate $p^*(\tilde{\boldsymbol{x}}_t \mid \boldsymbol{x}_{t-1:1})$ (Assumption (2) in Theorem 3 of Patrini et al. (2017)), the transition matrix $T_t$ can be estimated from this target data. Considering the supervised classification setting at step $t$, let $\mathcal{X}_t^i$ represent all target data samples where $\boldsymbol{x}_t = \boldsymbol{e}^i$ and the preceding tokens are $(\boldsymbol{x}_{t-1:1})$. A naive estimate of the transition matrix is: $\hat{T}_{t_{ij}} = b(\tilde{\boldsymbol{x}}_t = \boldsymbol{e}^j|\boldsymbol{x}_t = \boldsymbol{e}^i) = \frac{1}{|\mathcal{X}_t^i|} \sum_{x \in \mathcal{X}_t^i} b(\tilde{\boldsymbol{x}}_t = \boldsymbol{e}^j|\boldsymbol{x}_{t-1:1})$. While this setup works for a single step $t$, there are two key challenges in extending it across all steps in the token prediction task:

1. **Limited sample availability:** The number of samples where $\boldsymbol{x}_t = \boldsymbol{e}^i$ and the preceding tokens $(\boldsymbol{x}_{t-1}, \ldots, \boldsymbol{x}_1)$ match exactly is limited in the clean data, especially with large vocabulary sizes (e.g., $|V| = O(100K)$ for LLaMA (Dubey et al., 2024)). This necessitates modeling the transition matrix as a function of features derived from $\boldsymbol{x}_{t-1:1}$, akin to text-based autoregressive models.

2. **Large parameter space:** With a vocabulary size of $|V| = O(100K)$, the transition matrix $T_t$ has approximately 10 billion parameters. This scale may exceed the size of the closed-source LLM and cannot be effectively learned from limited target data. Thus, structural restrictions must be imposed on $T_t$ to reduce its complexity.

To address these challenges, we impose the restriction that the transition matrix $T_t$ is diagonal. While various constraints could be applied to simplify the problem, assuming $T_t$ is diagonal offers two key advantages. First, it allows the transition matrix—effectively a vector in this case—to be modeled using standard autoregressive language models, such as a *GPT-2 model with $k$ transformer blocks*, a *LLaMA model with $d$-dimensional embeddings*, or a fine-tuned *GPT-2-small* model. These architectures can be adjusted based on the size of the target data. Second, a diagonal transition matrix corresponds to a symmetric or class-independent label noise setup, where $\boldsymbol{x}_t = \boldsymbol{e}^i$ flips to any other class with equal probability in the training data. This assumption, while simplifying, remains realistic within the framework of label noise models.

Enforcing a diagonal structure ensures efficient estimation of the transition matrix while maintaining practical applicability within our framework. Next, we outline our approach for adapting closed-source language models to target data.

## 4. Proposed Method: The Plugin Approach

To estimate the autoregressive transition vector, we train an autoregressive language model on target data, which oper-

ates alongside the black-box model during inference. This model acts as an autoregressive reweighting mechanism, adjusting the token probabilities produced by the black-box model. The combined approach, integrating probabilities from the black-box and reweighting models, is referred to as the *Plugin* model. The term *Plugin* is inspired by classification literature, where plugin methods reweight probabilities to adapt to distribution shifts (Koyejo et al., 2014; Narasimhan et al., 2015; Hiranandani et al., 2021). We now detail the training and inference phases, summarized in Algorithm 1 (Appendix A) and illustrated in Figure 1.

### 4.1. Training the Plugin Model

During each training iteration, a sequence $s$ of $m$ tokens is passed through both the black-box model and the reweighting model to obtain token probabilities $\{\boldsymbol{b}_1, \boldsymbol{b}_2, \dots, \boldsymbol{b}_m\}$ and $\{\boldsymbol{r}_1, \boldsymbol{r}_2, \dots, \boldsymbol{r}_m\}$, respectively, where each $\boldsymbol{b}_i, \boldsymbol{r}_i \in \Delta^{|V|-1}$. The final token probability from the *Plugin* model is computed by normalizing the element-wise product of these probabilities:

$$\boldsymbol{p}_i = \frac{\boldsymbol{b}_i \odot \boldsymbol{r}_i}{\|\boldsymbol{b}_i \odot \boldsymbol{r}_i\|_1}. \tag{4}$$

The sequence-level cross-entropy loss is given by:

$$\ell_s = -\frac{1}{m} \sum_{i=1}^{m} \sum_{j=1}^{|V|} \log(\boldsymbol{p}_i) \odot \boldsymbol{e}_j, \tag{5}$$

where the $j$-th token appears at the $i$-th position in the sequence $s$. During backpropagation, only the reweighting model parameters are updated, while the black-box model remains frozen. This formulation extends naturally to batch training, refining $\boldsymbol{r}_i$ over iterations to approximate the transition vector governing label shifts in the target data.

### 4.2. Inference from the Plugin Model

Given a fully trained reweighting model and access to the black-box model, token generation proceeds autoregressively. At the first step, the black-box model produces token probabilities $\boldsymbol{b}_1$, while the reweighting model outputs $\boldsymbol{r}_1$. The *Plugin* model selects the first token as $\boldsymbol{x}_1 = \operatorname{argmax}_V(\boldsymbol{b}_1 \odot \boldsymbol{r}_1)$. For subsequent steps, given the previously generated tokens $\boldsymbol{x}_{t-1:1}$, we obtain probabilities $\boldsymbol{b}_t$ from the black-box model and $\boldsymbol{r}_t$ from the reweighting model. The *Plugin* model then predicts the next token as: $\boldsymbol{x}_t = \operatorname{argmax}_V(\boldsymbol{b}_t \odot \boldsymbol{r}_t)$.

The process continues until a stopping criterion is met. Note that, this manuscript focuses on greedy decoding for inference. Other decoding strategies, such as temperature scaling, top-$p$ sampling, or beam search, can be incorporated by normalizing the element-wise product of probabilities and using them as the final token distribution, as in Equation (4).

## 5. Theoretical Analysis

We establish the convergence properties of *Plugin*, showing that after $t$ tokens, it accurately estimates the autoregressive noise transition matrix. Modeling the matrix as a function of an unknown parameter $\boldsymbol{\theta}_*$, we prove that optimizing the autoregressive loss over token sequences enables consistent estimation of $\boldsymbol{\theta}_*$ with high probability. To our knowledge, this is the first finite-time convergence analysis for transition matrix estimation under autoregressive noisy loss.

Let $\mathcal{F}^{t-1}$ denote the history of selected tokens up to time $t-1$. Let an unknown parameter $\boldsymbol{\theta}_* \in \boldsymbol{\Theta} \subseteq \mathbb{R}^d$ governs the transition dynamics of label flipping between token pairs. The transition matrix at time $t$, denoted as $T_t(\boldsymbol{\theta}_* | \mathcal{F}^{t-1})$, depends on $\boldsymbol{\theta}_*$ and all previously observed tokens. Before proving our main result, we first make a few assumptions.

**Assumption 5.1.** Let $T_t(\boldsymbol{\theta}_*; x_i, x_j, \mathcal{F}^{t-1})$ denote the $(i, j)$-th component of the transition matrix, and let $f_{I_t}(\boldsymbol{\theta}_*; x_i, x_j, \mathcal{F}^{t-1})$ be the transition function that determines the transition from $x_i$ to $x_j$, where $I_t$ is the $x_i$ token selected at time $t$. Let $x_i, x_j \in \mathbb{R}^d$. We assume that $\nabla f_{I_t}(\boldsymbol{\theta}_*; x_i, x_j, \mathcal{F}^{t-1}) < \lambda_0$ and $\nabla^2 f_{I_t}(\boldsymbol{\theta}_*; x_i, x_j, \mathcal{F}^{t-1}) < \lambda_1$ for some constant $\lambda_0 > 0$, $\lambda_1 > 0$ and for all steps $t$.

Assumption 5.1 assumes the transition matrix depends on the history-dependent function $f_{I_t}(\cdot)$ with bounded gradient and Hessian, similar to assumptions in (Singh et al., 2023; Zhang et al., 2024) for other deep models.

**Assumption 5.2.** We assume the cross-entropy loss (5) is clipped by $\epsilon > 0$ and upper bounded as $\ell_t^{clipped} \leq C|V|^2(Y_t - f_{I_t}(\boldsymbol{\theta}_*; x_i, x_j, \mathcal{F}^{t-1}))^2$ for any time $t$, where $Y_t$ is the predicted token class, $f_{I_t}$ determines the true class and satisfies Assumption 5.1, and $C > 0$ is a constant.

Assumption 5.2 ensures that the clipped log loss is upper bounded by a smoother squared loss. For the remaining of this section we refer to this squared loss at time $t$ as $\ell_t(\boldsymbol{\theta})$. Let the *Plugin* model minimize the loss $\ell_1(\boldsymbol{\theta}), \ell_2(\boldsymbol{\theta}), \cdots, \ell_t(\boldsymbol{\theta})$ over $t$ iterations. Let $\widehat{\boldsymbol{\theta}}_t = \operatorname{argmin}_{\boldsymbol{\theta} \in \boldsymbol{\Theta}} \sum_{s=1}^{t} \ell_s(\boldsymbol{\theta})$. At every iteration $t$, the *Plugin* algorithm looks into the history $\mathcal{F}^{t-1}$ and samples a token $\boldsymbol{x}_t \sim \boldsymbol{p}_{\hat{\theta}_t} = \boldsymbol{b}_t \odot \boldsymbol{r}_{\hat{\theta}_t}$.

Let $\widehat{\mathcal{L}}_t(\boldsymbol{\theta}) = \frac{1}{t} \sum_{s=1}^{t} \ell_s(\boldsymbol{\theta})$ and its expectation $\mathcal{L}_t(\boldsymbol{\theta}) = \frac{1}{t} \sum_{s=1}^{t} \mathbb{E}_{x_s \sim \mathbf{P}_{\hat{\theta}_{s-1}}}[\ell_s(\boldsymbol{\theta}) | \mathcal{F}^{s-1}]$. We impose regularity and smoothness assumptions on the loss function $\ell_t(\boldsymbol{\theta})$ as stated in Assumption B.1 (Appendix B). We are now ready to prove the main theoretical result of the paper.

**Theorem 1.** *Suppose $\ell_1(\boldsymbol{\theta}), \cdots, \ell_t(\boldsymbol{\theta}) : \mathbb{R}^{|V|} \to \mathbb{R}$ are loss functions from a distribution that satisfies Assumptions 5.1, 5.2, and B.1. Define $\mathcal{L}_t(\boldsymbol{\theta}) = \frac{1}{t} \sum_{s=1}^{t} \mathbb{E}_{x_s \sim \mathbf{P}_{\hat{\theta}_{s-1}}}[\ell_s(\boldsymbol{\theta}) | \mathcal{F}^{s-1}]$ where,*

$\widehat{\boldsymbol{\theta}}_t = \operatorname{argmin}_{\boldsymbol{\theta} \in \boldsymbol{\Theta}} \sum_{s=1}^{t} \ell_s(\boldsymbol{\theta})$. *If $t$ is large enough such that*

$$\frac{\gamma \log(dt)}{t} \leq c' \min \left\{ \frac{1}{C_1 C_2 |V|^4}, \frac{\max_{\boldsymbol{\theta} \in \boldsymbol{\Theta}} (\mathcal{L}_t(\boldsymbol{\theta}) - \mathcal{L}_t(\boldsymbol{\theta}_*))}{C_2} \right\}$$

*then for a constant $\gamma \geq 2$, universal constants $C_1, C_2, c'$, we have that*

$$(1 - \rho_t) \frac{\sigma_t^2}{t} - \frac{C_1^2}{t^{\gamma/2}} \leq \mathbb{E} \left[ \mathcal{L}_t(\widehat{\boldsymbol{\theta}}_t) - \mathcal{L}_t(\boldsymbol{\theta}_*) \right]$$

$$\leq (1 + \rho_t) \frac{\sigma_t^2}{t} + \frac{\max_{\boldsymbol{\theta} \in \boldsymbol{\Theta}} (\mathcal{L}_t(\boldsymbol{\theta}) - \mathcal{L}_t(\boldsymbol{\theta}_*))}{t^{\gamma}},$$

*where* $\sigma_t^2 := \mathbb{E} \left[ \frac{1}{2} \left\| \nabla \widehat{\mathcal{L}}_t(\boldsymbol{\theta}_*) \right\|_{(\nabla^2 \mathcal{L}_t(\boldsymbol{\theta}_*))^{-1}}^2 \right]$, *and* $\rho_t :=$ $\left( C_1 C_2 + 2\eta^2 \lambda_1^2 \right) \sqrt{\frac{\gamma \log(dt)}{t}}$.

Theorem 1 bounds the difference between the estimated and true average loss functions, showing that this gap diminishes as the number of training tokens increases. Since $\widehat{\boldsymbol{\theta}}_t = \operatorname{argmin}_{\boldsymbol{\theta} \in \boldsymbol{\Theta}} \sum_{s=1}^{t} \ell_s(\boldsymbol{\theta})$, the *Plugin* model progressively refines its estimation of the unknown parameter $\boldsymbol{\theta}_*$. As the transition matrix $T_t(\boldsymbol{\theta}_*; x_i, x_j, \mathcal{F}^{t-1})$ is derived from $f_{I_t}(\boldsymbol{\theta}_*; x_i, x_j, \mathcal{F}^{t-1})$, which depends on $\boldsymbol{\theta}_*$, training on sufficiently many tokens ensures an accurate estimation of each component of $T_t(\boldsymbol{\theta}_* | \mathcal{F}^{t-1})$.

Our proof reformulates the problem as a sequential hypothesis testing setting to estimate the average loss function $\mathcal{L}_t(\widehat{\boldsymbol{\theta}}_t)$ using the sequence of losses $\ell_1(\boldsymbol{\theta}), \dots, \ell_t(\boldsymbol{\theta})$ (Naghshvar & Javidi, 2013; Lattimore & Szepesvári, 2020). Unlike prior work (Frostig et al., 2015; Chaudhuri et al., 2015), which assumes i.i.d. losses, the loss at time $t$ in our setting depends on all previous losses. Additionally, Mukherjee et al. (2022) study a different active regression setting without considering cross-entropy loss or transition noise matrices as in Patrini et al. (2017). We provide a brief overview of the proof technique in Remark B.9 (Appendix B), highlighting key novelties.

## 6. Related Work

**Parameter-Efficient Fine-Tuning (PEFT).** PEFT methods adapt LLMs to downstream tasks while minimizing computational overhead. LoRA (Hu et al., 2021) and QLoRA (Dettmers et al., 2024) introduce low-rank updates and quantization for efficient fine-tuning, while prefix tuning (Li & Liang, 2021), adapters (Hu et al., 2023b), and soft prompting (Lester et al., 2021) modify task-specific representations through trainable layers or embeddings. Torroba-Hennigen et al. (2025) further explore the equivalence between gradient-based transformations and adapter-based tuning. However, these methods require access to model weights, gradients, or architecture details, making them unsuitable for closed-source LLMs and inapplicable as baselines in our setup. In contrast, our approach operates solely

on token logits, enabling adaptation without modifying the underlying model. Thus, we emphasize that the *Plugin* model is not an alternative to fine-tuning, but rather an approach that uniquely stands for adapting black-box LLMs which only provide logit access.

**Steering and Aligning LLMs.** LLM alignment methods primarily use reinforcement learning or instruction tuning. RLHF and DPO (Christiano et al., 2017; Ouyang et al., 2022; Rafailov et al., 2024) optimize model behavior via human preferences, with DPO eliminating reward modeling. Constitutional AI (Bai et al., 2022b) aligns models using self-generated principles, while instruction tuning (Wei et al., 2021; Sanh et al., 2022) adapts them via task-specific demonstrations. Unlike our approach, these methods require model weights and training data, limiting their applicability as baselines in our setup.

**Calibration of LLMs.** LLM calibration methods aim to align model confidence with predictive accuracy and adjust confidence scores but do not alter token predictions (Ulmer et al., 2024; Shen et al., 2024; Huang et al., 2024; Kapoor et al., 2024; Zhu et al., 2023; Zhang et al., 2023). In contrast, our method reweights token probabilities at inference, enabling adaptation of black-box LLMs without modifying the model or requiring fine-tuning.

**Black-box LLMs.** Prior work explores various approaches for adapting black-box LLMs without fine-tuning, though they differ fundamentally from our method. (Gao et al., 2024) infer user preferences through interactive edits but do not adapt models based on past language data. Diffusion-LM (Li et al., 2022) formulates text generation as a non-autoregressive denoising process, whereas our approach reweights token probabilities autoregressively without requiring black-box model weights. Discriminator-based methods (Dathathri et al., 2020; Mireshghallah et al., 2022; Yang & Klein, 2021; Krause et al., 2021) control generation based on predefined attributes, contrasting with our method, which enables free-form text adaptation. DExperts (Liu et al., 2021; 2024) combines expert and anti-expert probabilities; we incorporate a similar probability combining strategy in a modified baseline without a de-expert component. In-context learning (Long et al., 2023; Dong et al., 2024) offers a common adaptation technique for black-box models and serves as a baseline in our setup.

## 7. Experiments

We divide this section into four parts. Section 7.1 evaluates *Plugin* on four text generation datasets across three black-box language models. Since the *Plugin* model is trained on top of black-box models, we refer to black-box models interchangeably as *base models*. Section 7.2 discusses how *Plugin* can be applied on top of any prompt-tuning method as a wrapper, when logits are accessible. Section 7.3 presents ablation studies analyzing the impact of black-box

*Table 1.* Performance comparison on E2E NLG dataset. We show mean and standard deviation of the metrics over five seeds.

| Model | Method | BLEU | Rouge-1 | Rouge-2 | Rouge-L | METEOR | CIDEr | NIST |
|---|---|---|---|---|---|---|---|---|
| GPT2-M | Zeroshot | 0.0247 | 0.3539 | 0.1003 | 0.2250 | 0.3015 | 0.0156 | 0.6133 |
| GPT2-M | ICL-1 | $0.0543_{\pm0.026}$ | $0.3431_{\pm0.048}$ | $0.1299_{\pm0.033}$ | $0.2280_{\pm0.047}$ | $0.3434_{\pm0.051}$ | $0.0260_{\pm0.042}$ | $0.7767_{\pm0.060}$ |
| GPT2-M | ICL-3 | $0.0750_{\pm0.035}$ | $0.3955_{\pm0.028}$ | $0.1676_{\pm0.020}$ | $0.2649_{\pm0.052}$ | $0.3977_{\pm0.063}$ | $0.0252_{\pm0.049}$ | $0.8993_{\pm0.076}$ |
| GPT2-M | NewModel | $\mathbf{0.2377}_{\pm0.011}$ | $\underline{0.5049}_{\pm0.014}$ | $\mathbf{0.2742}_{\pm0.013}$ | $\mathbf{0.3902}_{\pm0.006}$ | $\underline{0.4521}_{\pm0.016}$ | $\mathbf{0.3938}_{\pm0.019}$ | $\mathbf{1.1927}_{\pm0.069}$ |
| GPT2-M | WeightedComb | $0.1709_{\pm0.008}$ | $0.4817_{\pm0.020}$ | $0.2447_{\pm0.011}$ | $0.3720_{\pm0.014}$ | $0.4071_{\pm0.025}$ | $0.3329_{\pm0.027}$ | $1.0864_{\pm0.002}$ |
| GPT2-M | TempNet | $0.1036_{\pm0.014}$ | $0.3425_{\pm0.016}$ | $0.1526_{\pm0.012}$ | $0.2735_{\pm0.016}$ | $0.2615_{\pm0.016}$ | $0.4116_{\pm0.016}$ | $0.2826_{\pm0.057}$ |
| GPT2-M | **Plugin (Ours)** | $\underline{0.1863}_{\pm0.010}$ | $\mathbf{0.5227}_{\pm0.011}$ | $\underline{0.2612}_{\pm0.013}$ | $\underline{0.3728}_{\pm0.003}$ | $\mathbf{0.4857}_{\pm0.012}$ | $\underline{0.3544}_{\pm0.013}$ | $\underline{1.1241}_{\pm0.009}$ |
| GPT2-XL | Zeroshot | 0.0562 | 0.4013 | 0.1636 | 0.2862 | 0.3697 | 0.0187 | 0.5338 |
| GPT2-XL | ICL-1 | $0.0686_{\pm0.032}$ | $0.4016_{\pm0.042}$ | $0.1404_{\pm0.052}$ | $0.2745_{\pm0.025}$ | $0.3503_{\pm0.019}$ | $0.0353_{\pm0.015}$ | $0.7944_{\pm0.067}$ |
| GPT2-XL | ICL-3 | $0.0980_{\pm0.035}$ | $0.4188_{\pm0.040}$ | $0.1923_{\pm0.046}$ | $0.2912_{\pm0.031}$ | $0.3925_{\pm0.027}$ | $0.0250_{\pm0.017}$ | $0.9390_{\pm0.054}$ |
| GPT2-XL | NewModel | $\underline{0.2377}_{\pm0.011}$ | $\underline{0.5049}_{\pm0.014}$ | $\underline{0.2742}_{\pm0.013}$ | $\underline{0.3902}_{\pm0.006}$ | $\underline{0.4521}_{\pm0.016}$ | $\underline{0.3938}_{\pm0.019}$ | $\underline{1.1927}_{\pm0.069}$ |
| GPT2-XL | WeightedComb | $0.1184_{\pm0.010}$ | $0.4237_{\pm0.016}$ | $0.1858_{\pm0.012}$ | $0.3004_{\pm0.010}$ | $0.3776_{\pm0.016}$ | $0.1818_{\pm0.023}$ | $1.0261_{\pm0.057}$ |
| GPT2-XL | TempNet | $0.1325_{\pm0.013}$ | $0.4642_{\pm0.017}$ | $0.2516_{\pm0.016}$ | $0.3021_{\pm0.022}$ | $0.4126_{\pm0.025}$ | $0.3627_{\pm0.033}$ | $0.8027_{\pm0.047}$ |
| GPT2-XL | **Plugin (Ours)** | $\mathbf{0.2470}_{\pm0.009}$ | $\mathbf{0.5536}_{\pm0.007}$ | $\mathbf{0.3084}_{\pm0.007}$ | $\mathbf{0.4213}_{\pm0.008}$ | $\mathbf{0.5057}_{\pm0.009}$ | $\mathbf{0.5455}_{\pm0.013}$ | $\mathbf{1.2736}_{\pm0.051}$ |
| LLaMA-3.1-8B | Zeroshot | 0.3226 | 0.6917 | 0.4050 | 0.5004 | 0.6041 | 0.9764 | 1.1310 |
| LLaMA-3.1-8B | ICL-1 | $0.3301_{\pm0.037}$ | $0.6914_{\pm0.027}$ | $0.4126_{\pm0.026}$ | $0.5023_{\pm0.018}$ | $0.6037_{\pm0.015}$ | $0.9715_{\pm0.057}$ | $0.1735_{\pm0.066}$ |
| LLaMA-3.1-8B | ICL-3 | $\underline{0.3527}_{\pm0.033}$ | $0.6936_{\pm0.036}$ | $0.4217_{\pm0.017}$ | $0.5127_{\pm0.017}$ | $0.6202_{\pm0.009}$ | $0.9927_{\pm0.018}$ | $1.1672_{\pm0.047}$ |
| LLaMA-3.1-8B | NewModel | $0.2452_{\pm0.008}$ | $0.5347_{\pm0.005}$ | $0.2905_{\pm0.006}$ | $0.4097_{\pm0.005}$ | $0.4812_{\pm0.009}$ | $0.4571_{\pm0.021}$ | $\mathbf{1.2281}_{\pm0.041}$ |
| LLaMA-3.1-8B | WeightedComb | $0.3517_{\pm0.004}$ | $\underline{0.7040}_{\pm0.004}$ | $\underline{0.4249}_{\pm0.004}$ | $\underline{0.5181}_{\pm0.003}$ | $\underline{0.6206}_{\pm0.002}$ | $\underline{1.0947}_{\pm0.010}$ | $1.1737_{\pm0.015}$ |
| LLaMA-3.1-8B | TempNet | $0.3502_{\pm0.023}$ | $0.6927_{\pm0.006}$ | $0.4216_{\pm0.023}$ | $0.5027_{\pm0.017}$ | $0.6124_{\pm0.019}$ | $0.9625_{\pm0.025}$ | $1.1713_{\pm0.027}$ |
| LLaMA-3.1-8B | **Plugin (Ours)** | $\mathbf{0.3691}_{\pm0.013}$ | $\mathbf{0.7113}_{\pm0.002}$ | $\mathbf{0.4374}_{\pm0.004}$ | $\mathbf{0.5247}_{\pm0.002}$ | $\mathbf{0.6392}_{\pm0.009}$ | $\mathbf{1.1441}_{\pm0.030}$ | $\underline{1.1749}_{\pm0.034}$ |

*Table 2.* Performance comparison on Web NLG dataset. We show mean and standard deviation of the metrics over five seeds.

| Model | Method | BLEU | Rouge-1 | Rouge-2 | Rouge-L | METEOR | CIDEr | NIST |
|---|---|---|---|---|---|---|---|---|
| GPT2-M | Zeroshot | 0.0213 | 0.2765 | 0.1014 | 0.1872 | 0.2111 | 0.0479 | 0.2340 |
| GPT2-M | ICL-1 | $0.0317_{\pm0.013}$ | $0.3388_{\pm0.021}$ | $0.1318_{\pm0.013}$ | $0.2346_{\pm0.019}$ | $0.2876_{\pm0.042}$ | $0.0732_{\pm0.053}$ | $0.2715_{\pm0.042}$ |
| GPT2-M | ICL-3 | $0.0461_{\pm0.014}$ | $0.3388_{\pm0.018}$ | $0.1378_{\pm0.016}$ | $0.2291_{\pm0.010}$ | $\underline{0.3408}_{\pm0.027}$ | $0.0748_{\pm0.031}$ | $\mathbf{0.3283}_{\pm0.037}$ |
| GPT2-M | NewModel | $\underline{0.1071}_{\pm0.005}$ | $0.3260_{\pm0.010}$ | $0.1496_{\pm0.014}$ | $0.2724_{\pm0.013}$ | $0.2642_{\pm0.008}$ | $\underline{0.4327}_{\pm0.023}$ | $0.2916_{\pm0.031}$ |
| GPT2-M | WeightedComb | $0.0692_{\pm0.007}$ | $\underline{0.3593}_{\pm0.010}$ | $\underline{0.1568}_{\pm0.008}$ | $\underline{0.2834}_{\pm0.015}$ | $0.2379_{\pm0.030}$ | $0.1916_{\pm0.028}$ | $0.2996_{\pm0.037}$ |
| GPT2-M | TempNet | $0.1045_{\pm0.012}$ | $0.3526_{\pm0.014}$ | $0.1526_{\pm0.014}$ | $0.2731_{\pm0.018}$ | $0.3326_{\pm0.026}$ | $0.4237_{\pm0.033}$ | $0.3002_{\pm0.048}$ |
| GPT2-M | **Plugin (Ours)** | $\mathbf{0.1280}_{\pm0.007}$ | $\mathbf{0.4590}_{\pm0.005}$ | $\mathbf{0.2226}_{\pm0.005}$ | $\mathbf{0.3515}_{\pm0.006}$ | $\mathbf{0.3832}_{\pm0.010}$ | $\mathbf{0.7280}_{\pm0.039}$ | $\underline{0.3060}_{\pm0.017}$ |
| GPT2-XL | Zeroshot | 0.0317 | 0.2992 | 0.1321 | 0.2417 | 0.1969 | 0.0491 | 0.1826 |
| GPT2-XL | ICL-1 | $0.0510_{\pm0.024}$ | $0.3223_{\pm0.026}$ | $0.1526_{\pm0.016}$ | $0.2562_{\pm0.031}$ | $0.2591_{\pm0.009}$ | $0.1336_{\pm0.029}$ | $0.2235_{\pm0.033}$ |
| GPT2-XL | ICL-3 | $0.0744_{\pm0.016}$ | $0.3383_{\pm0.036}$ | $\underline{0.1682}_{\pm0.016}$ | $0.2651_{\pm0.028}$ | $\underline{0.3071}_{\pm0.014}$ | $0.1675_{\pm0.024}$ | $0.2550_{\pm0.021}$ |
| GPT2-XL | NewModel | $\underline{0.1071}_{\pm0.005}$ | $0.3260_{\pm0.010}$ | $0.1496_{\pm0.014}$ | $0.2724_{\pm0.013}$ | $0.2642_{\pm0.008}$ | $\underline{0.4327}_{\pm0.023}$ | $0.2916_{\pm0.031}$ |
| GPT2-XL | WeightedComb | $0.0636_{\pm0.006}$ | $\underline{0.3453}_{\pm0.007}$ | $0.1666_{\pm0.003}$ | $\underline{0.2782}_{\pm0.005}$ | $0.2871_{\pm0.006}$ | $0.2460_{\pm0.005}$ | $\mathbf{0.2981}_{\pm0.018}$ |
| GPT2-XL | TempNet | $0.0925_{\pm0.008}$ | $0.3357_{\pm0.009}$ | $0.1663_{\pm0.014}$ | $0.2764_{\pm0.011}$ | $0.3025_{\pm0.009}$ | $0.4226_{\pm0.013}$ | $0.2837_{\pm0.027}$ |
| GPT2-XL | **Plugin (Ours)** | $\mathbf{0.1673}_{\pm0.004}$ | $\mathbf{0.4616}_{\pm0.007}$ | $\mathbf{0.2527}_{\pm0.007}$ | $\mathbf{0.3757}_{\pm0.008}$ | $\mathbf{0.3895}_{\pm0.007}$ | $\mathbf{0.8987}_{\pm0.013}$ | $0.2646_{\pm0.003}$ |
| LLaMA-3.1-8B | Zeroshot | 0.1453 | 0.5278 | 0.3030 | 0.3982 | 0.4314 | 0.6991 | $\underline{0.2684}$ |
| LLaMA-3.1-8B | ICL-1 | $\underline{0.2166}_{\pm0.031}$ | $0.5944_{\pm0.027}$ | $0.3706_{\pm0.025}$ | $\underline{0.4667}_{\pm0.013}$ | $0.5651_{\pm0.045}$ | $\underline{1.5719}_{\pm0.024}$ | $0.2462_{\pm0.038}$ |
| LLaMA-3.1-8B | ICL-3 | $0.2031_{\pm0.027}$ | $0.5937_{\pm0.019}$ | $\underline{0.3821}_{\pm0.015}$ | $0.4653_{\pm0.024}$ | $\underline{0.5682}_{\pm0.046}$ | $1.3826_{\pm0.051}$ | $0.2469_{\pm0.045}$ |
| LLaMA-3.1-8B | NewModel | $0.1284_{\pm0.005}$ | $0.3506_{\pm0.009}$ | $0.1673_{\pm0.007}$ | $0.2879_{\pm0.009}$ | $0.2921_{\pm0.008}$ | $0.4999_{\pm0.030}$ | $\mathbf{0.2973}_{\pm0.008}$ |
| LLaMA-3.1-8B | WeightedComb | $0.1922_{\pm0.012}$ | $\underline{0.5986}_{\pm0.019}$ | $0.3612_{\pm0.012}$ | $0.4659_{\pm0.008}$ | $0.4470_{\pm0.030}$ | $1.1855_{\pm0.075}$ | $0.2575_{\pm0.020}$ |
| LLaMA-3.1-8B | TempNet | $0.2315_{\pm0.010}$ | $0.5916_{\pm0.015}$ | $0.3794_{\pm0.012}$ | $0.4620_{\pm0.010}$ | $0.5581_{\pm0.036}$ | $1.4826_{\pm0.043}$ | $0.2513_{\pm0.020}$ |
| LLaMA-3.1-8B | **Plugin (Ours)** | $\mathbf{0.2542}_{\pm0.004}$ | $\mathbf{0.6375}_{\pm0.005}$ | $\mathbf{0.3873}_{\pm0.005}$ | $\mathbf{0.4869}_{\pm0.007}$ | $\mathbf{0.5724}_{\pm0.004}$ | $\mathbf{1.5911}_{\pm0.046}$ | $0.2590_{\pm0.003}$ |

model quality, *Plugin*'s complexity, and architecture choices. Section 7.4 shows qualitative analysis and case studies.

We evaluate *Plugin* on four text generation benchmarks: (a) E2E NLG (Dušek et al., 2020), (b) Web NLG (Gardent et al., 2017), (c) CommonGen (Lin et al., 2020), and (d) the Adidas product description dataset (adi, 2023). For the first three datasets, we use the train-validation-test splits from the Transformers library (Wolf, 2020). To introduce distribution shifts, we filter Web NLG's training data to include only *infrastructure* descriptions, while validation and test sets retain *person* descriptions. Similarly, CommonGen's training set is restricted to samples having *man*, while validation and test sets remain unchanged. Details of this setup are in Section 7.4. The Adidas dataset is split into validation and test sets. Data statistics are provided in Table 6, Appendix C.1.

## 7.1. Text Generation Performance Comparison

We evaluate *Plugin* on the text generation task using only the validation and test splits of all four datasets, reserving the train split for ablation studies (Section 7.3). *Plugin* and baseline models are trained on the small validation set, with performance measured on the test set. Additionally, we allocate 40% of the validation data as *hyper-validation* for cross-validation of hyperparameters.

Performance is reported using seven standard natural language generation metrics: (a) BLEU (Papineni et al., 2002), (b) ROUGE-1 (Lin, 2004), (c) ROUGE-2 (Lin, 2004), (d) ROUGE-L (Lin & Och, 2004), (e) METEOR (Banerjee & Lavie, 2005), (f) CIDEr (Vedantam et al., 2015), and (g) NIST (Doddington, 2002). All experiments are repeated

*Table 3.* Performance comparison on CommonGen dataset. We show mean and standard deviation of the metrics over five seeds.

| Model | Method | BLEU | Rouge-1 | Rouge-2 | Rouge-L | METEOR | CIDEr | NIST |
|---|---|---|---|---|---|---|---|---|
| GPT2-M | Zeroshot | 0.0153 | 0.2216 | 0.0409 | 0.1527 | 0.2848 | 0.0001 | 0.3686 |
| GPT2-M | ICL-1 | $0.0157_{\pm0.013}$ | $0.2580_{\pm0.024}$ | $0.0362_{\pm0.096}$ | $0.1388_{\pm0.102}$ | $0.2871_{\pm0.107}$ | $0.0222_{\pm0.076}$ | $0.3704_{\pm0.101}$ |
| GPT2-M | ICL-3 | $0.0552_{\pm0.010}$ | $0.3610_{\pm0.019}$ | $0.1248_{\pm0.045}$ | $0.2680_{\pm0.089}$ | $0.4079_{\pm0.133}$ | $0.1366_{\pm0.125}$ | $0.5340_{\pm0.087}$ |
| GPT2-M | NewModel | $0.1260_{\pm0.007}$ | $0.4106_{\pm0.016}$ | $0.1683_{\pm0.013}$ | $0.3740_{\pm0.009}$ | $0.3600_{\pm0.024}$ | $0.4570_{\pm0.058}$ | $0.7113_{\pm0.025}$ |
| GPT2-M | WeightedComb | $0.0567_{\pm0.005}$ | $0.3918_{\pm0.010}$ | $0.1353_{\pm0.005}$ | $0.3280_{\pm0.010}$ | $0.2929_{\pm0.016}$ | $0.2623_{\pm0.042}$ | $0.4353_{\pm0.028}$ |
| GPT2-M | TempNet | $0.1248_{\pm0.005}$ | $0.4048_{\pm0.015}$ | $0.1528_{\pm0.015}$ | $0.3526_{\pm0.014}$ | $0.3883_{\pm0.027}$ | $0.4492_{\pm0.023}$ | $0.4037_{\pm0.058}$ |
| GPT2-M | **Plugin (Ours)** | $\mathbf{0.1366}_{\pm0.003}$ | $\mathbf{0.4533}_{\pm0.007}$ | $\mathbf{0.1878}_{\pm0.003}$ | $\mathbf{0.3934}_{\pm0.006}$ | $\mathbf{0.4095}_{\pm0.011}$ | $\mathbf{0.5572}_{\pm0.022}$ | $0.6395_{\pm0.061}$ |
| GPT2-XL | Zeroshot | 0.0317 | 0.2992 | 0.1321 | 0.2417 | 0.1969 | 0.0491 | 0.1826 |
| GPT2-XL | ICL-1 | $0.0508_{\pm0.023}$ | $0.3201_{\pm0.035}$ | $0.1526_{\pm0.097}$ | $0.2562_{\pm0.103}$ | $0.2591_{\pm0.089}$ | $0.1336_{\pm0.092}$ | $0.2235_{\pm0.069}$ |
| GPT2-XL | ICL-3 | $0.0744_{\pm0.011}$ | $0.3383_{\pm0.014}$ | $0.1682_{\pm0.030}$ | $0.2651_{\pm0.072}$ | $0.3071_{\pm0.073}$ | $0.1675_{\pm0.066}$ | $0.2550_{\pm0.047}$ |
| GPT2-XL | NewModel | $0.1260_{\pm0.007}$ | $0.4106_{\pm0.016}$ | $0.1683_{\pm0.013}$ | $0.3740_{\pm0.009}$ | $0.3600_{\pm0.024}$ | $0.4570_{\pm0.058}$ | $0.7113_{\pm0.025}$ |
| GPT2-XL | WeightedComb | $0.0614_{\pm0.020}$ | $0.3364_{\pm0.024}$ | $0.1347_{\pm0.009}$ | $0.2969_{\pm0.019}$ | $0.2921_{\pm0.018}$ | $0.2763_{\pm0.010}$ | $0.3352_{\pm0.051}$ |
| GPT2-XL | TempNet | $0.1154_{\pm0.020}$ | $0.3937_{\pm0.026}$ | $0.1482_{\pm0.017}$ | $0.3625_{\pm0.013}$ | $0.3389_{\pm0.019}$ | $0.4376_{\pm0.018}$ | $0.5927_{\pm0.047}$ |
| GPT2-XL | **Plugin (Ours)** | $\mathbf{0.1791}_{\pm0.014}$ | $\mathbf{0.4932}_{\pm0.007}$ | $\mathbf{0.2288}_{\pm0.004}$ | $\mathbf{0.4347}_{\pm0.007}$ | $\mathbf{0.4702}_{\pm0.006}$ | $\mathbf{0.7283}_{\pm0.012}$ | $0.6554_{\pm0.038}$ |
| LLaMA-3.1-8B | Zeroshot | 0.0643 | 0.2776 | 0.1181 | 0.2488 | 0.3857 | 0.3155 | 0.3347 |
| LLaMA-3.1-8B | ICL-1 | $0.0615_{\pm0.027}$ | $0.2697_{\pm0.033}$ | $0.1158_{\pm0.062}$ | $0.2469_{\pm0.087}$ | $0.3822_{\pm0.069}$ | $0.3005_{\pm0.072}$ | $0.3059_{\pm0.094}$ |
| LLaMA-3.1-8B | ICL-3 | $0.0635_{\pm0.016}$ | $0.2748_{\pm0.024}$ | $0.1225_{\pm0.018}$ | $0.3120_{\pm0.047}$ | $0.4012_{\pm0.029}$ | $0.3250_{\pm0.022}$ | $0.3794_{\pm0.034}$ |
| LLaMA-3.1-8B | NewModel | $0.0753_{\pm0.004}$ | $0.3716_{\pm0.005}$ | $0.1122_{\pm0.003}$ | $0.3404_{\pm0.004}$ | $0.2665_{\pm0.006}$ | $0.1919_{\pm0.015}$ | $0.6900_{\pm0.046}$ |
| LLaMA-3.1-8B | WeightedComb | $0.1789_{\pm0.005}$ | $0.3485_{\pm0.012}$ | $0.1797_{\pm0.008}$ | $0.2981_{\pm0.012}$ | $0.3637_{\pm0.011}$ | $0.5503_{\pm0.046}$ | $0.5450_{\pm0.020}$ |
| LLaMA-3.1-8B | TempNet | $0.1524_{\pm0.008}$ | $0.3372_{\pm0.015}$ | $0.1524_{\pm0.010}$ | $0.3298_{\pm0.017}$ | $0.3676_{\pm0.015}$ | $0.3986_{\pm0.033}$ | $0.5286_{\pm0.023}$ |
| LLaMA-3.1-8B | **Plugin (Ours)** | $\mathbf{0.2665}_{\pm0.010}$ | $\mathbf{0.5800}_{\pm0.002}$ | $\mathbf{0.3139}_{\pm0.005}$ | $\mathbf{0.5037}_{\pm0.004}$ | $\mathbf{0.5829}_{\pm0.003}$ | $\mathbf{1.0876}_{\pm0.020}$ | $\mathbf{0.7031}_{\pm0.007}$ |

*Table 4.* Performance comparison on Adidas dataset. We show mean and standard deviation of the metrics over five seeds.

| Model | Method | BLEU | Rouge-1 | Rouge-2 | Rouge-L | METEOR | CIDEr | NIST |
|---|---|---|---|---|---|---|---|---|
| GPT2-M | Zeroshot | 0.0046 | 0.2488 | 0.0189 | 0.1353 | 0.1653 | 0.0312 | 0.6860 |
| GPT2-M | ICL-1 | $0.0088_{\pm0.054}$ | $0.2667_{\pm0.047}$ | $0.0247_{\pm0.66}$ | $0.1358_{\pm0.041}$ | $0.1762_{\pm0.028}$ | $0.0464_{\pm0.089}$ | $0.6793_{\pm0.078}$ |
| GPT2-M | ICL-3 | $0.0121_{\pm0.047}$ | $0.2693_{\pm0.028}$ | $0.0262_{\pm0.054}$ | $0.1470_{\pm0.020}$ | $0.1806_{\pm0.030}$ | $0.0415_{\pm0.104}$ | $0.7037_{\pm0.081}$ |
| GPT2-M | NewModel | $0.0515_{\pm0.016}$ | $0.2690_{\pm0.014}$ | $\mathbf{0.0637}_{\pm0.014}$ | $\mathbf{0.1697}_{\pm0.008}$ | $0.1918_{\pm0.013}$ | $0.0550_{\pm0.086}$ | $0.6682_{\pm0.047}$ |
| GPT2-M | WeightedComb | $\mathbf{0.0565}_{\pm0.014}$ | $0.2630_{\pm0.028}$ | $0.0495_{\pm0.018}$ | $0.1565_{\pm0.015}$ | $0.1938_{\pm0.019}$ | $0.0585_{\pm0.088}$ | $0.6456_{\pm0.156}$ |
| GPT2-M | TempNet | $0.0442_{\pm0.017}$ | $0.2672_{\pm0.019}$ | $0.0482_{\pm0.022}$ | $0.1582_{\pm0.020}$ | $0.1902_{\pm0.017}$ | $0.0525_{\pm0.031}$ | $0.6533_{\pm0.098}$ |
| GPT2-M | **Plugin (Ours)** | $0.0486_{\pm0.006}$ | $\mathbf{0.2766}_{\pm0.002}$ | $0.0515_{\pm0.007}$ | $0.1684_{\pm0.005}$ | $\mathbf{0.1994}_{\pm0.004}$ | $\mathbf{0.0626}_{\pm0.017}$ | $\mathbf{0.7919}_{\pm0.024}$ |
| GPT2-XL | Zeroshot | 0.0075 | 0.2309 | 0.0278 | 0.1438 | 0.1487 | 0.0184 | 0.4956 |
| GPT2-XL | ICL-1 | $0.0109_{\pm0.039}$ | $0.2567_{\pm0.082}$ | $0.0265_{\pm0.054}$ | $0.1519_{\pm0.038}$ | $0.1649_{\pm0.052}$ | $0.0318_{\pm0.171}$ | $0.5133_{\pm0.162}$ |
| GPT2-XL | ICL-3 | $0.0295_{\pm0.037}$ | $0.2509_{\pm0.071}$ | $0.0395_{\pm0.043}$ | $0.1536_{\pm0.039}$ | $0.1658_{\pm0.041}$ | $0.0321_{\pm0.109}$ | $0.5176_{\pm0.116}$ |
| GPT2-XL | NewModel | $0.0515_{\pm0.016}$ | $0.2690_{\pm0.014}$ | $0.0637_{\pm0.014}$ | $0.1697_{\pm0.008}$ | $0.1918_{\pm0.013}$ | $0.0550_{\pm0.086}$ | $\mathbf{0.6682}_{\pm0.047}$ |
| GPT2-XL | WeightedComb | $0.0567_{\pm0.016}$ | $0.2210_{\pm0.027}$ | $0.0714_{\pm0.015}$ | $0.1550_{\pm0.024}$ | $0.1674_{\pm0.017}$ | $0.0183_{\pm0.117}$ | $0.4105_{\pm0.109}$ |
| GPT2-XL | TempNet | $0.0539_{\pm0.018}$ | $0.2598_{\pm0.026}$ | $0.0686_{\pm0.014}$ | $0.1562_{\pm0.019}$ | $0.1863_{\pm0.029}$ | $0.0462_{\pm0.120}$ | $0.5263_{\pm0.117}$ |
| GPT2-XL | **Plugin (Ours)** | $\mathbf{0.0600}_{\pm0.017}$ | $\mathbf{0.2710}_{\pm0.025}$ | $\mathbf{0.0722}_{\pm0.018}$ | $\mathbf{0.1725}_{\pm0.017}$ | $\mathbf{0.1995}_{\pm0.018}$ | $\mathbf{0.1195}_{\pm0.138}$ | $0.6375_{\pm0.120}$ |
| LLaMA-3.1-8B | Zeroshot | 0.0120 | 0.2470 | 0.0318 | 0.1493 | 0.1526 | 0.0424 | 0.5285 |
| LLaMA-3.1-8B | ICL-1 | $0.0220_{\pm0.044}$ | $0.2472_{\pm0.072}$ | $0.0405_{\pm0.068}$ | $0.1434_{\pm0.057}$ | $0.1686_{\pm0.041}$ | $0.0555_{\pm0.133}$ | $0.5078_{\pm0.142}$ |
| LLaMA-3.1-8B | ICL-3 | $0.0177_{\pm0.041}$ | $0.2385_{\pm0.065}$ | $0.0364_{\pm0.071}$ | $0.1408_{\pm0.030}$ | $0.1712_{\pm0.029}$ | $0.0587_{\pm0.102}$ | $0.5775_{\pm0.145}$ |
| LLaMA-3.1-8B | NewModel | $0.0506_{\pm0.011}$ | $0.2700_{\pm0.011}$ | $0.0634_{\pm0.006}$ | $0.1749_{\pm0.006}$ | $\mathbf{0.1995}_{\pm0.009}$ | $0.0575_{\pm0.051}$ | $\mathbf{0.6570}_{\pm0.072}$ |
| LLaMA-3.1-8B | WeightedComb | $0.0357_{\pm0.017}$ | $0.2583_{\pm0.014}$ | $0.0661_{\pm0.015}$ | $0.1560_{\pm0.011}$ | $0.1706_{\pm0.016}$ | $0.0745_{\pm0.086}$ | $0.5927_{\pm0.077}$ |
| LLaMA-3.1-8B | TempNet | $0.0472_{\pm0.016}$ | $0.2647_{\pm0.022}$ | $0.0625_{\pm0.012}$ | $0.1625_{\pm0.020}$ | $0.1857_{\pm0.013}$ | $0.0586_{\pm0.103}$ | $0.5926_{\pm0.137}$ |
| LLaMA-3.1-8B | **Plugin (Ours)** | $\mathbf{0.0611}_{\pm0.018}$ | $\mathbf{0.2714}_{\pm0.029}$ | $\mathbf{0.0742}_{\pm0.020}$ | $\mathbf{0.1759}_{\pm0.019}$ | $0.1990_{\pm0.020}$ | $\mathbf{0.1293}_{\pm0.152}$ | $0.6361_{\pm0.134}$ |

over five random seeds, and we report the mean and standard deviation for each metric.

We compare *Plugin* with the following baselines: (a) **Zeroshot**: The black-box model directly performs text generation without additional adaptation. (b) **ICL-1** (Long et al., 2023): One randomly selected validation sample is used as an in-context example. (c) **ICL-3** (Long et al., 2023): Three randomly selected validation samples are used as in-context examples. (d) **NewModel**: A new language model is trained using the validation data. (e) **WeightedComb** (Liu et al., 2021): A new model is trained alongside the black-box model, with token probabilities computed as $\alpha\boldsymbol{n} + (1-\alpha)\boldsymbol{b}$, where $\boldsymbol{n}$ represents the probabilities from the new model and $\alpha$ is cross-validated in $\{0.25, 0.50, 0.75\}$. (f) **TempNet** (Qiu et al., 2024), a recent logit-scaling approach that

learns a global temperature per input and uniformly scales logits during generation. Since the black-box model weights are inaccessible, fine-tuning-based approaches are not applicable in our setting. Nonetheless, we include a comparison with LoRA in Appendix C.4 for completeness. This highlights *Plugin*'s competitiveness despite operating under stricter access constraints than required for PEFT.

All methods use the same prompts where applicable (Appendix C.2) and employ greedy decoding. The base (black-box) models used are GPT2-M (Radford et al., 2019), GPT2-XL (Radford et al., 2019), and LLaMA-3.1-8B (Dubey et al., 2024). *NewModel*, *WeightedComb*, and the reweighting model in *Plugin* share the same architecture. For GPT-based models, these use a Transformer encoder with one hidden layer and default configurations. For LLaMA-based mod-

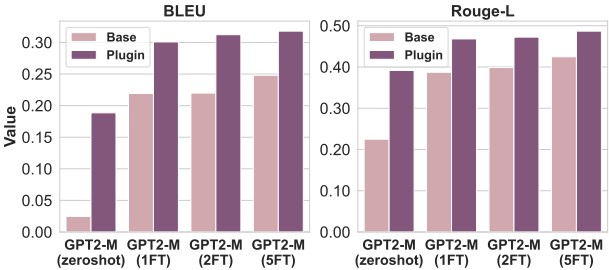

*Figure 2. Plugin* with increasingly fine-tuned GPT2-M models on the E2E NLG dataset. Results demonstrate that as the quality of the base model improves, the performance of the *Plugin* improves.

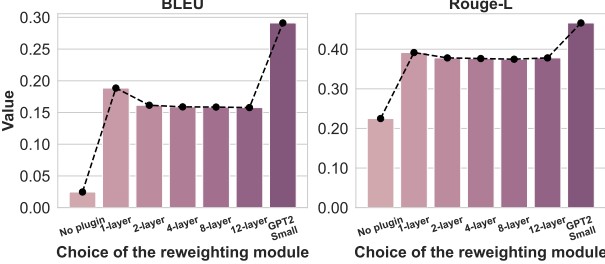

*Figure 3.* Performance of GPT2-M with varying reweighting model complexities on E2E NLG (BLEU, ROUGE-L). A single-layer reweighting model yields significant gains, while additional layers degrade performance due to overfitting. Initializing with GPT2-Small as the reweighting model improves performance, demonstrating the benefits of leveraging small pretrained models.

els, the architecture consists of a Transformer encoder with one hidden layer, 256 hidden size, 1024 intermediate size, and one attention head. Learning rate and weight decay are cross-validated over $\{1e-5, 5e-5, 1e-4, 5e-4, 1e-3, 5e-3\}$ and $\{0.01, 0.1, 1, 10\}$, respectively. Models are trained using AdamW with warmup followed by linear decay, and early stopping is applied if the *hyper-validation* loss does not decrease for five consecutive epochs.

As shown in Tables 1–4 (the best is bold, the second best is underlined), *Plugin* outperforms baselines across nearly all datasets, black-box models, and evaluation metrics. *New-Model* occasionally achieves higher NIST scores due to increased repetition of less-frequent input tokens, but this comes at the cost of coherence, as reflected by other metrics. *WeightedComb* does not perform well, indicating one combination for all tokens is not a good modeling choice. *TempNet*, which learns a single temperature per input and uniformly scales logits during generation, also underperforms. In contrast, *Plugin* reweights logits at each timestep, enabling finer, context-sensitive adjustments.

We note that the absolute numbers may not appear competitive with state-of-the-art results, because (a) we restrict to greedy decoding (Section 4.2), and (b) Web NLG and CommonGen use distribution-shifted subsets.

We also conduct a human evaluation on 100 Adidas dataset samples, where three subjects compare outputs from *Plugin*

*Table 5.* Performance comparison of BDPL and BDPL + Plugin.

| Dataset | Method | BLEU | Rouge-L | METEOR | CIDEr | NIST |
|---|---|---|---|---|---|---|
| E2E NLG | BDPL | 0.2287 | 0.3922 | 0.4628 | 0.4216 | 0.8625 |
| | **BDPL + Plugin** | **0.4527** | **0.6027** | **0.6214** | **0.7002** | **2.0817** |
| WEB NLG | BDPL | 0.1024 | 0.3017 | 0.3527 | 0.4321 | 0.2631 |
| | **BDPL + Plugin** | **0.2137** | **0.5928** | **0.5766** | **1.0826** | **0.6142** |
| CommonGen | BDPL | 0.1023 | 0.2936 | 0.3362 | 0.2517 | 0.4226 |
| | **BDPL + Plugin** | **0.2614** | **0.5241** | **0.5016** | **0.8251** | **0.9261** |
| Adidas | BDPL | 0.0417 | 0.1710 | 0.1826 | 0.0861 | 0.6034 |
| | **BDPL + Plugin** | **0.0623** | **0.1759** | **0.2148** | **0.1325** | **0.7024** |

and *ICL-3* using LLaMA-3.1 as the base model. Evaluators select the prediction closest to the ground truth, with *Plugin* preferred in 81% of cases. Details are in Appendix C.7.

### 7.2. Plugin as a Wrapper

If logit access is available, *Plugin* can be applied on top of any prompt-based method using its best-found prompt. For example, our **Zeroshot** prompt is reused across methods. We also apply *Plugin* to the Black-box Discrete Prompt Learning (BDPL) approach from Diao et al. (2022), following their recommended 75 API call budget. Table 5 shows results on all datasets with GPT2-XL as the base model. *Plugin* outperforms BDPL (see Tables 1–4), and their combination yields further gains, underscoring the utility of logit-level access in strengthening prompt-based methods.

### 7.3. Ablation Study

We now show ablation studies that reflect various aspects of the *Plugin* model. We display the results using GPT2-M as base model on the E2E NLG dataset. The observation is similar on other base models and datasets (Appendix C.5).

**Impact of Base Model Quality.** We fine-tune GPT2-M for varying epochs, denoted as 1FT (one epoch), 2FT (two epochs), and 5FT (five epochs), and train a *Plugin* model for each. Figure 2 shows that as the base model's task-specific quality improves, the *Plugin's* performance improves.

**Complexity of the Reweighting Model in *Plugin*.** We train *Plugin* models with reweighting architectures varying from 1 to 12 transformer layers while keeping other configurations unchanged. Additionally, we train a variant where the reweighting model is initialized with GPT2-Small. As shown in Figure 3, a single-layer reweighting model yields significant improvements over the base GPT2-M model, while additional layers (e.g., 2, 4, 8, 12) offer diminishing returns and slight performance decline due to overfitting on the small validation set of E2E NLG. This suggests that more data is required for learning complex reweighting models. Notably, initializing with a pretrained GPT2-Small substantially improves performance, underscoring the advantage of using small pretrained models for reweighting due to their inherent autoregressive properties.

### 7.4. Qualitative Analysis and Case Study

***Plugin* adapting to distribution shift.** We evaluate *Plugin* on distribution-shifted Web NLG and CommonGen us-

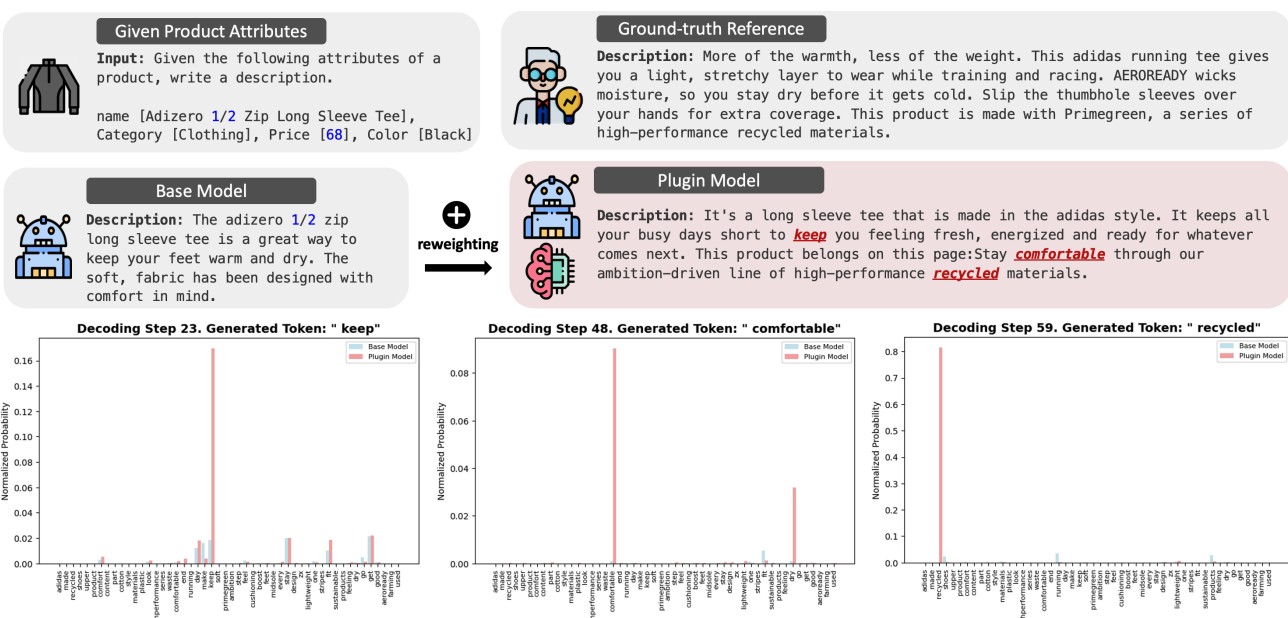

Figure 4. Comparison of the adaptation ability between the base model and *Plugin* on Adidas dataset. *Plugin*, enhanced with a reweighting model, generates text that better aligns with the "*Adidas domain*". The bottom row illustrates token probabilities for key Adidas-related words at different decoding steps, showing how the reweighting model influences token selection.

ing LLaMA-3.1-8B as the base model. Web NLG training data contains only *Infrastructure* concepts, while validation and test sets include *Person* concepts. Similarly, Common-Gen training data features *man*, whereas validation and test sets contain both *man* and *woman*. The base model is fine-tuned on the training data, and *Plugin* is trained on validation data using the fine-tuned model as the base. These settings reflect different degrees of domain shift, even *adversarial* to some extent as the training distributions induce biases (e.g., overemphasis on infrastructure or male-related concepts), and *Plugin* is tasked with correcting them during inference.

Using GPT-4o (Hurst et al., 2024) as an evaluator, the fine-tuned Web NLG model generates only 17.99% *Person*-related sentences, while *Plugin* increases this to 71.34%. On CommonGen, the fine-tuned model generates 10.37% *Woman*-related sentences, whereas *Plugin* improves this to 31.92%. These results highlight *Plugin*'s ability to adapt under distribution shift and mitigate biases in the base model.

**Case study: *Plugin* adapting to domain (extreme distirbution shifts).** We examine token probabilities during inference for LLaMA-3.1-8B and *Plugin* to assess domain adaptation in the Adidas dataset, which features product-centric language and a brand-specific tone that diverge significantly from the general pretraining distribution of the black-box LLMs. This experimental setup can also be viewed as *extreme distribution shift*. Removing stopwords, we extract the top-50 most frequent words, defining the "*Adidas domain*". Figure 4 illustrates this adaptation: the first row shows product attributes and ground-truth references; the second row compares outputs from the base model (left)

and *Plugin* (right); the third row visualizes model probabilities for "*Adidas domain*" words at three decoding steps.

As seen in Figure 4, *Plugin* dynamically reweights probabilities to align with domain-specific language. At step 23, "keep" is significantly upweighted. At step 48, "comfortable" and "dry" gain prominence over "fit," which the base model favors. At step 59, "recycled" is preferred by *Plugin*, aligning with the ground truth, while the base model favors "running" and "products". This demonstrates that *Plugin* effectively steers generation toward domain-specific terminology, whereas the base model, trained on broad corpora, lacks inherent domain preference.

Unlike methods that prune or suppress tokens, *Plugin* softly reweights token probabilities without eliminating any vocabulary candidates. This preserves full coverage while amplifying domain-specific terms. To quantify this, we measure the total occurrences of the top-50 "*Adidas domain*" words in generated outputs: *Plugin* includes 25.6% of these terms compared to 13.8% in the base model, indicating substantially improved alignment with domain language.

## 8. Conclusion

We propose *Plugin*, a token-level probability reweighting framework that adapts black-box LLMs using only logits and small task-specific data. Framing next-token prediction as a label noise correction problem, we demonstrate both theoretical guarantees and empirical effectiveness across multiple datasets and models. Our findings highlight the potential of logit-based adaptation and advocate for broader access to token logits in closed-source LLMs.

## Acknowledgements

HW acknowledges support by Fonds de recherche du Québec – Nature et technologies (FRQNT) and Borealis AI. SK acknowledges support by NSF 2046795 and 2205329, IES R305C240046, the MacArthur Foundation, Stanford HAI, OpenAI, and Google.

## Impact Statement

This work introduces a powerful middle ground between fully black-box APIs and fully white-box access to large language models (LLMs), addressing a critical constraint faced by developers: the inability to adapt models when weights and architecture are inaccessible. By leveraging token-level logits—without requiring access to model weights or architecture—our approach enables meaningful adaptation of closed-source LLMs for domain-specific tasks. This has far-reaching implications for both research and industry: it empowers developers to customize models within privacy-preserving, IP-sensitive environments while ensuring greater control, transparency, and safety. Our findings advocate for broader logit access as a scalable, secure, and effective interface—bridging the gap between usability and protection of proprietary models—and open new possibilities for equitable, context-aware language generation in real-world applications.

While *Plugin* effectively adapts black-box LLMs, it has some limitations, too. Since it only reweights token probabilities without modifying internal representations or embeddings, it may struggle with tasks requiring deep structural adaptations, such as executing complex reasoning. Further research on this aspect is needed. Additionally, although *Plugin* avoids full fine-tuning, training a separate reweighting model introduces computational overhead compared to prompt tuning or in-context learning, with efficiency depending on the complexity of the reweighting model and the availability of task-specific data.

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

# A. Algorithm Details

We provide summarized form of the training and inference algorithm for the *Plugin* model below.

---

**Algorithm 1** Training and Inference for the Plugin Model

---

**Input:** Black-box model $B$, reweighting model $R$, clean training data $\mathcal{D}$, vocabulary $V$
**Output:** Plugin model predictions $\boldsymbol{x}_{1:T}$ for a given sequence

1: **Training Phase:**
2: **for** each sequence $s \in \mathcal{D}$ **do**
3:     Compute token probabilities $\{\boldsymbol{b}_1, \boldsymbol{b}_2, \ldots, \boldsymbol{b}_m\}$ using $B$.
4:     Compute token probabilities $\{\boldsymbol{r}_1, \boldsymbol{r}_2, \ldots, \boldsymbol{r}_m\}$ using $R$.
5:     Combine probabilities: $\boldsymbol{p}_i = \frac{\boldsymbol{b}_i \odot \boldsymbol{r}_i}{\|\boldsymbol{b}_i \odot \boldsymbol{r}_i\|_1}$ for $i \in [m]$.
6:     Compute sequence loss $\ell_s = -\frac{1}{m} \sum_{i=1}^{m} \sum_{j=1}^{|V|} \log(\boldsymbol{p}_i) \odot \boldsymbol{e}_j$.
7:     Update parameters of $R$ using back-propagation. Freeze $B$.
8: **end for**
9: **Inference Phase:**
10: Initialize sequence $\boldsymbol{x}_{1:T} = \{\}$.
11: **for** each token position $t = 1$ to $T$ **do**
12:     Compute token probabilities $\boldsymbol{b}_t$ using $B$.
13:     Compute token probabilities $\boldsymbol{r}_t$ using $R$.
14:     Combine probabilities: $\boldsymbol{p}_t = \frac{\boldsymbol{b}_t \odot \boldsymbol{r}_t}{\|\boldsymbol{b}_t \odot \boldsymbol{r}_t\|_1}$.
15:     Predict token: $\boldsymbol{x}_t = \operatorname{argmax}_V(\boldsymbol{p}_t)$.
16:     Append $\boldsymbol{x}_t$ to $\boldsymbol{x}_{1:T}$.
17: **end for**
18: **Return:** $\boldsymbol{x}_{1:T}$

---

# B. Proof of Main Convergence Theorem

We define the following assumption on the smoothness and regularity of the loss function.

**Assumption B.1.** We assume the following assumptions hold with probability 1:

1. **(Convexity of $\ell_s$):** The loss function $\ell_s$ is convex for all time $s \in [t]$.

2. **(Smoothness of $\ell_s$):** The $\ell_s$ is smooth such that the first, second, and third derivatives exist at all interior points in $\boldsymbol{\Theta}$.

3. **(Regularity Conditions):**

    (a) $\boldsymbol{\Theta}$ is compact and $\ell_s(\boldsymbol{\theta})$ is bounded for all $\boldsymbol{\theta} \in \boldsymbol{\Theta}$ and for all $s \in [t]$.
    (b) $\boldsymbol{\theta}_*$ is an interior point in $\boldsymbol{\Theta}$.
    (c) $\nabla^2 \ell_s(\boldsymbol{\theta}_*)$ is positive definite, for all $s \in [t]$ .
    (d) There exists a neighborhood $\mathcal{B}$ of $\boldsymbol{\theta}_*$ and a constant $C_1$, such that $\nabla^2 \ell_s(\boldsymbol{\theta})$ is $C_1$-Lipschitz. Hence, we have that $\left\| \nabla^2 \ell_s(\boldsymbol{\theta}) - \nabla^2 \ell_s\left(\boldsymbol{\theta}'\right) \right\|_* \leq C_1 \left\| \boldsymbol{\theta} - \boldsymbol{\theta}' \right\|_{\nabla^2 \mathcal{L}_s(\boldsymbol{\theta}_*)}$, for $\boldsymbol{\theta}, \boldsymbol{\theta}'$ in this neighborhood.

4. **(Concentration at $\boldsymbol{\theta}_*$):** We further assume that $\left\| \nabla \ell_s\left(\boldsymbol{\theta}_*\right) \right\|_{(\nabla^2 \mathcal{L}_s(\boldsymbol{\theta}_*))^{-1}} \leq C_2$ hold with probability one.

**Lemma B.2.** *(Proposition 2 of ([Hsu et al., 2012](#)))* *Let $\mathbf{u}_1, \ldots, \mathbf{u}_n$ be a martingale difference vector sequence (i.e., $\mathbb{E}\left[\mathbf{u}_i \mid \mathbf{u}_1, \ldots, \mathbf{u}_{i-1}\right] = 0$ for all $i = 1, \ldots, n$ ) such that*

$$\sum_{i=1}^{n} \mathbb{E}\left[\|\mathbf{u}_i\|^2 \mid \mathbf{u}_1, \ldots, \mathbf{u}_{i-1}\right] \leq v \quad \text{and} \quad \|\mathbf{u}_i\| \leq b$$

*for all $i = 1, \ldots, n$, almost surely. For all $t > 0$*

$$\Pr\left[\left\|\sum_{i=1}^{n} \mathbf{u}_i\right\| > \sqrt{v} + \sqrt{8vt} + (4/3)bt\right] \leq e^{-t}$$

**Lemma B.3.** *The probability that $\|\nabla\widehat{\mathcal{L}}_t(\boldsymbol{\theta}_*)\|_{(\nabla^2\mathcal{L}(\boldsymbol{\theta}_*))^{-1}}$ crosses the threshold $\sqrt{\frac{c\gamma\log(dt)}{t}} > 0$ is bounded by*

$$\mathbb{P}\left(\|\nabla\widehat{\mathcal{L}}_t(\boldsymbol{\theta}_*)\|_{(\nabla^2\mathcal{L}_t(\boldsymbol{\theta}_*))^{-1}} \geq C_2\sqrt{\frac{c\gamma\log(dt)}{t}}\right) \leq \frac{1}{t^{c\gamma}}.$$

*Proof.* Define $\mathbf{u_s} := \nabla(Y_s - f_{I_s}(\boldsymbol{\theta}_*; x_i, x_j, \mathcal{F}^{s-1}))^2$. Then we have $\mathbf{u}_1, \mathbf{u}_2, \ldots, \mathbf{u}_t$ as random vectors such that

$$\mathbb{E}\left[\left\|\sum_{s=1}^{t}\mathbf{u_s}\right\|^2_{(\nabla^2\mathcal{L}_t(\boldsymbol{\theta}_*))^{-1}}\Bigg| \mathbf{u}_1, \ldots, \mathbf{u}_{s-1}\right] = \mathbb{E}\left[\sum_{s=1}^{t}\mathbf{u_s}^\top\left(\nabla^2\mathcal{L}_t(\boldsymbol{\theta}_*)\right)^{-1}\mathbf{u_s} \mid \mathbf{u}_1, \ldots, \mathbf{u}_{s-1}\right] \leq tC_2^2$$

Also we have that $\|\mathbf{u_s}\| \leq C_2$. Finally we have that

$$\mathbb{E}[\nabla_{\boldsymbol{\theta}=\boldsymbol{\theta}_*}\mathbf{u_s}] = -2\sum_{s=1}^{t}p_{\widehat{\boldsymbol{\theta}}_{s-1}}(f_{I_s}(\boldsymbol{\theta}_*; x_i, x_j, \mathcal{F}^{s-1}) - f_{I_s}(\boldsymbol{\theta}_*; x_i, x_j, \mathcal{F}^{s-1})\nabla_{\boldsymbol{\theta}=\boldsymbol{\theta}_*}f_{I_s}(\boldsymbol{\theta}_*; x_i, x_j, \mathcal{F}^{s-1}) = 0.$$

Then following Lemma B.2 and by setting $\epsilon = c\gamma\log(dt)$ we can show that

$$\mathbb{P}\left(\|\frac{1}{t}\sum_{s=1}^{t}\mathbf{u_s}\|^2_{(\nabla^2\mathcal{L}_t(\boldsymbol{\theta}_*))^{-1}} - \mathbb{E}\left[\|\frac{1}{t}\sum_{s=1}^{t}\mathbf{u_s}\|^2_{(\nabla^2\mathcal{L}_t(\boldsymbol{\theta}_*))^{-1}}\right] > \frac{1}{t}\sqrt{8tC_2^2\epsilon} + \frac{4C_2}{3\epsilon}\right)$$

$$= \mathbb{P}\left(\|\frac{1}{t}\sum_{s=1}^{t}\mathbf{u_s}\|^2_{(\nabla^2\mathcal{L}_t(\boldsymbol{\theta}_*))^{-1}} > C_1^2 + C_2\sqrt{\frac{8\epsilon}{t}} + \frac{4C_2}{3\epsilon}\right)$$

$$\leq \mathbb{P}\left(\|\sum_{s=1}^{t}\mathbf{u_s}\|^2_{(\nabla^2\mathcal{L}_t(\boldsymbol{\theta}_*))^{-1}} > C_2\sqrt{\frac{8\epsilon}{t}}\right) = \mathbb{P}\left(\|\sum_{s=1}^{t}\mathbf{u_s}\|^2_{(\nabla^2\mathcal{L}_t(\boldsymbol{\theta}_*))^{-1}} > 4C_2\sqrt{\frac{c\gamma\log(dt)}{t}}\right)$$

$$\leq \exp(-c\gamma\log(dt)) = \left(\frac{1}{dt}\right)^{c\gamma} \leq \frac{1}{t^{c\gamma}}$$

The claim of the lemma follows. $\square$

**Lemma B.4.** *Let the $j$-th row and $k$-th column entry in the Hessian matrix $\nabla^2_{\boldsymbol{\theta}=\boldsymbol{\theta}'}(\ell_s(\boldsymbol{\theta}))$ be denoted as $[\nabla^2_{\boldsymbol{\theta}=\boldsymbol{\theta}'}(\ell_s(\boldsymbol{\theta}))]_{jk}$. Then we have that*

$$[\nabla^2_{\boldsymbol{\theta}=\boldsymbol{\theta}'}(\ell_s(\boldsymbol{\theta}))]_{jk} = 2\frac{\partial f_{I_s}(\boldsymbol{\theta}; x_i, x_j, \mathcal{F}^{s-1})}{\partial\boldsymbol{\theta}_j}\frac{\partial f_{I_s}(\boldsymbol{\theta}; x_i, x_j, \mathcal{F}^{s-1})}{\partial\boldsymbol{\theta}_k} + 2\left(f_{I_s}(\boldsymbol{\theta}; x_i, x_j, \mathcal{F}^{s-1}) - Y_s\right)\frac{\partial^2 f_{I_s}(\boldsymbol{\theta}; x_i, x_j, \mathcal{F}^{s-1})}{\partial\boldsymbol{\theta}_j\partial\boldsymbol{\theta}_k}.$$

*Proof.* This lemma follows from Frostig et al. (2015); Mukherjee et al. (2022) adapted to our setting for the squared loss, and transition function $f_{I_s}(\boldsymbol{\theta}_*; x_i, x_j, \mathcal{F}^{s-1})$. We want to evaluate the Hessian $\nabla^2_{\boldsymbol{\theta}=\boldsymbol{\theta}'}(\ell_s(\boldsymbol{\theta}))$ at any $\boldsymbol{\theta}' \in \Theta$. We denote the $j$-th row and $k$-th column entry in the Hessian matrix as $[\nabla^2_{\boldsymbol{\theta}=\boldsymbol{\theta}'}(\ell_s(\boldsymbol{\theta}))]_{jk}$. Then we can show that

$$[\nabla^2_{\boldsymbol{\theta}=\boldsymbol{\theta}'}(\ell_s(\boldsymbol{\theta}))]_{jk} := \frac{\partial}{\partial\boldsymbol{\theta}_j}\left[\frac{\partial(f_{I_s}(\boldsymbol{\theta}; x_i, x_j, \mathcal{F}^{s-1}) - Y_s)^2}{\partial\boldsymbol{\theta}_k}\right] = \frac{\partial}{\partial\boldsymbol{\theta}_j}\left[2(f_{I_s}(\boldsymbol{\theta}; x_i, x_j, \mathcal{F}^{s-1}) - Y_s)\frac{\partial f_{I_s}(\boldsymbol{\theta}; x_i, x_j, \mathcal{F}^{s-1})}{\partial\boldsymbol{\theta}_k}\right]$$

$$= \frac{\partial}{\partial\boldsymbol{\theta}_j}\left[2f_{I_s}(\boldsymbol{\theta}; x_i, x_j, \mathcal{F}^{s-1})\frac{\partial f_{I_s}(\boldsymbol{\theta}; x_i, x_j, \mathcal{F}^{s-1})}{\partial\boldsymbol{\theta}_k} - 2Y_s\frac{\partial f_{I_s}(\boldsymbol{\theta}; x_i, x_j, \mathcal{F}^{s-1})}{\partial\boldsymbol{\theta}_k}\right]$$

$$= 2\frac{\partial f_{I_s}(\boldsymbol{\theta}; x_i, x_j, \mathcal{F}^{s-1})}{\partial\boldsymbol{\theta}_j}\frac{\partial f_{I_s}(\boldsymbol{\theta}; x_i, x_j, \mathcal{F}^{s-1})}{\partial\boldsymbol{\theta}_k} + 2f_{I_s}(\boldsymbol{\theta}; x_i, x_j, \mathcal{F}^{s-1})\frac{\partial^2 f_{I_s}(\boldsymbol{\theta}; x_i, x_j, \mathcal{F}^{s-1})}{\partial\boldsymbol{\theta}_j\partial\boldsymbol{\theta}_k}$$

$$- 2Y_s\frac{\partial^2 f_{I_s}(\boldsymbol{\theta}; x_i, x_j, \mathcal{F}^{s-1})}{\partial\boldsymbol{\theta}_j\partial\boldsymbol{\theta}_k} - 2\frac{\partial f_{I_s}(\boldsymbol{\theta}; x_i, x_j, \mathcal{F}^{s-1})}{\partial\boldsymbol{\theta}_j}\frac{\partial Y_s}{\partial\boldsymbol{\theta}_k}$$

$$= 2\frac{\partial f_{I_s}(\boldsymbol{\theta}; x_i, x_j, \mathcal{F}^{s-1})}{\partial\boldsymbol{\theta}_j}\frac{\partial f_{I_s}(\boldsymbol{\theta}; x_i, x_j, \mathcal{F}^{s-1})}{\partial\boldsymbol{\theta}_k} + 2\left(f_{I_s}(\boldsymbol{\theta}; x_i, x_j, \mathcal{F}^{s-1}) - Y_s\right)\frac{\partial^2 f_{I_s}(\boldsymbol{\theta}; x_i, x_j, \mathcal{F}^{s-1})}{\partial\boldsymbol{\theta}_j\partial\boldsymbol{\theta}_k}$$

The claim of the lemma follows. $\qquad\square$

**Lemma B.5.** *Let the $j$-th row and $k$-th column entry in the Hessian matrix $\nabla^2_{\boldsymbol{\theta}=\boldsymbol{\theta}'}(\mathbb{E}[\ell_s(\boldsymbol{\theta})|\mathcal{F}^{s-1}])$ be denoted as $[\nabla^2_{\boldsymbol{\theta}=\boldsymbol{\theta}'}(\mathbb{E}[\ell_s(\boldsymbol{\theta})|\mathcal{F}^{s-1}])]_{jk}$. Then we have that*

$$
\left[\nabla^2_{\boldsymbol{\theta}=\boldsymbol{\theta}'}\mathbb{E}[\ell_s(\boldsymbol{\theta})|\mathcal{F}^{s-1}]\right]_{jk} = 2\sum_{i=1}^{|V|} p_{\widehat{\boldsymbol{\theta}}_{s-1}}(i)\left(\frac{\partial f_{I_s}(\boldsymbol{\theta};x_i,x_j,\mathcal{F}^{s-1})}{\partial\boldsymbol{\theta}_j}\frac{\partial f_{I_s}(\boldsymbol{\theta};x_i,x_j,\mathcal{F}^{s-1})}{\partial\boldsymbol{\theta}_k}\right.
$$
$$
\left.+2\left(f_{I_s}(\boldsymbol{\theta};x_i,x_j,\mathcal{F}^{s-1}) - f_{I_s}(\boldsymbol{\theta}_*;x_i,x_j,\mathcal{F}^{s-1})\right)\frac{\partial^2 f_{I_s}(\boldsymbol{\theta};x_i,x_j,\mathcal{F}^{s-1})}{\partial\boldsymbol{\theta}_j\partial\boldsymbol{\theta}_k}\right).
$$

*Proof.* This lemma follows from Frostig et al. (2015); Mukherjee et al. (2022) adapted to our setting for the squared loss, transition function $f_{I_s}(\boldsymbol{\theta}_*;x_i,x_j,\mathcal{F}^{s-1})$, and the sampling distribution $\mathbf{p}_{\widehat{\boldsymbol{\theta}}_{s-1}}$. We show it here for completeness. Now we want to evaluate the Hessian $\nabla^2_{\boldsymbol{\theta}=\boldsymbol{\theta}'}(\mathbb{E}[\ell_s(\boldsymbol{\theta})|\mathcal{F}^{s-1}])$ at any $\boldsymbol{\theta}' \in \boldsymbol{\Theta}$. We denote the $j$-th row and $k$-th column entry in the Hessian matrix as $[\nabla^2_{\boldsymbol{\theta}=\boldsymbol{\theta}'}(\mathbb{E}[\ell_s(\boldsymbol{\theta})|\mathcal{F}^{s-1}])]_{jk}$. Then we can show that

$$
\nabla^2_{\boldsymbol{\theta}=\boldsymbol{\theta}'}\mathbb{E}[\ell_s(\boldsymbol{\theta})|\mathcal{F}^{s-1}] = \nabla^2_{\boldsymbol{\theta}=\boldsymbol{\theta}'}\left(f_{I_s}^2(\boldsymbol{\theta};x_i,x_j,\mathcal{F}^{s-1}) + \mathbb{E}[Y_s^2|\mathcal{F}^{s-1}] - 2\mathbb{E}[Y_s|\mathcal{F}^{s-1}]f_{I_s}(\boldsymbol{\theta};x_i,x_j,\mathcal{F}^{s-1})\right)
$$
$$
= \nabla^2_{\boldsymbol{\theta}=\boldsymbol{\theta}'}\sum_{i=1}^{|V|} p_{\widehat{\boldsymbol{\theta}}_{s-1}}(i)\left(f_i^2(\boldsymbol{\theta};x_i,x_j,\mathcal{F}^{s-1}) + f_i^2(\boldsymbol{\theta}';x_i,x_j,\mathcal{F}^{s-1}) + \frac{1}{2} - 2f_{I_s}(\boldsymbol{\theta}_*;x_i,x_j,\mathcal{F}^{s-1})f_{I_s}(\boldsymbol{\theta};x_i,x_j,\mathcal{F}^{s-1})\right)
$$
$$
= \nabla^2_{\boldsymbol{\theta}=\boldsymbol{\theta}'}\sum_{i=1}^{|V|} p_{\widehat{\boldsymbol{\theta}}_{s-1}}(i)\left(\left(f_{I_s}(\boldsymbol{\theta}_*;x_i,x_j,\mathcal{F}^{s-1}) - f_{I_s}(\boldsymbol{\theta};x_i,x_j,\mathcal{F}^{s-1})\right)^2 + \frac{1}{2}\right)
$$
$$
= \nabla^2_{\boldsymbol{\theta}=\boldsymbol{\theta}'}\sum_{i=1}^{|V|} p_{\widehat{\boldsymbol{\theta}}_{s-1}}(i)\left(\left(f_{I_s}(\boldsymbol{\theta}_*;x_i,x_j,\mathcal{F}^{s-1}) - f_{I_s}(\boldsymbol{\theta};x_i,x_j,\mathcal{F}^{s-1})\right)^2\right) \tag{6}
$$

We now denote the $j$-th row and $k$-th column entry of the Hessian Matrix $\nabla^2_{\boldsymbol{\theta}=\boldsymbol{\theta}'}((f_{I_s}(\boldsymbol{\theta};x_i,x_j,\mathcal{F}^{s-1}) - f_i(\boldsymbol{\theta}_*;x_i,x_j,\mathcal{F}^{s-1}))^2)$ as $\left[\nabla^2_{\boldsymbol{\theta}=\boldsymbol{\theta}'}((f_{I_s}(\boldsymbol{\theta};x_i,x_j,\mathcal{F}^{s-1}) - f_i(\boldsymbol{\theta}_*;x_i,x_j,\mathcal{F}^{s-1}))^2)\right]_{jk}$. Then we can show that

$$
\left[\nabla^2_{\boldsymbol{\theta}=\boldsymbol{\theta}_*}((f_{I_s}(\boldsymbol{\theta};x_i,x_j,\mathcal{F}^{s-1}) - f_{I_s}(\boldsymbol{\theta}_*;x_i,x_j,\mathcal{F}^{s-1}))^2)\right]_{jk}
$$
$$
:= \frac{\partial}{\partial\boldsymbol{\theta}_j}\left[\frac{\partial(f_{I_s}(\boldsymbol{\theta};x_i,x_j,\mathcal{F}^{s-1}) - f_{I_s}(\boldsymbol{\theta}_*;x_i,x_j,\mathcal{F}^{s-1}))^2}{\partial\boldsymbol{\theta}_k}\right]
$$
$$
= \frac{\partial}{\partial\boldsymbol{\theta}_j}\left[2(f_{I_s}(\boldsymbol{\theta};x_i,x_j,\mathcal{F}^{s-1}) - f_{I_s}(\boldsymbol{\theta}_*;x_i,x_j,\mathcal{F}^{s-1}))\frac{\partial f_{I_s}(\boldsymbol{\theta};x_i,x_j,\mathcal{F}^{s-1})}{\partial\boldsymbol{\theta}_k}\right]
$$
$$
= \frac{\partial}{\partial\boldsymbol{\theta}_j}\left[2f_{I_s}(\boldsymbol{\theta};x_i,x_j,\mathcal{F}^{s-1})\frac{\partial f_{I_s}(\boldsymbol{\theta};x_i,x_j,\mathcal{F}^{s-1})}{\partial\boldsymbol{\theta}_k} - 2f_i(\boldsymbol{\theta}_*)\frac{\partial f_{I_s}(\boldsymbol{\theta};x_i,x_j,\mathcal{F}^{s-1})}{\partial\boldsymbol{\theta}_k}\right]
$$
$$
= 2\frac{\partial f_{I_s}(\boldsymbol{\theta};x_i,x_j,\mathcal{F}^{s-1})}{\partial\boldsymbol{\theta}_j}\frac{\partial f_{I_s}(\boldsymbol{\theta};x_i,x_j,\mathcal{F}^{s-1})}{\partial\boldsymbol{\theta}_k} + 2f_{I_s}(\boldsymbol{\theta};x_i,x_j,\mathcal{F}^{s-1})\frac{\partial^2 f_{I_s}(\boldsymbol{\theta};x_i,x_j,\mathcal{F}^{s-1})}{\partial\boldsymbol{\theta}_j\boldsymbol{\theta}_k}
$$
$$
- 2f_{I_s}(\boldsymbol{\theta}_*;x_i,x_j,\mathcal{F}^{s-1})\frac{\partial^2 f_{I_s}(\boldsymbol{\theta};x_i,x_j,\mathcal{F}^{s-1})}{\partial\boldsymbol{\theta}_j\boldsymbol{\theta}_k} - 2\frac{\partial f_{I_s}(\boldsymbol{\theta};x_i,x_j,\mathcal{F}^{s-1})}{\partial\boldsymbol{\theta}_j}\frac{\partial f_{I_s}(\boldsymbol{\theta}_*;x_i,x_j,\mathcal{F}^{s-1})}{\partial\boldsymbol{\theta}_k}
$$
$$
= 2\frac{\partial f_{I_s}(\boldsymbol{\theta};x_i,x_j,\mathcal{F}^{s-1})}{\partial\boldsymbol{\theta}_j}\frac{\partial f_{I_s}(\boldsymbol{\theta};x_i,x_j,\mathcal{F}^{s-1})}{\partial\boldsymbol{\theta}_k} + 2\left(f_{I_s}(\boldsymbol{\theta};x_i,x_j,\mathcal{F}^{s-1}) - f_{I_s}(\boldsymbol{\theta}_*;x_i,x_j,\mathcal{F}^{s-1})\right)\frac{\partial^2 f_{I_s}(\boldsymbol{\theta};x_i,x_j,\mathcal{F}^{s-1})}{\partial\boldsymbol{\theta}_j\partial\boldsymbol{\theta}_k}
$$

Plugging this back in Equation (6) we get that

$$
\left[\nabla^2_{\boldsymbol{\theta}=\boldsymbol{\theta}'}\mathbb{E}[\ell_s(\boldsymbol{\theta})|\mathcal{F}^{s-1}]\right]_{jk} = 2\sum_{i=1}^{|V|} p_{\widehat{\boldsymbol{\theta}}_{s-1}}(i)\left(\frac{\partial f_{I_s}(\boldsymbol{\theta};x_i,x_j,\mathcal{F}^{s-1})}{\partial\boldsymbol{\theta}_j}\frac{\partial f_{I_s}(\boldsymbol{\theta};x_i,x_j,\mathcal{F}^{s-1})}{\partial\boldsymbol{\theta}_k}\right.
$$
$$
\left.+2\left(f_{I_s}(\boldsymbol{\theta};x_i,x_j,\mathcal{F}^{s-1}) - f_{I_s}(\boldsymbol{\theta}_*;x_i,x_j,\mathcal{F}^{s-1})\right)\frac{\partial^2 f_{I_s}(\boldsymbol{\theta};x_i,x_j,\mathcal{F}^{s-1})}{\partial\boldsymbol{\theta}_j\partial\boldsymbol{\theta}_k}\right).
$$

$\square$

**Lemma B.6.** *The sum of the difference of the Hessians $\sum_{s=1}^{t} \nabla_{\boldsymbol{\theta}=\boldsymbol{\theta}'}^2 \ell_s(\boldsymbol{\theta}) - \mathbb{E}\left[\nabla_{\boldsymbol{\theta}=\boldsymbol{\theta}'}^2 \ell_s(\boldsymbol{\theta}) \mid \mathcal{F}^{s-1}\right]$ is given by*

$$
\sum_{s=1}^{t} \nabla_{\boldsymbol{\theta}=\boldsymbol{\theta}'}^2 \ell_s(\boldsymbol{\theta}) - \mathbb{E}\left[\nabla_{\boldsymbol{\theta}=\boldsymbol{\theta}'}^2 \ell_s(\boldsymbol{\theta}) \mid \mathcal{F}^{s-1}\right] = \sum_{s=1}^{t} \bigg( -2(Y_s - f_{I_s}(\boldsymbol{\theta}; x_i, x_j, \mathcal{F}^{s-1})) \frac{\partial^2 f_{I_s}(\boldsymbol{\theta}; x_i, x_j, \mathcal{F}^{s-1})}{\partial \boldsymbol{\theta}_j \partial \boldsymbol{\theta}_k}
$$
$$
+ 2 \frac{\partial f_{I_s}(\boldsymbol{\theta}; x_i, x_j, \mathcal{F}^{s-1})}{\partial \boldsymbol{\theta}_j} \frac{\partial f_{I_s}(\boldsymbol{\theta}; x_i, x_j, \mathcal{F}^{s-1})}{\partial \boldsymbol{\theta}_k}
$$
$$
- 2 \sum_{i=1}^{|V|} p_{\widehat{\boldsymbol{\theta}}_{s-1}}(i) \frac{\partial f_{I_s}(\boldsymbol{\theta}; x_i, x_j, \mathcal{F}^{s-1})}{\partial \boldsymbol{\theta}_j} \frac{\partial f_{I_s}(\boldsymbol{\theta}; x_i, x_j, \mathcal{F}^{s-1})}{\partial \boldsymbol{\theta}_k} \bigg).
$$

*Proof.* This lemma directly follows from Lemma B.4 and Lemma B.5. First note that the difference $\nabla_{\boldsymbol{\theta}=\boldsymbol{\theta}'}^2 \ell_s(\boldsymbol{\theta}) - \mathbb{E}\left[\nabla_{\boldsymbol{\theta}=\boldsymbol{\theta}'}^2 \ell_s(\boldsymbol{\theta}) \mid \mathcal{F}^{s-1}\right]_{jk}$ is given by

$$
\nabla_{\boldsymbol{\theta}=\boldsymbol{\theta}'}^2 \ell_s(\boldsymbol{\theta}) - \mathbb{E}\left[\nabla_{\boldsymbol{\theta}=\boldsymbol{\theta}'}^2 \ell_s(\boldsymbol{\theta}) \mid \mathcal{F}^{s-1}\right]
$$
$$
\overset{(a)}{=} 2 \frac{\partial f_{I_s}(\boldsymbol{\theta}; x_i, x_j, \mathcal{F}^{s-1})}{\partial \boldsymbol{\theta}_j} \frac{\partial f_{I_s}(\boldsymbol{\theta}; x_i, x_j, \mathcal{F}^{s-1})}{\partial \boldsymbol{\theta}_k} + 2 \left( f_{I_s}(\boldsymbol{\theta}; x_i, x_j, \mathcal{F}^{s-1}) - Y_s \right) \frac{\partial^2 f_{I_s}(\boldsymbol{\theta}; x_i, x_j, \mathcal{F}^{s-1})}{\partial \boldsymbol{\theta}_j \partial \boldsymbol{\theta}_k}
$$
$$
- 2 \sum_{i=1}^{|V|} p_{\widehat{\boldsymbol{\theta}}_{s-1}}(i) \bigg( \frac{\partial f_{I_s}(\boldsymbol{\theta}; x_i, x_j, \mathcal{F}^{s-1})}{\partial \boldsymbol{\theta}_j} \frac{\partial f_{I_s}(\boldsymbol{\theta}; x_i, x_j, \mathcal{F}^{s-1})}{\partial \boldsymbol{\theta}_k} - \left( f_{I_s}(\boldsymbol{\theta}; x_i, x_j, \mathcal{F}^{s-1}) - f_{I_s}(\boldsymbol{\theta}_*; x_i, x_j, \mathcal{F}^{s-1}) \right) \cdot
$$
$$
\frac{\partial^2 f_{I_s}(\boldsymbol{\theta}; x_i, x_j, \mathcal{F}^{s-1})}{\partial \boldsymbol{\theta}_j \partial \boldsymbol{\theta}_k} \bigg)
$$
$$
= -2(Y_s - f_{I_s}(\boldsymbol{\theta}; x_i, x_j, \mathcal{F}^{s-1})) \frac{\partial^2 f_{I_s}(\boldsymbol{\theta}; x_i, x_j, \mathcal{F}^{s-1})}{\partial \boldsymbol{\theta}_j \partial \boldsymbol{\theta}_k} + 2 \frac{\partial f_{I_s}(\boldsymbol{\theta}; x_i, x_j, \mathcal{F}^{s-1})}{\partial \boldsymbol{\theta}_j} \frac{\partial f_{I_s}(\boldsymbol{\theta}; x_i, x_j, \mathcal{F}^{s-1})}{\partial \boldsymbol{\theta}_k}
$$
$$
- 2 \sum_{i=1}^{|V|} p_{\widehat{\boldsymbol{\theta}}_{s-1}}(i) \frac{\partial f_{I_s}(\boldsymbol{\theta}; x_i, x_j, \mathcal{F}^{s-1})}{\partial \boldsymbol{\theta}_j} \frac{\partial f_{I_s}(\boldsymbol{\theta}; x_i, x_j, \mathcal{F}^{s-1})}{\partial \boldsymbol{\theta}_k} \tag{7}
$$

where, $(a)$ follows from Lemma B.4 and Lemma B.5. Plugging this equality in Equation (7) below we get

$$
\sum_{s=1}^{t} \nabla_{\boldsymbol{\theta}=\boldsymbol{\theta}'}^2 \ell_s(\boldsymbol{\theta}) - \mathbb{E}\left[\nabla_{\boldsymbol{\theta}=\boldsymbol{\theta}'}^2 \ell_s(\boldsymbol{\theta}) \mid \mathcal{F}^{s-1}\right]
$$
$$
= \sum_{s=1}^{t} \bigg( -2(Y_s - f_{I_s}(\boldsymbol{\theta}; x_i, x_j, \mathcal{F}^{s-1})) \frac{\partial^2 f_{I_s}(\boldsymbol{\theta}; x_i, x_j, \mathcal{F}^{s-1})}{\partial \boldsymbol{\theta}_j \partial \boldsymbol{\theta}_k} + 2 \frac{\partial f_{I_s}(\boldsymbol{\theta}; x_i, x_j, \mathcal{F}^{s-1})}{\partial \boldsymbol{\theta}_j} \frac{\partial f_{I_s}(\boldsymbol{\theta}; x_i, x_j, \mathcal{F}^{s-1})}{\partial \boldsymbol{\theta}_k}
$$
$$
- 2 \sum_{i=1}^{|V|} p_{\widehat{\boldsymbol{\theta}}_{s-1}}(i) \bigg( \frac{\partial f_{I_s}(\boldsymbol{\theta}; x_i, x_j, \mathcal{F}^{s-1})}{\partial \boldsymbol{\theta}_j} \frac{\partial f_{I_s}(\boldsymbol{\theta}; x_i, x_j, \mathcal{F}^{s-1})}{\partial \boldsymbol{\theta}_k} - 2 \left( f_{I_s}(\boldsymbol{\theta}; x_i, x_j, \mathcal{F}^{s-1}) - f_{I_s}(\boldsymbol{\theta}_*; x_i, x_j, \mathcal{F}^{s-1}) \right) \cdot
$$
$$
\frac{\partial^2 f_{I_s}(\boldsymbol{\theta}; x_i, x_j, \mathcal{F}^{s-1})}{\partial \boldsymbol{\theta}_j \partial \boldsymbol{\theta}_k} \bigg) \bigg).
$$

The claim of the lemma follows. $\square$

**Lemma B.7.** *Let $\widehat{\mathcal{L}}_t(\boldsymbol{\theta}_*) = \frac{1}{t} \sum_{s=1}^{t} \ell_s(\boldsymbol{\theta}_*)$ and $\nabla^2 \mathcal{L}_t(\boldsymbol{\theta}_*) = \frac{1}{t} \sum_{s=1}^{t} \nabla^2 \mathbb{E}[\ell_s(\boldsymbol{\theta}_*) | \mathcal{F}^{s-1}]$. Then we can bound the*

$$
\mathbb{P}\left( \lambda_{\max}(\nabla^2 \widehat{\mathcal{L}}_t(\boldsymbol{\theta}_*) - \nabla^2 \mathcal{L}_t(\theta^*)) > \sqrt{\frac{8C|V|^2 \eta^2 \lambda_1^2 c\gamma \log(dt)}{t}} \right) \le \frac{2}{(dt)^\gamma},
$$

*where $c > 0$ is a constant.*

*Proof.* This lemma is different than Frostig et al. (2015); Mukherjee et al. (2022) as it requires a different concentration bound to take into account the squared loss Assumption 5.2 and the vocabulary size. Recall that $\widehat{\mathcal{L}}_t(\boldsymbol{\theta}_*) = \frac{1}{t}\sum_{s=1}^t \ell_s(\boldsymbol{\theta}_*)$ and $\nabla^2 \mathcal{L}_s(\theta^*) = \nabla^2 \mathbb{E}[\ell_s(\boldsymbol{\theta}_*)|\mathcal{F}^{s-1}]$. We define $\nabla^2 \mathcal{L}_t(\boldsymbol{\theta}_*) = \frac{1}{t}\sum_{s=1}^t \nabla^2 \mathbb{E}[\ell_s(\boldsymbol{\theta}_*)|\mathcal{F}^{s-1}]$. Denote, $\mathbf{V}_s = 2\nabla_{\boldsymbol{\theta}=\boldsymbol{\theta}_*} f_{I_s}(\boldsymbol{\theta}; x_i, x_j, \mathcal{F}^{s-1})\nabla_{\boldsymbol{\theta}=\boldsymbol{\theta}_*} f_{I_s}(\boldsymbol{\theta}; x_i, x_j, \mathcal{F}^{s-1})^\top - 2\sum_{i=1}^{|V|} p_{\widehat{\boldsymbol{\theta}}_{s-1}}(i)\nabla_{\boldsymbol{\theta}=\boldsymbol{\theta}_*} f_{I_s}(\boldsymbol{\theta}; x_i, x_j, \mathcal{F}^{s-1})\nabla_{\boldsymbol{\theta}=\boldsymbol{\theta}_*} f_{I_s}(\boldsymbol{\theta}; x_i, x_j, \mathcal{F}^{s-1})^\top$. Then we can show that,

$$\mathbb{P}\left(\lambda_{\max}(\nabla^2 \widehat{\mathcal{L}}_t(\boldsymbol{\theta}_*) - \nabla^2 \mathcal{L}_t(\theta^*)) > \sqrt{\frac{8C^2|V|^4\eta^2\lambda_1^2 c\gamma \log(dt)}{t}}\right)$$

$$= \mathbb{P}\left(\lambda_{\max}\left(\nabla^2_{\boldsymbol{\theta}=\boldsymbol{\theta}_*}\frac{1}{t}\sum_{s=1}^t \ell_s(\boldsymbol{\theta}) - \frac{1}{t}\sum_{s=1}^t \nabla^2_{\boldsymbol{\theta}=\boldsymbol{\theta}_*}\mathbb{E}[\ell_s(\boldsymbol{\theta})|\mathcal{F}^{s-1}]\right) > \sqrt{\frac{8C^2|V|^4\eta^2\lambda_1^2 c\gamma \log(dt)}{t}}\right)$$

$$= \mathbb{P}\left(\lambda_{\max}\left(\nabla^2_{\boldsymbol{\theta}=\boldsymbol{\theta}_*}\frac{1}{t}\sum_{s=1}^t \left(\ell_s(\boldsymbol{\theta}) - \nabla^2_{\boldsymbol{\theta}=\boldsymbol{\theta}_*}\mathbb{E}[\ell_s(\boldsymbol{\theta})|\mathcal{F}^{s-1}]\right)\right) > \sqrt{\frac{8C^2|V|^4\eta^2\lambda_1^2 c\gamma \log(dt)}{t}}\right)$$

$$\overset{(a)}{\leq} \mathbb{P}\left(\lambda_{\max}\left(\frac{C|V|^2}{t}\sum_{s=1}^t \left(Y_s - f_{I_s}(\boldsymbol{\theta}_*; x_i, x_j, \mathcal{F}^{s-1})\right)\nabla^2_{\boldsymbol{\theta}=\boldsymbol{\theta}_*} f_{I_s}(\boldsymbol{\theta}_*; x_i, x_j, \mathcal{F}^{s-1})\right.\right.$$
$$\left.\left.+\frac{C|V|^2}{t}\sum_{s=1}^t \mathbf{V}_s\right) > \sqrt{\frac{8C^2|V|^4\eta^2\lambda_1^2 c\gamma \log(dt)}{t}}\right)$$

$$\leq \mathbb{P}\left(\lambda_{\max}\left(\frac{1}{t}\sum_{s=1}^t -2\left(Y_s - f_{I_s}(\boldsymbol{\theta}_*; x_i, x_j, \mathcal{F}^{s-1})\right)\nabla^2_{\boldsymbol{\theta}=\boldsymbol{\theta}_*} f_{I_s}(\boldsymbol{\theta}_*; x_i, x_j, \mathcal{F}^{s-1})\right) > \frac{1}{2}\sqrt{\frac{8\eta^2\lambda_1^2 c\gamma \log(dt)}{t}}\right)$$
$$+ \mathbb{P}\left(\lambda_{\max}\left(\frac{1}{t}\sum_{s=1}^t \mathbf{V}_s\right) > \frac{1}{2}\sqrt{\frac{8\eta^2\lambda_1^2 c\gamma \log(dt)}{t}}\right)$$

$$\overset{(b)}{\leq} \mathbb{P}\left(\frac{1}{t}\sum_{s=1}^t -2\left(Y_s - f_{I_s}(\boldsymbol{\theta}_*; x_i, x_j, \mathcal{F}^{s-1})\right)\lambda_{\max}\left(\nabla^2_{\boldsymbol{\theta}=\boldsymbol{\theta}_*} f_{I_s}(\boldsymbol{\theta}_*; x_i, x_j, \mathcal{F}^{s-1})\right) > \frac{1}{2}\sqrt{\frac{8\eta^2\lambda_1^2 c\gamma \log(dt)}{t}}\right)$$
$$+ \mathbb{P}\left(\frac{1}{t}\sum_{s=1}^t \lambda_{\max}\left(\mathbf{V}_s\right) > \frac{1}{2}\sqrt{\frac{8\eta^2\lambda_1^2 c\gamma \log(dt)}{t}}\right)$$

$$\overset{(c)}{\leq} 2\exp\left(-\frac{t^2 8\eta^2\lambda_1^2 c\gamma \log(dt)}{4t}\cdot\frac{1}{2tc\eta^2\lambda_1^2}\right) \overset{(d)}{\leq} 2\left(\frac{1}{dt}\right)^\gamma. \tag{8}$$

where, $(a)$ follows from substituting the value of $\nabla^2_{\boldsymbol{\theta}=\boldsymbol{\theta}_*}\ell_s(\boldsymbol{\theta}) - \nabla^2_{\boldsymbol{\theta}=\boldsymbol{\theta}_*}\mathbb{E}[\ell_s(\boldsymbol{\theta})|\mathcal{F}^{s-1}]$ from Lemma B.6, and $(b)$ follows by triangle inequality, $(c)$ follows by using two concentration inequalities stated below, and $(d)$ follows by simplifying the equations.

Denote $Q_s = -2\left(Y_s - f_{I_s}(\boldsymbol{\theta}_*; x_i, x_j, \mathcal{F}^{s-1})\right)\lambda_{\max}\left(\nabla^2_{\boldsymbol{\theta}=\boldsymbol{\theta}_*} f_{I_s}(\boldsymbol{\theta}_*; x_i, x_j, \mathcal{F}^{s-1})\right)$. Also note that

$\lambda_{\max}\left(\nabla^2_{\boldsymbol{\theta}=\boldsymbol{\theta}_*} f_{I_s}(\boldsymbol{\theta}_*; x_i, x_j, \mathcal{F}^{s-1})\right) \leq \lambda_1$ for all time $s$ using Assumption B.1.

$$\mathbb{P}(\sum_{s=1}^{t} -2\left(Y_s - f_{I_s}(\boldsymbol{\theta}_*; x_i, x_j, \mathcal{F}^{s-1})\right) \lambda_{\max}\left(\nabla^2_{\boldsymbol{\theta}=\boldsymbol{\theta}_*} f_{I_s}(\boldsymbol{\theta}_*; x_i, x_j, \mathcal{F}^{s-1})\right) \geq \epsilon) = \mathbb{P}\left(-\sum_{s=1}^{t} Q_s \geq \epsilon\right)$$

$$= \mathbb{P}\left(e^{-\lambda \sum_{s=1}^{t} Q_s} \geq e^{\lambda \epsilon}\right) \overset{(a)}{\leq} e^{-\lambda \epsilon} \mathbb{E}\left[e^{-\lambda \sum_{s=1}^{t} Q_s}\right] = e^{-\lambda \epsilon} \mathbb{E}\left[\mathbb{E}\left[e^{-\lambda \sum_{s=1}^{t} Q_s} | \widehat{\boldsymbol{\theta}}_{t-1}\right]\right]$$

$$\overset{(b)}{=} e^{-\lambda \epsilon} \mathbb{E}\left[\mathbb{E}\left[e^{-\lambda Q_t} | \widehat{\boldsymbol{\theta}}_{t-1}\right] \mathbb{E}\left[e^{-\lambda \sum_{s=1}^{t-1} Q_s} | \widehat{\boldsymbol{\theta}}_{t-1}\right]\right]$$

$$\leq e^{-\lambda \epsilon} \mathbb{E}\left[\exp\left(2\lambda^2 \lambda_1^2 \eta^2\right) \mathbb{E}\left[e^{-\lambda \sum_{s=1}^{t-1} Q_s} | \widehat{\boldsymbol{\theta}}_{t-1}\right]\right]$$

$$= e^{-\lambda \epsilon} e^{2\lambda^2 \eta^2 \lambda_1^2} \mathbb{E}\left[e^{-\lambda \sum_{s=1}^{t-1} Q_s}\right]$$

$$\vdots$$

$$\overset{(c)}{\leq} e^{-\lambda \epsilon} e^{2\lambda^2 t \eta^2 \lambda_1^2} \overset{(d)}{\leq} \exp\left(-\frac{2\epsilon^2}{t\eta^2 \lambda_1^2}\right).$$

where $(a)$ follows by Markov's inequality, $(b)$ follows as $Q_s$ is conditionally independent given $\widehat{\boldsymbol{\theta}}_{s-1}$, $(c)$ follows by unpacking the term for $t$ times and $(d)$ follows by taking $\lambda = \epsilon/4t\lambda_1^2\eta^2$ where $\lambda_1$ is defined in Assumption 5.1. Next we bound the second term of (8) below.

$$\mathbb{P}(\sum_{s=1}^{t} \lambda_{\max}\left(\mathbf{V}_s\right) \geq \epsilon) = \mathbb{P}\left(\lambda \sum_{s=1}^{t} \lambda_{\max}\left(\mathbf{V}_s\right) \geq \lambda \epsilon\right) = \mathbb{P}\left(e^{\lambda \sum_{s=1}^{t} \lambda_{\max}(\mathbf{V}_s)} \geq e^{\lambda \epsilon}\right) \overset{(a)}{\leq} e^{-\lambda \epsilon} \mathbb{E}\left[e^{\lambda \sum_{s=1}^{t} \lambda_{\max}(\mathbf{V}_s)}\right]$$

$$= e^{-\lambda \epsilon} \mathbb{E}\left[\mathbb{E}\left[e^{\lambda \sum_{s=1}^{t} \lambda_{\max}(\mathbf{V}_s)} | \widehat{\boldsymbol{\theta}}_{t-1}\right]\right]$$

$$\overset{(b)}{=} e^{-\lambda \epsilon} \mathbb{E}\left[\mathbb{E}\left[e^{\lambda \lambda_{\max}(\mathbf{V}_t)} | \widehat{\boldsymbol{\theta}}_{t-1}\right] \mathbb{E}\left[e^{\lambda \sum_{s=1}^{t-1} \lambda_{\max}(\mathbf{V}_s)} | \widehat{\boldsymbol{\theta}}_{t-1}\right]\right]$$

$$\overset{(c)}{\leq} e^{-\lambda \epsilon} \mathbb{E}\left[\exp\left(2c\lambda^2 \lambda_1^2 \eta^2\right) \mathbb{E}\left[e^{\lambda \sum_{s=1}^{t-1} \lambda_{\max}(\mathbf{V}_s)} | \widehat{\boldsymbol{\theta}}_{t-1}\right]\right]$$

$$= e^{-\lambda \epsilon} e^{2c\lambda^2 \eta^2 \lambda_1^2} \mathbb{E}\left[e^{\lambda \sum_{s=1}^{t-1} \lambda_{\max}(\mathbf{V}_s)}\right]$$

$$\vdots$$

$$\overset{(d)}{\leq} e^{-\lambda \epsilon} e^{2c\lambda^2 t \eta^2 \lambda_1^2} \overset{(e)}{\leq} \exp\left(-\frac{2\epsilon^2}{tc\eta^2 \lambda_1^2}\right)$$

where $(a)$ follows by Markov's inequality, $(b)$ follows as $\lambda_{\max}(\mathbf{V}_s)$ is conditionally independent given $\widehat{\boldsymbol{\theta}}_{s-1}$. In the inequality $(c)$ using the always valid upper bound of $2\lambda_1$, we have that $\mathbb{E}[\lambda_{\max}(\mathbf{V}_t)] \leq 2\lambda_1$. So the term in inequality $(c)$ will become $e^{-\lambda \epsilon} e^{2\lambda^2 t \eta^2 \lambda_1^t + 4t\lambda\lambda_1}$. Hence, we can upper bound the inequality $(c)$ by a constant $c > 0$ such that we have $\mathbb{E}[e^{\lambda \lambda_{\max}(V_t)} | \widehat{\boldsymbol{\theta}}_{t-1}] \leq e^{2\lambda^2 \lambda_1^2 \eta^2} e^{2\lambda \times 2\lambda_1} = \exp(2\lambda^2 \lambda_1^2 \eta^2 + 4\lambda\lambda_1) \leq \exp(2c\lambda^2 \lambda_1^2 \eta^2)$. The inequality $(d)$ follows by unpacking the term for $t$ times and $(e)$ follows by taking $\lambda = \epsilon/4tc\lambda_1^2\eta^2$ and $\lambda_1$ defined in Assumption 5.1. $\square$

**Lemma B.8.** *Let* $\widehat{\boldsymbol{\theta}}_t - \boldsymbol{\theta}_* = \left(\nabla^2 \widehat{\mathcal{L}}_t(\widetilde{\boldsymbol{\theta}}_t)\right)^{-1} \nabla\widehat{\mathcal{L}}_t(\boldsymbol{\theta}_*)$ *where* $\widetilde{\boldsymbol{\theta}}_t$ *is between* $\widehat{\boldsymbol{\theta}}_t$ *and* $\boldsymbol{\theta}_*$*. Then we can show that*

$$\left\|\widehat{\boldsymbol{\theta}}_t - \boldsymbol{\theta}_*\right\|_{\nabla^2 \mathcal{L}_t(\boldsymbol{\theta}_*)} \leq \left\|\left(\nabla^2 \mathcal{L}_t(\boldsymbol{\theta}_*)\right)^{1/2} \left(\nabla^2 \widehat{\mathcal{L}}_t(\widetilde{\boldsymbol{\theta}}_t)\right)^{-1} \left(\nabla^2 \mathcal{L}_t(\boldsymbol{\theta}_*)\right)^{1/2}\right\| \left\|\nabla\widehat{\mathcal{L}}_t(\boldsymbol{\theta}_*)\right\|_{\left(\nabla^2 \mathcal{L}_t(\boldsymbol{\theta}_*)\right)^{-1}}.$$

*Proof.* We begin with the definition of $\left\|\widehat{\boldsymbol{\theta}}_t - \boldsymbol{\theta}_*\right\|_{\nabla^2 \mathcal{L}_t(\boldsymbol{\theta}_*)}$ as follows:

$$\left\|\widehat{\boldsymbol{\theta}}_t - \boldsymbol{\theta}_*\right\|_{\nabla^2 \mathcal{L}_t(\boldsymbol{\theta}_*)} \overset{(a)}{=} \sqrt{(\widehat{\boldsymbol{\theta}}_t - \boldsymbol{\theta}_*)^T \nabla^2 \mathcal{L}_t(\boldsymbol{\theta}_*)(\widehat{\boldsymbol{\theta}}_t - \boldsymbol{\theta}_*)}$$

$$\overset{(b)}{=} \sqrt{\left(\left(\nabla^2 \widehat{\mathcal{L}}_t(\widetilde{\boldsymbol{\theta}}_t)\right)^{-1} \nabla \widehat{\mathcal{L}}_t(\boldsymbol{\theta}_*)\right)^T \nabla^2 \mathcal{L}_t(\boldsymbol{\theta}_*)\left(\left(\nabla^2 \widehat{\mathcal{L}}_t(\widetilde{\boldsymbol{\theta}}_t)\right)^{-1} \nabla \widehat{\mathcal{L}}_t(\boldsymbol{\theta}_*)\right)}$$

$$\overset{(c)}{\leq} \left\|\nabla^2 \mathcal{L}_t(\boldsymbol{\theta}_*)^{1/2}\left(\nabla^2 \widehat{\mathcal{L}}_t(\widetilde{\boldsymbol{\theta}}_t)\right)^{-1} \nabla^2 \mathcal{L}_t(\boldsymbol{\theta}_*)^{1/2}\right\| \sqrt{\left(\nabla \widehat{\mathcal{L}}_t(\boldsymbol{\theta}_*)^T \left(\nabla^2 \mathcal{L}_t(\boldsymbol{\theta}_*)\right)^{-1} \nabla \widehat{\mathcal{L}}_t(\boldsymbol{\theta}_*)\right)}$$

$$= \left\|\left(\nabla^2 \mathcal{L}_t(\boldsymbol{\theta}_*)\right)^{1/2}\left(\nabla^2 \widehat{\mathcal{L}}_t(\widetilde{\boldsymbol{\theta}}_t)\right)^{-1}\left(\nabla^2 \mathcal{L}_t(\boldsymbol{\theta}_*)\right)^{1/2}\right\| \left\|\nabla \widehat{\mathcal{L}}_t(\boldsymbol{\theta}_*)\right\|_{\left(\nabla^2 \mathcal{L}_t(\boldsymbol{\theta}_*)\right)^{-1}}.$$

where, $(a)$ follows as $\|x\|_M = \sqrt{x^T M x}$, $(b)$ follows as $\|\widehat{\boldsymbol{\theta}}_t - \boldsymbol{\theta}_*\|_{\nabla^2 \mathcal{L}_t(\boldsymbol{\theta}_*)} = \left(\nabla^2 \widehat{\mathcal{L}}_t(\widetilde{\boldsymbol{\theta}})\right)^{-1} \nabla \widehat{\mathcal{L}}_t(\boldsymbol{\theta}_*)$, and $(c)$ follows from Cauchy Schwarz inequality. The claim of the lemma follows. □

*Remark* B.9. The proof of Theorem 1 consists of several steps. In the first step we relate $\nabla^2 \widehat{\mathcal{L}}_t(\boldsymbol{\theta})$ to $\nabla^2 \mathcal{L}_t(\boldsymbol{\theta}_*)$ for any $\boldsymbol{\theta}$ in a ball $\mathcal{B}$ around $\boldsymbol{\theta}_*$. The ball $\mathcal{B}$ is assumed in Assumption B.1 to be a neighborhood where $\nabla^2 \ell_s(\boldsymbol{\theta})$ satisfies a Lipschitz property. Assumption B.1 in Appendix B are standard and have also been made by Frostig et al. (2015); Chaudhuri et al. (2015); Mukherjee et al. (2022). Using Assumption 5.1 and Assumption B.1, we can show that for a large enough sequences of tokens $t$ stated in Theorem 1 we have the following: (1) $\nabla^2 \mathcal{L}_t(\boldsymbol{\theta}_*)$ lies between in the positive semidefinite order by scaled multiples of $\nabla^2 \widehat{\mathcal{L}}_t(\boldsymbol{\theta})$ for any $\boldsymbol{\theta} \in \mathcal{B}$, and (2) the empirical error minimizing $\widehat{\boldsymbol{\theta}}_t$ is in the ball $\mathcal{B}$ with probability $1 - 1/t^\gamma$, which is the good event $\mathcal{E}$. Then we use a Taylor series expansion around $\widehat{\boldsymbol{\theta}}_t$ and using the fact that $\nabla \widehat{\mathcal{L}}_t(\widehat{\boldsymbol{\theta}}(t)) = 0$ along with the relation between $\nabla^2 \widehat{\mathcal{L}}_t(\boldsymbol{\theta})$ and $\nabla^2 \mathcal{L}_t(\boldsymbol{\theta}_*)$, we can obtain an upper bound to $\|\widehat{\boldsymbol{\theta}}(t) - \boldsymbol{\theta}_*\|_{\nabla^2 \mathcal{L}_t(\boldsymbol{\theta}_*)}$ in terms of $\|\nabla \widehat{\mathcal{L}}_t(\boldsymbol{\theta}_*)\|_{(\nabla^2 \mathcal{L}_t(\boldsymbol{\theta}_*))^{-1}}$ that can be shown to be decreasing with $t$. Further, $\|\widehat{\boldsymbol{\theta}}(t) - \boldsymbol{\theta}_*\|_{\nabla^2 \mathcal{L}_t(\boldsymbol{\theta}_*)}$ can also be used to obtain an upper bound to $\mathcal{L}_t(\widehat{\boldsymbol{\theta}}(t)) - \mathcal{L}_t(\boldsymbol{\theta}_*)$ using a Taylor series expansion. Finally we can bound $\mathbb{E}[\mathcal{L}_t(\widehat{\boldsymbol{\theta}}_t) - \mathcal{L}_t(\boldsymbol{\theta}^*)] = \mathbb{E}[(\mathcal{L}_t(\widehat{\boldsymbol{\theta}}_t) - \mathcal{L}_t(\boldsymbol{\theta}^*))I(\mathcal{E})] + \mathbb{E}[(\mathcal{L}_t(\widehat{\boldsymbol{\theta}}_t) - \mathcal{L}_t(\boldsymbol{\theta}^*))I(\mathcal{E}^\complement)]$ where $I(\cdot)$ is the indicator. Since $\mathbb{P}(\mathcal{E}^\complement) \leq 1/t^\gamma$, the second term can be bounded as $\max_{\boldsymbol{\theta} \in \boldsymbol{\Theta}}\left(\mathcal{L}_t(\boldsymbol{\theta}) - \mathcal{L}_t(\boldsymbol{\theta}^*)\right)/t^\gamma$, while the first term simplifies to $(1 + \rho_t)\sigma_t^2/t$.

**Theorem 1.** *(Restatement of main theorem)* Suppose $\ell_1(\boldsymbol{\theta}), \ell_2(\boldsymbol{\theta}), \cdots, \ell_t(\boldsymbol{\theta}) : \mathbb{R}^{|V|} \to \mathbb{R}$ are loss functions from a distribution that satisfies Assumptions 5.1 , 5.2, and B.1. Define $\mathcal{L}_t(\boldsymbol{\theta}) = \frac{1}{t}\sum_{s=1}^t \mathbb{E}_{x_s \sim \mathbf{P}_{\widehat{\boldsymbol{\theta}}_{s-1}}}[\ell_s(\boldsymbol{\theta})|\mathcal{F}^{s-1}]$ where, $\widehat{\boldsymbol{\theta}}_t = \mathrm{argmin}_{\boldsymbol{\theta} \in \boldsymbol{\Theta}}\sum_{s=1}^t \ell_s(\boldsymbol{\theta})$. If $t$ is large enough such that $\frac{\gamma \log(dt)}{t} \leq c' \min\left\{\frac{1}{C_1 C_2 |V|^4}, \frac{\max_{\boldsymbol{\theta} \in \boldsymbol{\Theta}}(\mathcal{L}_t(\boldsymbol{\theta}) - \mathcal{L}_t(\boldsymbol{\theta}_*))}{C_2}\right\}$ then for a constant $\gamma \geq 2$, universal constants $C_1, C_2, c'$, we can show that

$$(1 - \rho_t)\frac{\sigma_t^2}{t} - \frac{C_1^2}{t^{\gamma/2}} \leq \mathbb{E}\left[\mathcal{L}_t(\widehat{\boldsymbol{\theta}}_t) - \mathcal{L}_t(\boldsymbol{\theta}_*)\right] \leq (1 + \rho_t)\frac{\sigma_t^2}{t} + \frac{\max_{\boldsymbol{\theta} \in \boldsymbol{\Theta}}\left(\mathcal{L}_t(\boldsymbol{\theta}) - \mathcal{L}_t(\boldsymbol{\theta}_*)\right)}{t^\gamma},$$

where $\sigma_t^2 := \mathbb{E}\left[\frac{1}{2}\left\|\nabla \widehat{\mathcal{L}}_t(\boldsymbol{\theta}_*)\right\|_{(\nabla^2 \mathcal{L}_t(\boldsymbol{\theta}_*))^{-1}}^2\right]$, and $\rho_t := \left(C_1 C_2 + 2\eta^2 \lambda_1^2\right)\sqrt{\frac{\gamma \log(dt)}{t}}$.

*Proof.* **Step 1:** We first bound the $\left\|\nabla^2 \widehat{\mathcal{L}}_t(\boldsymbol{\theta}) - \nabla^2 \mathcal{L}_t(\boldsymbol{\theta}_*)\right\|_*$ as follows

$$\left\|\nabla^2 \widehat{\mathcal{L}}_t(\boldsymbol{\theta}) - \nabla^2 \mathcal{L}_t(\boldsymbol{\theta}_*)\right\|_* \overset{(a)}{\leq} \left\|\nabla^2 \widehat{\mathcal{L}}_t(\boldsymbol{\theta}) - \nabla^2 \widehat{\mathcal{L}}_t(\boldsymbol{\theta}_*)\right\|_* + \left\|\nabla^2 \widehat{\mathcal{L}}_t(\boldsymbol{\theta}_*) - \nabla^2 \mathcal{L}_t(\boldsymbol{\theta}_*)\right\|_*$$

$$\overset{(b)}{\leq} C_1 \|\boldsymbol{\theta} - \boldsymbol{\theta}_*\|_{\nabla^2 \mathcal{L}_t(\boldsymbol{\theta}_*)} + \sqrt{\frac{8C^2 |V|^4 \eta^2 \lambda_1^2 c\gamma \log(dt)}{t}} \tag{9}$$

where, $(a)$ follows from triangle inequality, and $(b)$ is due to Assumption B.1.3.d and Lemma B.7.

**Step 2 (Approximation of $\nabla^2 \mathcal{L}_t(\boldsymbol{\theta}_*)$):** By choosing a sufficiently smaller ball $\mathcal{B}_1$ of radius of $\min\left\{1/(10C_1), \text{ diameter}(\mathcal{B})\right\}$ ), the first term in (9) can be made small for $\boldsymbol{\theta} \in \mathcal{B}_1$. Also, for sufficiently large $t$, the second term in (9) can be made

arbitrarily small (smaller than $1/10$), which occurs if $\sqrt{\frac{\gamma \log(dt)}{t}} \leq \frac{c'}{\sqrt{2C^2|V|^4\eta^2\lambda_1^2}}$. Hence for large $t$ and $\boldsymbol{\theta} \in \mathcal{B}_1$ we have

$$\frac{1}{2}\nabla^2\widehat{\mathcal{L}}_t(\boldsymbol{\theta}) \preceq \nabla^2\mathcal{L}_t(\boldsymbol{\theta}_*) \preceq 2\nabla^2\widehat{\mathcal{L}}_t(\boldsymbol{\theta}) \tag{10}$$

**Step 3 (Show $\widehat{\boldsymbol{\theta}}_t$ in $\mathcal{B}_1$):** Fix a $\widetilde{\boldsymbol{\theta}}$ between $\boldsymbol{\theta}$ and $\boldsymbol{\theta}_*$ in $\mathcal{B}_1$. Apply Taylor's series approximation

$$\widehat{\mathcal{L}}_t(\boldsymbol{\theta}) = \widehat{\mathcal{L}}_t(\boldsymbol{\theta}_*) + \nabla\widehat{\mathcal{L}}_t(\boldsymbol{\theta}_*)^\top (\boldsymbol{\theta} - \boldsymbol{\theta}_*) + \frac{1}{2}(\boldsymbol{\theta} - \boldsymbol{\theta}_*)^\top \nabla^2\widehat{\mathcal{L}}_t(\widetilde{\boldsymbol{\theta}})(\boldsymbol{\theta} - \boldsymbol{\theta}_*)$$

We can further reduce this as follows:

$$\begin{aligned}
\widehat{\mathcal{L}}_t(\boldsymbol{\theta}) - \widehat{\mathcal{L}}_t(\boldsymbol{\theta}_*) &\overset{(a)}{=} \nabla\widehat{\mathcal{L}}_t(\boldsymbol{\theta}_*)^\top (\boldsymbol{\theta} - \boldsymbol{\theta}_*) + \frac{1}{2}\|\boldsymbol{\theta} - \boldsymbol{\theta}_*\|_{\nabla^2\widehat{\mathcal{L}}_t(\widetilde{\boldsymbol{\theta}})}^2 \\
&\overset{(b)}{\geq} \nabla\widehat{\mathcal{L}}_t(\boldsymbol{\theta}_*)^\top (\boldsymbol{\theta} - \boldsymbol{\theta}_*) + \frac{1}{4}\|\boldsymbol{\theta} - \boldsymbol{\theta}_*\|_{\nabla^2\mathcal{L}_t(\boldsymbol{\theta}_*)}^2 \\
&\geq -\|\boldsymbol{\theta} - \boldsymbol{\theta}_*\|_{\nabla^2\mathcal{L}_t(\boldsymbol{\theta}_*)}\left\|\nabla\widehat{\mathcal{L}}_t(\boldsymbol{\theta}_*)\right\|_{(\nabla^2\mathcal{L}_t(\boldsymbol{\theta}_*))^{-1}} + \frac{1}{4}\left(\|\boldsymbol{\theta} - \boldsymbol{\theta}_*\|_{\nabla^2\mathcal{L}_t(\boldsymbol{\theta}_*)}\right)^\top\left(\|\boldsymbol{\theta} - \boldsymbol{\theta}_*\|_{\nabla^2\mathcal{L}_t(\boldsymbol{\theta}_*)}\right) \\
&= \|\boldsymbol{\theta} - \boldsymbol{\theta}_*\|_{\nabla^2\mathcal{L}_t(\boldsymbol{\theta}_*)}\left(-\left\|\nabla\widehat{\mathcal{L}}_t(\boldsymbol{\theta}_*)\right\|_{(\nabla^2\mathcal{L}_t(\boldsymbol{\theta}_*))^{-1}} + \frac{1}{4}\|\boldsymbol{\theta} - \boldsymbol{\theta}_*\|_{\nabla^2\mathcal{L}_t(\boldsymbol{\theta}_*)}\right) \tag{11}
\end{aligned}$$

where, $(a)$ follows as $\|\boldsymbol{\theta} - \boldsymbol{\theta}_*\|_{\nabla^2\widehat{\mathcal{L}}_t(\widetilde{\boldsymbol{\theta}})}^2 := (\boldsymbol{\theta} - \boldsymbol{\theta}_*)^\top \nabla^2\widehat{\mathcal{L}}_t(\widetilde{\boldsymbol{\theta}})(\boldsymbol{\theta} - \boldsymbol{\theta}_*)$, and $(b)$ follows as $\widetilde{\boldsymbol{\theta}}$ is in between $\boldsymbol{\theta}$ and $\boldsymbol{\theta}_*$ and then using (10). Note that in (11) if the right hand side is positive for some $\boldsymbol{\theta} \in \mathcal{B}_1$, then $\boldsymbol{\theta}$ is not a local minimum. Also, since $\left\|\nabla\widehat{\mathcal{L}}_t(\boldsymbol{\theta}_*)\right\| \to 0$, for a sufficiently small value of $\left\|\nabla\widehat{\mathcal{L}}_t(\boldsymbol{\theta}_*)\right\|$, all points on the boundary of $\mathcal{B}_1$ will have values greater than that of $\boldsymbol{\theta}_*$. Hence, we must have a local minimum of $\widehat{\mathcal{L}}_t(\boldsymbol{\theta})$ that is strictly inside $\mathcal{B}_1$ (for $t$ large enough). We can ensure this local minimum condition is achieved by choosing an $t$ large enough so that $\sqrt{\frac{\gamma \log(dt)}{t}} \leq c' \min\left\{\frac{1}{C_1C_2}, \frac{\text{diameter}(\mathcal{B})}{C_2}\right\}$, using Lemma B.3 (and our bound on the diameter of $\mathcal{B}_1$). By convexity, we have that this is the global minimum, $\widehat{\boldsymbol{\theta}}_t$, and so $\widehat{\boldsymbol{\theta}}_t \in \mathcal{B}_1$ for $t$ large enough. We will assume now that $t$ is this large from here on.

**Step 4 (Bound $\left\|\widehat{\boldsymbol{\theta}}_t - \boldsymbol{\theta}_*\right\|_{\nabla^2\mathcal{L}_t(\boldsymbol{\theta}_*)}$):** For the $\widehat{\boldsymbol{\theta}}(t)$ that minimizes the sum of squared errors, $0 = \nabla\widehat{\mathcal{L}}_t(\widehat{\boldsymbol{\theta}}_t)$. Again, using Taylor's theorem if $\widehat{\boldsymbol{\theta}}_t$ is an interior point, we have:

$$0 = \nabla\widehat{\mathcal{L}}_t(\widehat{\boldsymbol{\theta}}_t) = \nabla\widehat{\mathcal{L}}_t(\boldsymbol{\theta}_*) + \nabla^2\widehat{\mathcal{L}}_t(\widetilde{\boldsymbol{\theta}}_t)\left(\widehat{\boldsymbol{\theta}}_t - \boldsymbol{\theta}_*\right) \tag{12}$$

for some $\widetilde{\boldsymbol{\theta}}_t$ between $\boldsymbol{\theta}_*$ and $\widehat{\boldsymbol{\theta}}_t$. Now observe that $\widetilde{\boldsymbol{\theta}}_t$ is in $B_1$ (since, for $t$ large enough, $\widehat{\boldsymbol{\theta}}_t \in \mathcal{B}_1$). Thus it follows from (12) that,

$$\widehat{\boldsymbol{\theta}}_t - \boldsymbol{\theta}_* = \left(\nabla^2\widehat{\mathcal{L}}_t(\widetilde{\boldsymbol{\theta}}_t)\right)^{-1}\nabla\widehat{\mathcal{L}}_t(\boldsymbol{\theta}_*) \tag{13}$$

where the invertibility is guaranteed by (10) and the positive definiteness of $\nabla^2\mathcal{L}_t(\boldsymbol{\theta}_*)$ (by Assumption B.1 (3c)). We finally derive the upper bound to $\left\|\widehat{\boldsymbol{\theta}}_t - \boldsymbol{\theta}_*\right\|_{\nabla^2\mathcal{L}_t(\boldsymbol{\theta}_*)}$ as follows

$$\begin{aligned}
\left\|\widehat{\boldsymbol{\theta}}_t - \boldsymbol{\theta}_*\right\|_{\nabla^2\mathcal{L}_t(\boldsymbol{\theta}_*)} &\overset{(a)}{\leq} \left\|\left(\nabla^2\mathcal{L}_t(\boldsymbol{\theta}_*)\right)^{1/2}\left(\nabla^2\widehat{\mathcal{L}}_t(\widetilde{\boldsymbol{\theta}}_t)\right)^{-1}\left(\nabla^2\mathcal{L}_t(\boldsymbol{\theta}_*)\right)^{1/2}\right\|\left\|\nabla\widehat{\mathcal{L}}_t(\boldsymbol{\theta}_*)\right\|_{(\nabla^2\mathcal{L}_t(\boldsymbol{\theta}_*))^{-1}} \\
&\overset{(b)}{\leq} cC_2\sqrt{\frac{\gamma \log(dt)}{t}} \tag{14}
\end{aligned}$$

where $(a)$ follows from Lemma B.8, and $(b)$ from Lemma B.3, (11), and $c$ is some universal constant.

**Step 5 (Introducing $\widetilde{\mathbf{z}}$):** Fix a $\widetilde{\mathbf{z}}_t$ between $\boldsymbol{\theta}_*$ and $\widehat{\boldsymbol{\theta}}_t$. Apply Taylor's series

$$\mathcal{L}_t(\widehat{\boldsymbol{\theta}}_t) - \mathcal{L}_t(\boldsymbol{\theta}_*) = \frac{1}{2}\left(\widehat{\boldsymbol{\theta}}_t - \boldsymbol{\theta}_*\right)^\top \nabla^2\mathcal{L}_t(\widetilde{\mathbf{z}}_t)\left(\widehat{\boldsymbol{\theta}}_t - \boldsymbol{\theta}_*\right) \tag{15}$$

Now note that both $\widetilde{\boldsymbol{\theta}}_t$ and $\widetilde{\mathbf{z}}_t$ are between $\widehat{\boldsymbol{\theta}}_t$ and $\boldsymbol{\theta}_*$, which implies $\widetilde{\boldsymbol{\theta}}_t \to \boldsymbol{\theta}_*$ and $\widetilde{\mathbf{z}}_t \to \boldsymbol{\theta}_*$ since $\widehat{\boldsymbol{\theta}}_t \to \boldsymbol{\theta}_*$. By (9) and (14) and applying the concentration inequalities give us

$$\left\| \nabla^2 \widehat{\mathcal{L}}_t(\widetilde{\boldsymbol{\theta}}_t) - \nabla^2 \mathcal{L}_t(\boldsymbol{\theta}_*) \right\|_* \le \rho_t \tag{16}$$

$$\left\| \nabla^2 \mathcal{L}_t(\widetilde{\mathbf{z}}_t) - \nabla^2 \mathcal{L}_t(\boldsymbol{\theta}_*) \right\|_* \le C_1 \left\| \widetilde{\mathbf{z}}_t - \boldsymbol{\theta}_* \right\|_{\nabla^2 \mathcal{L}_t(\boldsymbol{\theta}_*)} \le \rho_t \tag{17}$$

where $\rho_t = c \left( C_1 C_2 + 2\eta^2 \lambda_1^2 \right) \sqrt{\frac{\gamma \log(dt)}{t}}$.

**Step 6 (Define $\mathbf{M}_{1,t}$ and $\mathbf{M}_{2,t}$):** It follows from the inequality (16) that

$$\nabla^2 \widehat{\mathcal{L}}_t(\widetilde{\boldsymbol{\theta}}_t) \preceq (1 + \rho_t) \nabla^2 \mathcal{L}_t(\boldsymbol{\theta}_*) \implies \nabla^2 \widehat{\mathcal{L}}_t(\widetilde{\boldsymbol{\theta}}_t) - \nabla^2 \mathcal{L}_t(\boldsymbol{\theta}_*) \preceq \rho_t \nabla^2 \mathcal{L}_t(\boldsymbol{\theta}_*)$$
$$\implies \nabla^2 \mathcal{L}_t(\boldsymbol{\theta}_*)^{-1/2} (\widehat{\mathcal{L}}_t(\widetilde{\boldsymbol{\theta}}_t) - \nabla^2 \mathcal{L}_t(\boldsymbol{\theta}_*)) \nabla^2 \mathcal{L}_t(\boldsymbol{\theta}_*)^{-1/2} \preceq \rho_t I$$
$$\implies \| \nabla^2 \widehat{\mathcal{L}}_t(\widetilde{\boldsymbol{\theta}}_t) - \nabla^2 \mathcal{L}_t(\boldsymbol{\theta}_*) \|_* \le \rho_t.$$

Then we can use the inequalities (16) and (17) to show that

$$(1 - \rho_t) \nabla^2 \mathcal{L}_t(\boldsymbol{\theta}_*) \preceq \nabla^2 \widehat{\mathcal{L}}_t(\widetilde{\boldsymbol{\theta}}_t) \preceq (1 + \rho_t) \nabla^2 \mathcal{L}_t(\boldsymbol{\theta}_*)$$
$$(1 - \rho_t) \nabla^2 \mathcal{L}_t(\boldsymbol{\theta}_*) \preceq \nabla^2 \mathcal{L}_t(\widetilde{\mathbf{z}}_t) \preceq (1 + \rho_t) \nabla^2 \mathcal{L}_t(\boldsymbol{\theta}_*).$$

Now we define the two quantities $\mathbf{M}_{1,t}$ and $\mathbf{M}_{2,t}$ as follows:

$$\mathbf{M}_{1,t} := \left( \nabla^2 \mathcal{L}_t(\boldsymbol{\theta}_*) \right)^{1/2} \left( \nabla^2 \widehat{\mathcal{L}}_t(\widetilde{\boldsymbol{\theta}}_t) \right)^{-1} \left( \nabla^2 \mathcal{L}_t(\boldsymbol{\theta}_*) \right)^{1/2}$$

$$\mathbf{M}_{2,t} := \left( \nabla^2 \mathcal{L}_t(\boldsymbol{\theta}_*) \right)^{-1/2} \nabla^2 \mathcal{L}_t(\widetilde{\mathbf{z}}_t) \left( \nabla^2 \mathcal{L}_t(\boldsymbol{\theta}_*) \right)^{-1/2}.$$

**Step 7 (Lower bound $\mathcal{L}_t(\widehat{\boldsymbol{\theta}}_t) - \mathcal{L}_t(\boldsymbol{\theta}_*)$):** Now for the lower bound it follows from Equation (15) that

$$\mathcal{L}_t(\widehat{\boldsymbol{\theta}}_t) - \mathcal{L}_t(\boldsymbol{\theta}_*) = \frac{1}{2} \left( \widehat{\boldsymbol{\theta}}_t - \boldsymbol{\theta}_* \right)^\top \nabla^2 \mathcal{L}_t(\widetilde{\mathbf{z}}_t) \left( \widehat{\boldsymbol{\theta}}_t - \boldsymbol{\theta}_* \right)$$
$$= \frac{1}{2} \left( \widehat{\boldsymbol{\theta}}_t - \boldsymbol{\theta}_* \right)^\top \nabla^2 \mathcal{L}_t(\boldsymbol{\theta}_*)^{\frac{1}{2}} \nabla^2 \mathcal{L}_t(\boldsymbol{\theta}_*)^{-\frac{1}{2}} \nabla^2 \mathcal{L}_t(\widetilde{\mathbf{z}}_t) \nabla^2 \mathcal{L}_t(\boldsymbol{\theta}_*)^{-\frac{1}{2}} \nabla^2 \mathcal{L}_t(\boldsymbol{\theta}_*)^{\frac{1}{2}} \left( \widehat{\boldsymbol{\theta}}_t - \boldsymbol{\theta}_* \right)$$
$$\overset{(a)}{=} \frac{1}{2} \mathbf{u}^T \mathbf{M}_{2,t} \mathbf{u}$$

where, in $(a)$ we define the vector $\mathbf{u} := \left( \widehat{\boldsymbol{\theta}}_t - \boldsymbol{\theta}_* \right)^\top \nabla^2 \mathcal{L}_t(\boldsymbol{\theta}_*)^{\frac{1}{2}}$. Now observe from the definition of and then using the min-max theorem we can show that

$$\mathcal{L}_t(\widehat{\boldsymbol{\theta}}_t) - \mathcal{L}_t(\boldsymbol{\theta}_*) \ge \frac{1}{2} \lambda_{\min}(\mathbf{M}_{2,t}) \mathbf{u}^T \mathbf{u}$$
$$= \frac{1}{2} \lambda_{\min}(\mathbf{M}_{2,t}) \left\| \widehat{\boldsymbol{\theta}}_t - \boldsymbol{\theta}_* \right\|_{\nabla^2 \mathcal{L}_t(\boldsymbol{\theta}_*)}^2$$
$$= \frac{1}{2} \lambda_{\min}(\mathbf{M}_{2,t}) \left\| \nabla^2 \widehat{\mathcal{L}}_t(\widetilde{\boldsymbol{\theta}}_t) \left( \widehat{\boldsymbol{\theta}}_t - \boldsymbol{\theta}_* \right) \right\|_{\left( \nabla^2 \widehat{\mathcal{L}}_t(\widetilde{\boldsymbol{\theta}}_t) \right)^{-1} \nabla^2 \mathcal{L}_t(\boldsymbol{\theta}_*) \left( \nabla^2 \widehat{\mathcal{L}}_t(\widetilde{\boldsymbol{\theta}}_t) \right)^{-1}}^2$$
$$\ge \frac{1}{2} \left( \lambda_{\min}(\mathbf{M}_{1,t}) \right)^2 \lambda_{\min}(\mathbf{M}_{2,t}) \left\| \nabla^2 \widehat{\mathcal{L}}_t(\widetilde{\boldsymbol{\theta}}_t) \left( \widehat{\boldsymbol{\theta}}_t - \boldsymbol{\theta}_* \right) \right\|_{\left( \nabla^2 \mathcal{L}_t(\boldsymbol{\theta}_*) \right)^{-1}}^2$$
$$\overset{(a)}{=} \frac{1}{2} \left( \lambda_{\min}(\mathbf{M}_{1,t}) \right)^2 \lambda_{\min}(\mathbf{M}_{2,t}) \left\| \nabla \widehat{\mathcal{L}}_t(\boldsymbol{\theta}_*) \right\|_{\left( \nabla^2 \mathcal{L}_t(\boldsymbol{\theta}_*) \right)^{-1}}^2$$

where, in $(a)$ we use the Equation (13).

**Step 8:** Define $I(\mathcal{E})$ as the indicator that the desired previous events hold, which we can ensure with probability greater than $1 - 2\left(\frac{1}{dt}\right)^{\gamma}$. Then we can show that:

$$
\begin{aligned}
\mathbb{E}\left[\mathcal{L}_t(\widehat{\boldsymbol{\theta}}_t) - \mathcal{L}_t(\boldsymbol{\theta}_*)\right] \geq & \mathbb{E}\left[\left(\mathcal{L}_t(\widehat{\boldsymbol{\theta}}_t) - \mathcal{L}_t(\boldsymbol{\theta}_*)\right) I(\mathcal{E})\right] \\
\geq & \frac{1}{2}\mathbb{E}\left[\left(\lambda_{\min}\left(\mathbf{M}_{1,t}\right)\right)^2 \lambda_{\min}\left(\mathbf{M}_{2,t}\right) \left\|\nabla\widehat{\mathcal{L}}_t\left(\boldsymbol{\theta}_*\right)\right\|^2_{\left(\nabla^2 \mathcal{L}_t(\boldsymbol{\theta}_*)\right)^{-1}} I(\mathcal{E})\right] \\
\geq & \left(1 - c'\rho_t\right) \frac{1}{2}\mathbb{E}\left[\left\|\nabla\widehat{\mathcal{L}}_t\left(\boldsymbol{\theta}_*\right)\right\|^2_{\left(\nabla^2 \mathcal{L}_t(\boldsymbol{\theta}_*)\right)^{-1}} I(\mathcal{E})\right] \\
= & \left(1 - c'\rho_t\right) \frac{1}{2}\mathbb{E}\left[\left\|\nabla\widehat{\mathcal{L}}_t\left(\boldsymbol{\theta}_*\right)\right\|^2_{\left(\nabla^2 \mathcal{L}_t(\boldsymbol{\theta}_*)\right)^{-1}} (1 - I(\text{not } \mathcal{E}))\right] \\
\stackrel{(a)}{=} & \left(1 - c'\rho_t\right) \left(\sigma_t^2 - \frac{1}{2}\mathbb{E}\left[\left\|\nabla\widehat{\mathcal{L}}_t\left(\boldsymbol{\theta}_*\right)\right\|^2_{\left(\nabla^2 \mathcal{L}_t(\boldsymbol{\theta}_*)\right)^{-1}} I(\text{not } \mathcal{E})\right]\right) \\
\geq & \left(1 - c'\rho_t\right) \sigma_t^2 - \mathbb{E}\left[\left\|\nabla\widehat{\mathcal{L}}_t\left(\boldsymbol{\theta}_*\right)\right\|^2_{\left(\nabla^2 \mathcal{L}_t(\boldsymbol{\theta}_*)\right)^{-1}} I(\text{not } \mathcal{E})\right]
\end{aligned}
$$

where, in $(a)$ we have $\sigma_t^2 := \left\|\nabla\widehat{\mathcal{L}}_t\left(\boldsymbol{\theta}_*\right)\right\|^2_{\left(\nabla^2 \mathcal{L}_t(\boldsymbol{\theta}_*)\right)^{-1}}$, and $c'$ is an universal constant.

**Step 9:** Define the random variable $Z = \left\|\nabla\widehat{\mathcal{L}}_t\left(\boldsymbol{\theta}_*\right)\right\|_{\left(\nabla^2 \mathcal{L}_t(\boldsymbol{\theta}_*)\right)^{-1}}$. With a failure event probability of less than $2\left(\frac{1}{dt}\right)^{\gamma}$ for any $z_0$, we have:

$$
\begin{aligned}
\mathbb{E}\left[Z^2 I(\text{not } \mathcal{E})\right] = & \mathbb{E}\left[Z^2 I(\text{not } \mathcal{E}) I\left(Z^2 < z_0\right)\right] + \mathbb{E}\left[Z^2 I(\text{not } \mathcal{E}) I\left(Z^2 \geq z_0\right)\right] \\
\leq & z_0 \mathbb{E}[I(\text{not } \mathcal{E})] + \mathbb{E}\left[Z^2 I\left(Z^2 \geq z_0\right)\right] \\
\leq & \frac{z_0}{2t^{\gamma}} + \mathbb{E}\left[Z^2 \frac{Z^2}{z_0}\right] \\
\leq & \frac{z_0}{2t^{\gamma}} + \frac{\mathbb{E}\left[Z^4\right]}{z_0} \\
\leq & \frac{\sqrt{\mathbb{E}\left[Z^4\right]}}{t^{\gamma/2}}
\end{aligned}
$$

where $z_0 = t^{\gamma/2}\sqrt{\mathbb{E}\left[Z^4\right]}$.

**Step 10 (Upper Bound):** For an upper bound we have that:

$$
\begin{aligned}
\mathbb{E}\left[\mathcal{L}_t(\widehat{\boldsymbol{\theta}}_t) - \mathcal{L}_t(\boldsymbol{\theta}_*)\right] = & \mathbb{E}\left[\left(\mathcal{L}_t(\widehat{\boldsymbol{\theta}}_t) - \mathcal{L}_t(\boldsymbol{\theta}_*)\right) I(\mathcal{E})\right] + \mathbb{E}\left[\left(\mathcal{L}_t(\widehat{\boldsymbol{\theta}}_t) - \mathcal{L}_t(\boldsymbol{\theta}_*)\right) I(\text{not } \mathcal{E})\right] \\
\leq & \mathbb{E}\left[\left(\mathcal{L}_t(\widehat{\boldsymbol{\theta}}_t) - \mathcal{L}_t(\boldsymbol{\theta}_*)\right) I(\mathcal{E})\right] + \frac{\max_{\boldsymbol{\theta}\in\Theta}\left(\mathcal{L}_t(\boldsymbol{\theta}) - \mathcal{L}_t(\boldsymbol{\theta}_*)\right)}{t^{\gamma}}
\end{aligned}
$$

since the probability of not $\mathcal{E}$ is less than $\dfrac{1}{t^{\gamma}}$. Now for an upper bound of the first term, observe that

$$
\begin{aligned}
\mathbb{E}\left[\left(\mathcal{L}_t(\widehat{\boldsymbol{\theta}}_t) - \mathcal{L}_t(\boldsymbol{\theta}_*)\right) I(\mathcal{E})\right] \leq & \frac{1}{2}\mathbb{E}\left[\left(\lambda_{\max}\left(\mathbf{M}_{1,t}\right)\right)^2 \lambda_{\max}\left(\mathbf{M}_{2,t}\right)\left\|\nabla\widehat{\mathcal{L}}_t(\boldsymbol{\theta}_*)\right\|^2_{\left(\nabla^2\mathcal{L}_t(\boldsymbol{\theta}_*)\right)^{-1}} I(\mathcal{E})\right] \\
\leq & \left(1 + c'\rho_t\right)\frac{1}{2}\mathbb{E}\left[\left\|\nabla\widehat{\mathcal{L}}_t(\boldsymbol{\theta}_*)\right\|^2_{\left(\nabla^2\mathcal{L}_t(\boldsymbol{\theta}_*)\right)^{-1}} I(\mathcal{E})\right] \\
\leq & \left(1 + c'\rho_t\right)\frac{1}{2}\mathbb{E}\left[\left\|\nabla\widehat{\mathcal{L}}_t(\boldsymbol{\theta}_*)\right\|^2_{\left(\nabla^2\mathcal{L}_t(\boldsymbol{\theta}_*)\right)^{-1}}\right] \\
= & \left(1 + c'\rho_t\right)\frac{\sigma_t^2}{t}
\end{aligned}
$$

where, $c'$ is another universal constant. $\qquad\square$

## C. Experimental Details

### C.1. Dataset Statistics

We provide the processed data statistics in Table 6. We highlight that due to the black-box assumption of the base model, the training set is used for ablation and qualitative analysis in Section 7.3 and Section 7.4.

Table 6. Processed Dataset Statistics. Training set is only used for ablation and qualitative analysis due to the black-box model assumption.

| Dataset | Train | Validation | Test |
|---|---|---|---|
| E2E NLG | 33,525 | 4,299 | 4,693 |
| Web NLG | 2,732 (filtered by categories) | 844 | 720 |
| CommonGen | 1,476 (filtered for "man") | 2,026 | 1,992 |
| Adidas | — | 745 | 100 |

### C.2. Prompts

We now describe the prompts we used for the four datasets and three models.

**E2E NLG Dataset**

- For the **GPT2-M** model, we use the prompt:

  ```
  Given the following aspects of a restaurant, [attributes], a natural
  language sentence describing the restaurant is:
  ```

- For the **GPT2-XL** model, the prompt is:

  ```
  Imagine you are writing a one-sentence description for a restaurant, given
  the following aspects:  [attributes], a human-readable natural language
  sentence describing the restaurant is:
  ```

- For the **LLaMA-3.1-8B** model, we use:

  ```
  Please convert the following attributes into a coherent sentence.  Do not
  provide an explanation.
  ```

**Web NLG Dataset**

- For the **GPT2-M** model, we use the prompt:

```
Convert the following facts into a coherent sentence: Facts: [facts]
Sentence:
```

- For the **GPT2-XL** model, the prompt is:

```
You are given the following facts. Facts: [facts] A short, coherent
sentence summarizing the facts is:
```

- For the **LLaMA-3.1-8B** model, we use:

```
Do not provide an explanation or follow-up. Just convert the following
facts of an entity into a coherent sentence. Facts: [facts] Sentence:
```

**CommonGen Dataset**

- For the **GPT2-M** and **GPT2-XL** models, we use the same prompt:

```
One coherent sentence that uses all the following concepts: [concepts],
is:
```

- For the **LLaMA-3.1-8B** model, we use:

```
Please write a coherent sentence that uses all the following concepts.
Concepts: [concepts] Sentence:
```

**Adidas Dataset**

- For the **GPT2-M** and **GPT2-XL** models, we use the same prompt:

```
Given the following attributes of a product, write a description.
Attributes: [attributes] Description:
```

- For the **LLaMA-3.1-8B** model, we use:

```
Please write a description of this product given the following attributes.
Attributes: [attributes] Description:
```

For **in-context learning**, we simply add a sentence at the beginning of the prompt before adding the samples in the prompt:
`Below are a list of demonstrations:`.

For the qualitative analysis on the distribution shift in Section 7.4, we ask GPT-4o with the following prompt:
For Web NLG dataset:

```
Focus on all the samples, how much percentage is related to ``Person''?
```

For CommonGen dataset:

```
Focus on those samples whose target is related to gender, how much percentage
is related to ``woman''?
```

## C.3. Metrics

We report performance using seven standard metrics often used in the natural language generation tasks. These are: (a) BLEU (Papineni et al., 2002) (measures n-gram overlap between the generated and reference texts, emphasizing precision), (b) ROUGE-1 (Lin, 2004) (computes unigram recall to measure the overlap between generated and reference texts), (c) ROUGE-2 (Lin, 2004) (extends ROUGE-1 to bigrams, measuring the recall of two-word sequences), (d) ROUGE-L (Lin & Och, 2004) (uses the longest common subsequence to evaluate recall), (e) METEOR (Banerjee & Lavie, 2005) (combines unigram precision, recall, and semantic matching to assess similarity), (f) CIDEr (Vedantam et al., 2015) (measures consensus in n-gram usage across multiple references, with tf-idf weighting), and (g) NIST (Doddington, 2002) (similar to BLEU but weights n-grams by their informativeness, favoring less frequent and meaningful phrases).

## C.4. Performance and Efficiency Comparision with Parameter-Efficient Fine-Tuning

While our work focuses on black-box LLM adaptation where model weights are inaccessible, we include a controlled comparison with Parameter-Efficient Fine-Tuning (PEFT) methods. Specifically, we implement LoRA (Hu et al., 2021) with

*Table 7.* Comparison between *Plugin* and PEFT (LoRA, r=8) on four datasets using GPT2-XL and LLaMA-3.1-8B as base models. We show mean and standard deviation of the metrics over five seeds.

| Model | Method | BLEU | Rouge-1 | Rouge-2 | Rouge-L | METEOR | CIDEr | NIST |
|---|---|---|---|---|---|---|---|---|
| | | | | **E2E NLG** | | | | |
| GPT2-XL | Zeroshot | 0.0562 | 0.4013 | 0.1636 | 0.2862 | 0.3697 | 0.0187 | 0.5338 |
| GPT2-XL | LoRA (r=8) | $0.2517_{\pm0.012}$ | $0.5712_{\pm0.010}$ | $0.3079_{\pm0.013}$ | $0.4317_{\pm0.011}$ | $0.5162_{\pm0.014}$ | $0.5225_{\pm0.012}$ | $1.2172_{\pm0.011}$ |
| GPT2-XL | Plugin (Ours) | $0.2470_{\pm0.009}$ | $0.5536_{\pm0.007}$ | $0.3084_{\pm0.007}$ | $0.4213_{\pm0.008}$ | $0.5057_{\pm0.009}$ | $0.5455_{\pm0.013}$ | $1.2736_{\pm0.051}$ |
| LLaMA-3.1-8B | Zeroshot | 0.3226 | 0.6917 | 0.4050 | 0.5004 | 0.6041 | 0.9764 | 1.1310 |
| LLaMA-3.1-8B | LoRA (r=8) | $0.3702_{\pm0.016}$ | $0.7125_{\pm0.010}$ | $0.4236_{\pm0.014}$ | $0.5345_{\pm0.012}$ | $0.6413_{\pm0.017}$ | $1.1028_{\pm0.033}$ | $1.1827_{\pm0.035}$ |
| LLaMA-3.1-8B | Plugin (Ours) | $0.3691_{\pm0.013}$ | $0.7113_{\pm0.002}$ | $0.4374_{\pm0.004}$ | $0.5247_{\pm0.002}$ | $0.6392_{\pm0.009}$ | $1.1441_{\pm0.030}$ | $1.1749_{\pm0.034}$ |
| | | | | **Web NLG** | | | | |
| GPT2-XL | Zeroshot | 0.0317 | 0.2992 | 0.1321 | 0.2417 | 0.1969 | 0.0491 | 0.1826 |
| GPT2-XL | LoRA (r=8) | $0.1723_{\pm0.007}$ | $0.4604_{\pm0.010}$ | $0.2618_{\pm0.011}$ | $0.3628_{\pm0.015}$ | $0.4012_{\pm0.017}$ | $0.9018_{\pm0.028}$ | $0.2736_{\pm0.014}$ |
| GPT2-XL | Plugin (Ours) | $0.1673_{\pm0.004}$ | $0.4616_{\pm0.007}$ | $0.2527_{\pm0.007}$ | $0.3757_{\pm0.008}$ | $0.3895_{\pm0.007}$ | $0.8987_{\pm0.013}$ | $0.2646_{\pm0.003}$ |
| LLaMA-3.1-8B | Zeroshot | 0.1453 | 0.5278 | 0.3030 | 0.3982 | 0.4314 | 0.6991 | 0.2684 |
| LLaMA-3.1-8B | LoRA (r=8) | $0.2638_{\pm0.008}$ | $0.6238_{\pm0.010}$ | $0.3927_{\pm0.009}$ | $0.4726_{\pm0.009}$ | $0.5927_{\pm0.013}$ | $1.6421_{\pm0.028}$ | $0.2379_{\pm0.008}$ |
| LLaMA-3.1-8B | Plugin (Ours) | $0.2542_{\pm0.004}$ | $0.6375_{\pm0.005}$ | $0.3873_{\pm0.005}$ | $0.4869_{\pm0.007}$ | $0.5724_{\pm0.004}$ | $1.5911_{\pm0.046}$ | $0.2590_{\pm0.003}$ |
| | | | | **CommonGen** | | | | |
| GPT2-XL | Zeroshot | 0.0317 | 0.2992 | 0.1321 | 0.2417 | 0.1969 | 0.0491 | 0.1826 |
| GPT2-XL | LoRA (r=8) | $0.1826_{\pm0.027}$ | $0.5027_{\pm0.010}$ | $0.2137_{\pm0.014}$ | $0.4447_{\pm0.016}$ | $0.4726_{\pm0.009}$ | $0.7182_{\pm0.027}$ | $0.6725_{\pm0.043}$ |
| GPT2-XL | Plugin (Ours) | $0.1791_{\pm0.014}$ | $0.4932_{\pm0.007}$ | $0.2288_{\pm0.004}$ | $0.4347_{\pm0.007}$ | $0.4702_{\pm0.006}$ | $0.7283_{\pm0.012}$ | $0.6554_{\pm0.038}$ |
| LLaMA-3.1-8B | Zeroshot | 0.0643 | 0.2776 | 0.1181 | 0.2488 | 0.3857 | 0.3155 | 0.3347 |
| LLaMA-3.1-8B | LoRA (r=8) | $0.2736_{\pm0.018}$ | $0.5829_{\pm0.009}$ | $0.3206_{\pm0.009}$ | $0.5026_{\pm0.012}$ | $0.5927_{\pm0.016}$ | $1.1121_{\pm0.034}$ | $0.7926_{\pm0.028}$ |
| LLaMA-3.1-8B | Plugin (Ours) | $0.2665_{\pm0.010}$ | $0.5800_{\pm0.002}$ | $0.3139_{\pm0.005}$ | $0.5037_{\pm0.004}$ | $0.5829_{\pm0.003}$ | $1.0876_{\pm0.020}$ | $0.7031_{\pm0.007}$ |
| | | | | **Adidas** | | | | |
| GPT2-XL | Zeroshot | 0.0075 | 0.2309 | 0.0278 | 0.1438 | 0.1487 | 0.0184 | 0.4956 |
| GPT2-XL | LoRA (r=8) | $0.0629_{\pm0.028}$ | $0.2816_{\pm0.030}$ | $0.0719_{\pm0.029}$ | $0.1816_{\pm0.038}$ | $0.2037_{\pm0.018}$ | $0.1231_{\pm0.126}$ | $0.6576_{\pm0.134}$ |
| GPT2-XL | Plugin (Ours) | $0.0600_{\pm0.017}$ | $0.2710_{\pm0.025}$ | $0.0722_{\pm0.018}$ | $0.1725_{\pm0.017}$ | $0.1995_{\pm0.018}$ | $0.1195_{\pm0.138}$ | $0.6375_{\pm0.120}$ |
| LLaMA-3.1-8B | Zeroshot | 0.0120 | 0.2470 | 0.0318 | 0.1493 | 0.1526 | 0.0424 | 0.5285 |
| LLaMA-3.1-8B | LoRA (r=8) | $0.0721_{\pm0.020}$ | $0.2697_{\pm0.031}$ | $0.0756_{\pm0.028}$ | $0.1821_{\pm0.020}$ | $0.2023_{\pm0.038}$ | $0.1302_{\pm0.178}$ | $0.6137_{\pm0.172}$ |
| LLaMA-3.1-8B | Plugin (Ours) | $0.0611_{\pm0.018}$ | $0.2714_{\pm0.029}$ | $0.0742_{\pm0.020}$ | $0.1759_{\pm0.019}$ | $0.1990_{\pm0.020}$ | $0.1293_{\pm0.152}$ | $0.6361_{\pm0.134}$ |

rank-8 matrices on the `query` and `value` projections of GPT2-XL and LLaMA-3.1-8B, and fine-tune the base models using the same task-specific data.

The performance results are shown in Table 7. Consider GPT2-XL as a reference example, *Plugin* adds a 1-layer autoregressive Transformer with 30.72M parameters, while LoRA (r=8) introduces only 2.46M trainable parameters. However, *Plugin* requires no modification of the base model and can be deployed post hoc. Despite the access advantage of LoRA, the performance gap is minimal. As for computational efficiency, *Plugin* requires 196.2B FLOPs (up to 64 decoding steps), while LoRA uses 188.8B FLOPs—a difference of less than 5%. The gap narrows or inverts depending on model configuration. These results suggest that *Plugin* offers a competitive adaptation solution even under white-box conditions, while maintaining broader applicability in black-box settings.

## C.5. Further Quantitative Analysis and Ablation

Following Section 7.3, we present the same ablation analysis using GPT2-M on the remaining three datasets. As shown in Figure 5, the trends mirror those in Figure 2: the *Plugin* model consistently improves performance as the base model becomes stronger with additional fine-tuning, underscoring the robustness and versatility of our approach. Similarly, Figure 6 confirms the pattern observed in Figure 3: a single-layer reweighting model yields optimal performance, while deeper configurations tend to overfit and degrade quality. Across all datasets, initializing the reweighting model with a pretrained GPT2-Small consistently boosts effectiveness.

## C.6. Influence of the architecture of the reweighting model in *Plugin*

We vary the choice of the reweighting model architecture. We find that a causal transformer layer identical to those used in the base model performs best, as it can leverage the base model's logits and aggregate contextual information from prior tokens to better adapt the base model to the new data distribution. This conclusion is reinforced by Figure 7, where the transformer architecture consistently outperforms both the MLP (two layer with ReLU activation) and linear layers across

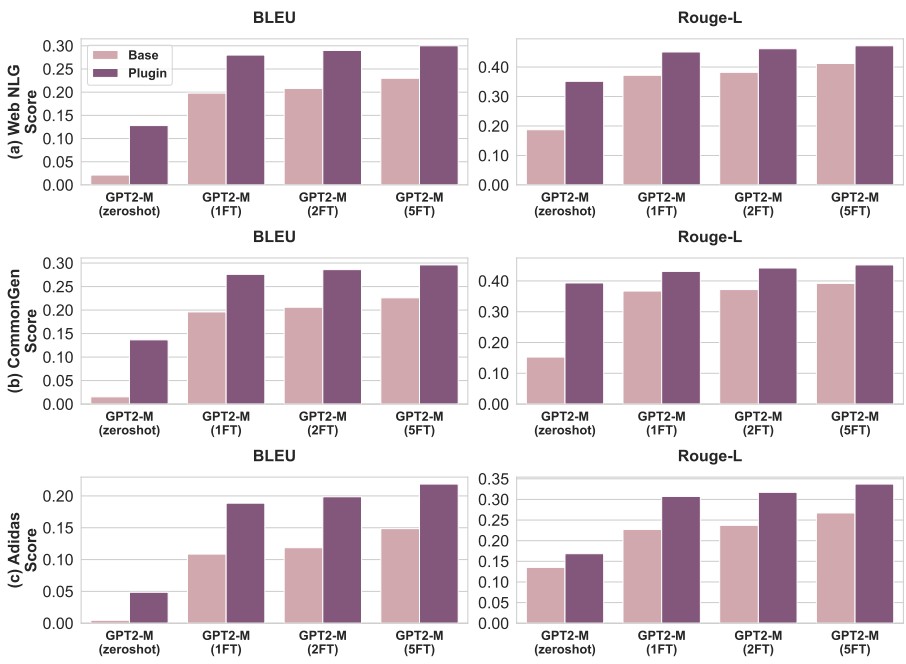

*Figure 5.* Performance of applying a single-layer reweighting model across increasingly fine-tuned GPT2-M models on the three datasets. Results demonstrate consistent improvements introduced by our method regardless of the strength of the base model.

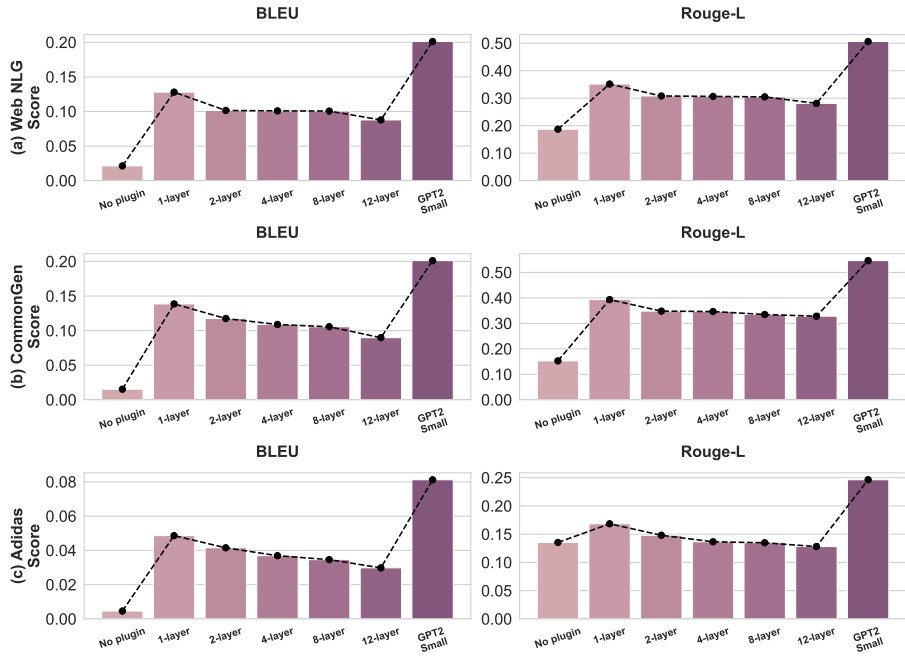

*Figure 6.* Performance of GPT2-M with varying reweighting model complexities on the three datasets, measured by BLEU and Rouge-L. Results demonstrate that a single reweighting layer achieves significant improvements, while increasing the number of layers beyond this leads to performance degradation, likely due to overfitting. Using a pretrained GPT2-Small as the reweighting model largely boosts the performance, highlighting the benefits of leveraging pretrained models.

all metrics, as indicated by higher means and narrower standard deviation bands. These results highlight the importance of leveraging the architectural capacity of transformers to effectively adapt the logits of the base black-box model.

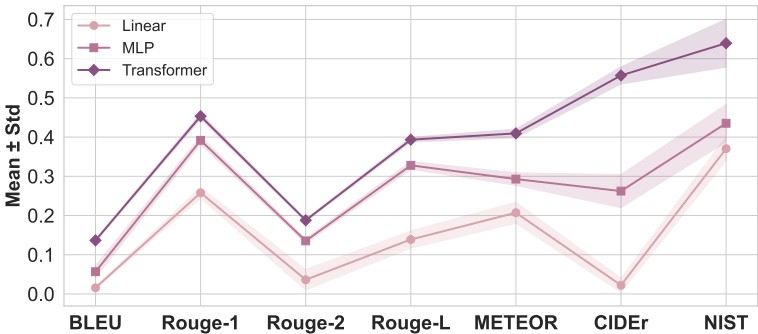

*Figure 7.* Performance comparison of the weighting model architecture in *Plugin*. The transformer layer achieves the best performance with consistently higher means and narrower standard deviations. Shaded bands represent the standard deviation around the mean.

### C.7. Details for Adidas Qualitative Studies

**Human Evaluation.** We conduct a human evaluation on 100 test passages from the Adidas product dataset, comparing outputs generated with and without applying the reweighting model, using LLaMA-3.1-8B as the base model. Three human evaluators are presented with a ground-truth Adidas product description and two randomly ordered descriptions: one generated with the reweighting layer and one without (i.e., we use the base model with ICL-3 as a much stronger baseline due to the low quality of the zero-shot). Evaluators are prompted to select the prediction closest to the ground truth. Results show that the output generated with the reweighting model is preferred on an average of 80.7 out of all 100 cases. The output descriptions from the base model without the reweighting are generally short and general. This demonstrates that our approach effectively adapts a closed model to the unique style of the given dataset.

In this section, we display some details for the qualitative analysis on the Adidas product description dataset.

**Details of Extracting Adidas Style Words.** We discuss the details on extracting the most frequent 50 words in the Adidas product description dataset as the "Adidas style" words. We argue that there does not exist a gold-standard way to define the "style" words for a dataset. We extract these style words through a minimal preprocessing pipeline: converting text to lowercase, removing special characters and numbers, and filtering out common English stopwords. We deliberately preserve the original word forms without lemmatization or stemming to maintain distinct style markers (e.g., keeping "comfortable" distinct from "comfort", "running" distinct from "run"). After tokenization using NLTK's word tokenizer, we count word frequencies across all product descriptions and select the top 50 most frequent words. This approach captures the exact vocabulary used in Adidas' product descriptions, including specific product features.

A statistics of the frequency of these top-50 words is shown in Figure 8.

**Case Studies for Human Side-by-side Evaluation.** Below we provide more case studies on the Adidas product description datasets. We display the given product attributes, base model's zero-shot outputs, base model with ICL-3's outputs, *Plugin* model's outputs, and the ground-truth references for each case.

> ♣ **Case 1**
> **Given Product Attributes:**
>
> - name [Cushioned Angle Stripe Low-Cut Socks 6 Pairs]
> - category [Accessories]
> - price [13]
> - color [Multicolor]
>
> **Base Model (Zero-shot):** Cushioned Angle Stripe Low-Cut Socks 6 Pairs, Multicolor.
>
> **Base Model (ICL-3):** These adidas socks are made with a cushioned sole for extra comfort and support. The low-cut

**Word Frequency of the Top-50 Words in Adidas Dataset**

*Figure 8.* Word Frequency of the Top-50 Words in Adidas Dataset.

design is perfect for everyday wear, whether you're running errands or hitting the gym.

**Plugin Model:** Cushioned Angle Stripe Low-Cut Socks 6 Pairs are the perfect blend of style and comfort. The low-cut design gives you a sleek, modern look while still providing plenty of support for your feet. Made with high-quality materials, these socks will keep your feet feeling comfortable and looking chic all day long.

**Ground-truth Reference:** With lots of cushioning in the heel and toe and a soft, stretchy feel, these kids' socks are big on comfort. Whether they're bouncing around between classes, practice or after-school activities, their little feet will stay dry thanks to moisture-wicking yarn and a mesh upper. An adidas Badge of Sport and 3-Stripes give the socks a classic adidas feel.

♠ **Case 2**
**Given Product Attributes:**

  - name [Badge of Sport Tee]
  - category [Clothing]
  - price [21]
  - color [Grey]

**Base Model (Zero-shot):** A white t-shirt with a badge of the sport you love on it.

**Base Model (ICL-3):** The adidas Badge of Sport tee is a classic. It's made with soft cotton and features the iconic 3-Stripes on the shoulders. The crew neck has a ribbed hem for a comfortable fit.

**Plugin Model:** There's nothing more classic than a white tee with your favorite team's logo. This adidas Badge of Sport Tee is an everyday essential that can be worn to the gym or on game days. It's made with recycled materials, so you're making a sustainable choice. The soft cotton jersey is lightweight, breathable, and comfortable.

**Ground-truth Reference:** Put your brand pride on full display. A simple yet versatile look, this long sleeve t-shirt

keeps you comfortable while on the move or lounging at home. A contrast adidas Badge of Sport stands out on the chest. This product is made with Primegreen, a series of high-performance recycled materials.

◆ **Case 3**
**Given Product Attributes:**

- name [Stretch Cotton Trunk Briefs 3 Pairs]
- category [Clothing]
- price [21]
- color [Grey]

**Base Model (Zero-shot):** Stretch Cotton Trunk Briefs 3 Pairs, Grey.

**Base Model (ICL-3):** These adidas men's trunks are made from a soft cotton blend that feels great against the skin. The stretchy fabric moves with you, while the flatlock seams prevent chafing and irritation. A drawcord waistband allows for an adjustable fit.

**Plugin Model:** Stretch Cotton Trunk Briefs 3 Pairs. The stretch cotton fabric is comfortable to wear. These briefs are made with a classic design in mind, so you can feel confident and comfortable. They have been designed with a low waist, which gives them a flattering fit and makes them easy to get on.

**Ground-truth Reference:** Just look at these adidas boxer briefs. They're made with soft, quick-dry fabric that keeps you feeling fresh and a stay-put design that keeps you covered from your first rep to your last. Get comfortable and give it your all. Our cotton products support sustainable cotton farming. This is part of our ambition to end plastic waste.

♥ **Case 4**
**Given Product Attributes:**

- name [adidas x Zoe Saldana Long Sleeve Tee]
- category [Clothing]
- price [28]
- color [Black]

**Base Model (Zero-shot):** This is a long sleeve tee with the adidas logo on the front and Zoe Saldana's name on the back. It has a crew neck, short sleeves, and a straight hem. The fabric is made of 100% cotton and is machine washable.

**Base Model (ICL-3):** This adidas x Zoe Saldana long sleeve tee is made with 100% recycled polyester. The soft, lightweight fabric has a smooth finish and features a crew neck and short sleeves. It's finished with an adidas logo on the chest.

**Plugin Model:** This is not your average training t-shirt. A modern, slim fit and the softest fabric make it one of our most comfortable tops to date. The lightweight feel makes this tee perfect for all-day wear. Pair it with a pair of leggings or joggers for that ultimate athleisure look you're always striving for ambition and comfort.

**Ground-truth Reference:** Your growth, your goals and the challenges you face along the way are all part of your journey. This adidas tee was created in collaboration with Zoe Saldana to inspire you to move with confidence, no matter where you are on your path. Smooth material lays softly on your body so you can feel comfortable in your own skin. A solid color offers styling versatility. This product is made with recycled content as part of our ambition to end plastic waste.

