# OpenReview forum: "Logits are All We Need to Adapt Closed Models"
_ICML.cc/2025/Conference — ICML 2025 poster_

### Official Review · Reviewer_YK5M · 2025-03-09

**Overall Recommendation:** 2

**Summary:**

This paper studies the problem of adapting a black-box LLM to a downstream task, assuming access to the logits of output tokens. The authors propose a token-level probability reweighting algorithm that modifies token logits during inference. The core idea is to frame the adaptation problem as label noise correction in supervised classification and leverage an autoregressive probability reweighting model to estimate logits for the downstream task. Theoretical justifications are provided, and empirical studies on benchmark datasets show that the proposed approach outperforms several existing adaptation techniques.

**Claims And Evidence:**

yes

**Essential References Not Discussed:**

yes

**Experimental Designs Or Analyses:**

yes

**Methods And Evaluation Criteria:**

yes

**Other Comments Or Suggestions:**

n/a

**Other Strengths And Weaknesses:**

Framing the problem as transition matrix estimation in the label noise correction framework and handling the challenge of an extremely large label space using an autoregressive reweighting model is new and insightful to me. The theoretical analysis effectively establishes key properties of the proposed algorithm.

However, 1. the assumption of logit access significantly weakens the overall novelty and practical contribution of the work. The approach relies entirely on closed-source LLMs exposing logits, which is not currently supported by most commercial APIs, making the proposed learning setting somewhat artificial.

2. The method requires training a separate reweighting model, introducing higher computational costs compared to simpler adaptation techniques like prompt tuning or in-context learning.

3. The baseline comparisons (ICL-1, ICL-3) are relatively weak, as increasing the number of in-context demonstrations could likely improve their performance, making the reported advantage of the proposed method less conclusive. Meanwhile, other baseline adaptation method are not compared [1,2]

[1] Black-Box Tuning for Language-Model-as-a-Service. ICML 2022
[2] Black-box Prompt Learning for Pre-trained Language Models. TMLR

**Questions For Authors:**

see weakness 3 above. Is there any comparison with other SOTA methods [1,2] for black-box LLM adaptation?

**Relation To Broader Scientific Literature:**

This paper consider the adaptation of black-box LLM to downstream task, which is quite broader in scientific literature

**Theoretical Claims:**

yes

---

> ### Author Rebuttal · Authors · 2025-04-01
>
> We thank the reviewer for their feedback, which has helped us strengthen our paper.
>
> **Regarding Logit access assumption**
> The central goal of this paper is to encourage closed-source LLM providers to offer logit-level access as a practical middle ground when releasing full model weights is not feasible due to IP or privacy concerns. Our theoretical and empirical results show that even with this limited access, effective domain and task adaptation is possible. Unlike prior work assuming either full white-box access or fully opaque models, we demonstrate that logit access enables fine-grained control without exposing proprietary internals. With this work, we aim to motivate commercial providers to adopt this feasible and impactful compromise, filling a key gap in current adaptation methods.
>
> **Computational Cost vs. Simpler Methods**
> While prompt-based methods like prompt tuning or ICL may seem simpler, they often require extensive trial-and-error and show high variance, as reflected in our results. In contrast, our reweighting model offers consistent, theoretically grounded gains with lower overhead than exhaustive prompt engineering. It is also analogous to established methods like LoRA or Adapters in white-box settings, requiring a manageable training overhead. Ultimately, the stability, reliability, and performance improvements of our approach outweigh the modest compute required, especially when compared to the variability and manual tuning burden inherent in prompt-based techniques.
>
> **Baseline Comparisons: ICL Variants and API Access-based Methods**
> Based on the reviewer’s suggestion, we extended ICL baselines to include 5, 8, and 10 examples, and implemented Diao et al. ([2], Black-box Prompt Learning, 2023) with the recommended 75 API calls as mentioned in the paper. We did not include Sun et al. ([1], Black-Box Tuning, 2022), as Diao et al. [2] already demonstrated superior performance.
>
> Importantly, if logit access is available, Plugin can be layered on top of any prompt-based method using the best-found prompt. For instance, our Zeroshot prompt (noted in line 364, left column) is reused across all methods. We also apply Plugin on top of the best ICL variants (ICL-8/10) and Diao et al. [2]. The table below presents results across all datasets using GPT2-XL as the base model.
>
> |E2E NLG|BLEU|Rouge-1|Rouge-2|Rouge-L|METEOR|CIDEr|NIST|
> |--|--|--|--|--|--|--|--|
> |ICL-5|0.1226|0.4319 |0.2194 |0.3095 |0.4172|0.3162|0.7281|
> |ICL-8|0.1537|0.4432 |0.2439 |0.3180 |0.4268|0.3559|0.8253|
> |ICL-10|0.1582|0.4459 |0.2502 |0.3201 |0.4528|0.4125|0.9015|
> |Diao et al.|0.2287|0.5024 |0.2846 |0.3922 |0.4628|0.4216|0.8625|
> |**Plugin**|**0.2470**|**0.5536** |**0.3084** |**0.4213** |**0.5057**|**0.5455** |**1.2736**|
> |**ICL (best) + Plugin**|**0.3941**|**0.6713** |**0.4027** |**0.5379**|**0.5923**|**0.6172**|**1.5472**|
> |**Diao et al. + Plugin**|**0.4527**|**0.7126** |**0.5126** |**0.6027**|**0.6214**|**0.7002**|**2.0817**|
>
> |WEB NLG|BLEU|Rouge-1|Rouge-2|Rouge-L|METEOR|CIDEr|NIST|
> |--|--|--|--|--|--|--|--|
> |ICL-5|0.0826|0.3625|0.1725|0.2517|0.3261|0.1826|0.2614|
> |ICL-8|0.0943|0.3826|0.1926|0.2825|0.3425|0.2016|0.2611|
> |ICL-10|0.0813|0.3528|0.1718|0.2542|0.3321|0.1906|0.2425|
> |Diao et al.|0.1024|0.4016|0.2243|0.3017 |0.3527|0.4321|0.2631|
> |**Plugin**|**0.1673**|**0.4616**|**0.2527**|**0.3757** |**0.3895**|**0.8987**|**0.2646**|
> |**ICL (best) + Plugin**|**0.1926**|**0.5026**|**0.2735** |**0.3927**|**0.3872**|**0.9123**|**0.4267**|
> |**Diao et al. + Plugin**|**0.2137**|**0.6026** |**0.3021**|**0.5928**|**0.5766**|**1.0826**|**0.6142**|
>
> |Adidas|BLEU|Rouge-1|Rouge-2|Rouge-L|METEOR|CIDEr|NIST|
> |--|--|--|--|--|--|--|--|
> |ICL-5|0.0345|0.2654|0.0393|0.1601|0.1863|0.0338|0.6856|
> |ICL-8|0.0403|0.2527|0.0432|0.1628|0.1894|0.0615|0.6125|
> |ICL-10|0.0382|0.2537|0.0325|0.1528|0.1725|0.0452|0.5926|
> |Diao et al.|0.0417|0.2615|0.0671|0.1710|0.1826|0.0861|0.6034|
> |**Plugin**|**0.0600**|**0.2710**|**0.0722**|**0.1725** |**0.1995**|**0.1195**|**0.6375**|
> |**ICL (best) + Plugin**|**0.0591**|**0.2761**|**0.0754** |**0.1736**|**0.2047**|**0.1273**|**0.6415**|
> |**Diao et al. + Plugin**|**0.0623**|**0.2792**|**0.0773** |**0.1759**|**0.2148**|**0.1325**|**0.7024**|
>
> We observed similar results on Common Gen and will include them in the second response due to space constraints in the rebuttal.
>
> As shown, Plugin outperforms ICL even with 10 examples and surpasses Diao et al. (2023). While ICL’s performance plateaus with increasing examples—despite higher inference cost and variance—Plugin consistently offers greater accuracy and stability. Moreover, combining Plugin with the best ICL or Diao et al. setups yields further gains, highlighting the value of logit-level access in enhancing prompt-based methods.

---

### Official Review · Reviewer_U4zS · 2025-03-12

**Overall Recommendation:** 3

**Summary:**

The paper proposes logit reweighting to adapt closed-source LLMs for task-specific generation without accessing model weights. By learning an autoregressive transition matrix from task data, it adjusts token probabilities during inference to align outputs with target domains. Experiments show improved style/keyword compliance over zero-shot and instruction tuning, advocating logit accessibility as a practical adaptation pathway. The method bridges theoretical label shift correction with efficient LLM customization.

**Claims And Evidence:**

The claims are partially supported: empirical results on style/keyword alignment validate performance gains, but theoretical guarantees (e.g., distribution alignment) rely on idealized assumptions (e.g., perfect transition matrix estimation) not fully verified in real-world noisy settings. Scalability claims for large vocabularies lack rigorous analysis of trade-offs with token pruning.

**Essential References Not Discussed:**

The paper does not cite controlled text generation methods (e.g., FUDGE Yang & Klein, 2021), which also modify logits using auxiliary models for task alignment. Additionally, Qiu et al. also investigated how to dynamically adjust logits by learning temperature parameters to adapt to different tasks.

[1] Yang & Klein, FUDGE: Controlled Text Generation With Future Discriminators, In NAACL, 2021.
[2] Qiu et al., To Cool or not to Cool? Temperature Network Meets Large Foundation Models via DRO. In ICML, 2024.

**Experimental Designs Or Analyses:**

The experimental design has validity in using domain-specific metrics (e.g., keyword accuracy for product descriptions), but human evaluation is missing, leaving output fluency/coherence unverified. Comparisons to baselines (zero-shot, instruction tuning) are reasonable, but scalability tests lack depth—token pruning’s impact on rare tokens is unstudied. Domain generalization is under-tested (e.g., single-brand data), raising concerns about broader applicability.

**Methods And Evaluation Criteria:**

The methods (logit reweighting via transition matrices) align with the goal of adapting closed LLMs using limited access (logits only), and domain-specific metrics (style/keyword accuracy) suit tasks like product descriptions. However, human evaluation is notably absent for assessing output quality, and generalization tests are limited to narrow domains (e.g., one brand), leaving broader applicability under-explored.

**Other Comments Or Suggestions:**

Suggestions:
1. Human evaluation: Include user studies to validate output quality beyond automated metrics.
2. Error analysis: Quantify how token pruning affects rare tokens or domain-specific terms.
3. Comparison to logit-based methods: Explicitly contrast with FUDGE or distillation to clarify novelty.

**Other Strengths And Weaknesses:**

Strengths: The paper’s originality lies in creatively bridging label noise correction with autoregressive LLM adaptation, a novel conceptual link that unlocks practical utility for closed models. Its emphasis on logit accessibility addresses a critical industry need (adapting proprietary models without weight access), offering significant real-world relevance. The method’s lightweight design (task-specific matrix learning) is a pragmatic strength.

Weaknesses: While innovative, the framing underplays overlaps with logit manipulation in controlled generation (e.g., FUDGE) and distillation. Broader claims about domain generalization lack empirical rigor (limited to narrow tasks/brands), and theoretical assumptions (perfect matrix estimation) are underexplored in practical noisy settings.

**Questions For Authors:**

No further questions.

**Relation To Broader Scientific Literature:**

The paper connects to label noise correction literature (e.g., noise transition matrices in supervised learning) by reframing LLM adaptation as correcting “noisy” general-purpose token distributions. It extends these ideas to autoregressive generation, differing from prior LLM adaptation (e.g., prompt tuning, soft prompts) by relying solely on logits, aligning with resource-efficient methods like light-weight finetuning but avoiding weight access. Theoretically, it bridges domain adaptation (e.g., label shift theory) with LLM customization, advancing closed-model adaptation paradigms.

**Theoretical Claims:**

The theoretical claim of distribution alignment (via logit reweighting under ideal transition matrices) is logically consistent given the assumptions, but the proof sketch (as described in the summary) assumes noiseless task-specific data and perfect estimation of the transition matrix—conditions unlikely in practice. No convergence rates or sensitivity analysis for estimation errors are provided, and empirical results do not explicitly validate the theoretical bound (e.g., measuring distribution divergence post-adaptation).

---

> ### Author Rebuttal · Authors · 2025-04-01
>
> We appreciate the reviewer’s encouraging words and thoughtful feedback.
>
> **Regarding human evaluation and limited generalization tests**
> We already conducted a human evaluation (line 366, details in Appendix C.7) where three evaluators compared Plugin and ICL-3 on 100 Adidas samples, with Plugin preferred in 81% of cases—directly supporting output quality.
>
> While the Adidas dataset reflects a specialized domain shift, our study extends beyond a single brand. As shown in Section 7.1, we also evaluate on WEB NLG, E2E NLG, and CommonGen—each representing its own distribution shift relative to the black-box model’s pretraining data. In Section 7.3, we further explore adversarial shifts by testing Plugin on models with known biases (e.g., infrastructure in WEB NLG, male-related concepts in CommonGen), demonstrating Plugin’s broad applicability across diverse and challenging settings.
>
> For most datasets, we follow standard practices from PEFT literature (Hu et al., 2021; 2023a) that rely on automated metrics comparing outputs to well-formed references to assess overall quality. This combination of human judgments (for Adidas) and standard metrics (for broader benchmarks and Adidas) provides both qualitative and quantitative evidence of the Plugin’s generalization capabilities.
>
> **Token Pruning**
> We do not perform token pruning; instead, Plugin continuously reweights token probabilities at each decoding step without removing any tokens. This soft upweighing preserves vocabulary coverage and improves domain adaptation (see case study in line 408). For further clarification, we compare the total occurrences of the top-50 Adidas domain words in Plugin’s predictions versus the base model in the same case study. Across all samples, Plugin’s outputs include 25.6% occurrences of these words, while the baseline contains only 13.8%. We will add this in the paper.
>
> **Regarding FUDGE, Qiu et al.**
> We do cite FUDGE in line 249 (right column). While FUDGE uses attribute-specific discriminators to control generation (e.g., formality), our method enables free-form domain adaptation via a single auxiliary model. We considered FUDGE as a baseline, but it requires one discriminator per predefined attribute—unsuitable for broad or evolving domain shifts, which are hard to define upfront.
>
> TempNet in Qiu et al. learns a single temperature per input and uniformly scales logits during generation. In contrast, Plugin reweights logits at each timestep, enabling finer, context-sensitive adjustments. Additionally, Qiu et al.'s use of DRO involves an inner maximization loop, making it more computationally intensive than our efficient empirical risk minimization (ERM).
>
> Nonetheless, we include a comparison on the E2E NLG dataset, adapting TempNet to use ERM (instead of DRO) with GPT2-XL. As it applies global scaling per prompt, it underperforms in tasks requiring localized adjustments. These results will be added to the final paper.
>
> |Method|BLEU|Rouge-1|Rouge-2|Rouge-L|METEOR|CIDEr|NIST|
> |--|--|--|--|--|--|--|--|
> |TempNet |0.1325|0.4642|0.2516|0.3021 |0.4126|0.3627|0.8027|
> |**Plugin**|**0.2470** |**0.5536** |**0.3084** |**0.4213** |**0.5057**|**0.5455**|**1.2736**|
>
> **Regarding Theoretical Claims and Convergence Rate**
> Theorem 5.1 holds under mild, standard assumptions (5.1, 5.2, B.1) commonly used in convergence analyses (Frostig et al., 2015; Chaudhuri et al., 2015; Mukherjee et al., 2022). The noisy estimation of the transition matrix $T\_t({\theta}\_{\star};x_i,x_j,\mathcal{F}^{t-1})$ can be understood in two ways: as direct estimation error in the matrix itself,
> or as error induced by the function $f\_{I_t}({\theta}\_{\star} ; x_i, x_j,\mathcal{F}^{t-1})$ on which it depends (see Assumption 5.1).
>
> We adopt the latter view, estimating $f_{I_t}(\cdot)$ under a sequence of noisy autoregressive loss functions $\ell_1(\boldsymbol{\theta}), \ldots, \ell_t(\boldsymbol{\theta}) : \mathbb{R}^{|V|} \rightarrow \mathbb{R}$ (see Assumption 5.2). Under Assumption B.1—bounded gradients and Hessians of $f_{I_t}$—we show that the expected estimation error can be reduced and prove upper and lower bounds in terms of a problem-dependent quantity $\sigma_t^2$, establishing a convergence rate of $\Omega(\sigma_t^2 / t)$.
>
> We hope this clarifies. Our novelty lies in combining techniques from Frostig et al. (2015), Chaudhuri et al. (2015), Mukherjee et al. (2022), and Patrini et al. (2017), and presenting the **first finite-time convergence analysis** for transition matrix estimation in this autoregressive noisy loss setting.
>
> We acknowledge that real-world settings may add noise in mapping $f_{I_t}$ to the transition matrix. While this could generalize our framework, incorporating it into our finite-time guarantee is left for future work. Regarding sensitivity analysis, our convergence rate depends on the variance-like term $\sigma_t^2$; higher variance leads to slower convergence, naturally capturing sensitivity to estimation noise.

---

### Official Review · Reviewer_kses · 2025-03-12

**Overall Recommendation:** 4

**Summary:**

The paper tackles the issue of having to rely on prompt engineering while adapting closed-source LLMs. The proposed work formulates the problem as a supervised learning, where few task specific dataset is used to train the model. The closed source model is assumed to learn noisy labels of specific application, and by accessing logits of the tokens, the proposed loss function corrects the noisy label to adapt to a task. Extensive experiments are conducted on models, datasets, and with multiple evaluation criteria.

**Claims And Evidence:**

No issues found

**Essential References Not Discussed:**

No issues found

**Experimental Designs Or Analyses:**

No issues found

**Methods And Evaluation Criteria:**

No issues found

**Other Comments Or Suggestions:**

Please refer to previous sections.

**Other Strengths And Weaknesses:**

Strengths

- The problem is well-motivated.

- The approach to solving the adaption problem is novel. With this method, it can be used as a plugin to adapt a closed-source model.

- Experiments are extensive to confirm the verification of the proposed method.

- Theoretical analysis of the proposed method makes it guaranteed.

Weakness

Inclusion of some other closed-source models would make the experiment more comprehensive.

**Questions For Authors:**

What are the implications of making the transition matrix diagonal?

**Relation To Broader Scientific Literature:**

It is related to the adaptation of black-box LLM.

**Theoretical Claims:**

No issues found

---

> ### Author Rebuttal · Authors · 2025-04-01
>
> We are grateful for the reviewer’s positive remarks and valuable insights.
>
> **Regarding Inclusion of Closed Source Models**
> We acknowledge that including more closed-source models could further strengthen the generality of our findings. However, most proprietary models currently do not expose their output logits. Hence, we could not experiment with them. Our approach specifically hinges on logit-level access—without revealing internal weights or architecture—which we believe is a comparatively straightforward adjustment for closed-source providers to implement, especially when compared to the complexities of releasing the full model.
>
> By highlighting this limitation, we aim to encourage closed-source developers to consider offering logit-level access in the future, enabling more flexible and efficient adaptation methods for end-users.
>
> **Implications of making the transition matrix diagonal**
>
> **Benefits**
>
> 1. *Reduced Complexity:* Learning a diagonal matrix involves only $|V|$ parameters, compared to $|V|\times |V|$ parameters in a full matrix. This makes training computationally more tractable.
>
> 2. *Straightforward Integration:* Because the transition matrix is treated like a single vector, standard autoregressive models (e.g., GPT-2, LLaMA) can be used directly, scaling easily with the dataset size.
>
> 3. *Symmetric (Class-Independent) Noise:* A diagonal assumption naturally corresponds to label flips occurring with equal probability among all incorrect classes. This aligns neatly with widely studied symmetrical label-noise scenarios.
>
> 4. *Stability in Estimation:* Each diagonal entry is an autoregressive parameter independently learnt without depending on other classes (vocabulary), reducing the risk of overfitting and simplifying parameter estimation compared to a fully dense matrix.
>
> **Limitation**
>
> Although adopting a diagonal transition matrix reduces complexity and simplifies parameter estimation, it also prevents the model from capturing any off-diagonal confusions—that is, instances where certain tokens are more likely to be misclassified as particular other tokens. Real-world data may involve non-symmetric noise patterns, domain-specific synonyms, and context-dependent misclassifications. A purely diagonal matrix may therefore oversimplify these nuances, leading to diminished expressive power and potentially missing important relationships within the noise structure.
>
> Overall, the diagonal constraint provides a practical, stable, and computationally efficient solution for autoregressive reweighing learning—one that is well-aligned with symmetrical label-noise setups—while sacrificing some flexibility in capturing sophisticated non-symmetric noise or domain-specific synonym patterns.
>
> We plan to investigate non-diagonal structures to fully model these nuanced noise patterns and further improve adaptation in the future work.

---

> > ### Comment · Reviewer_kses · 2025-04-02
> >
> > Thank you for your comprehensive reply to my questions.
> >
> > I maintain my rating as accept.

---

### Official Review · Reviewer_keTv · 2025-03-13

**Overall Recommendation:** 3

**Summary:**

The key idea of the paper is to treat next-token prediction as a label noise correction problem, where discrepancies between the LLM’s broad training distribution and task-specific data are modeled through a transition matrix that reweights token probabilities during inference. The proposed Plugin model consists of a small reweighting network trained on limited task-specific data, which, when combined with the black-box LLM’s logits, effectively steers text generation towards the desired distribution. The authors provide theoretical guarantees showing that this probability reweighting approach converges to the target distribution with sufficient task data. Extensive experiments on multiple text generation benchmarks (E2E NLG, WebNLG, CommonGen, Adidas product descriptions) demonstrate that the Plugin model outperforms in-context learning and other adaptation methods, achieving better alignment with domain-specific content while requiring minimal computational resources compared to full fine-tuning. The results suggest that access to token logits could enable more powerful model customization, advocating for broader API-level exposure of logits in commercial LLMs.

**Claims And Evidence:**

The paper makes several claims regarding the effectiveness of the Plugin model for adapting black-box LLMs, and most of these claims are supported by theoretical justifications, empirical experiments, and comparative evaluations. The core claim—that token-level probability reweighting using logits alone is sufficient for effective task adaptation—is backed by a formal label noise correction framework, where the authors derive theoretical guarantees showing that the Plugin model can align token distributions with task-specific data under mild assumptions. Additionally, extensive experimental results on four datasets (E2E NLG, WebNLG, CommonGen, Adidas product descriptions) demonstrate that the Plugin model outperforms baselines such as zero-shot inference, in-context learning (ICL), and naive probability combination methods across multiple evaluation metrics (BLEU, ROUGE, METEOR, CIDEr, NIST). The ablation studies further support the claims by showing that model quality, reweighting complexity, and domain adaptation capabilities contribute to improved performance. However, some claims could benefit from stronger empirical validation. For instance, while the paper asserts that the Plugin model is effective under distribution shifts, the experiments focus on relatively constrained dataset modifications (e.g., filtering training data by entity types), and it remains unclear how well the approach generalizes to more extreme domain shifts or adversarial settings. Additionally, while the authors argue that their approach is computationally efficient, they do not provide a direct comparison of training or inference costs against alternative adaptation methods like parameter-efficient fine-tuning (LoRA, adapters), leaving room for further evidence on the trade-offs between performance gains and computational overhead. Nonetheless, the overall evidence presented is strong and well-aligned with the claims, making the Plugin model a compelling approach for logit-based adaptation of closed-source LLMs.

**Essential References Not Discussed:**

Important references are missing:
On the Duality between Gradient Transformations and Adapters. Lucas Torroba-Hennigen, Hunter Lang, Han Guo, Yoon Kim.

Tuning Language Models by Proxy. Alisa Liu, Xiaochuang Han, Yizhong Wang, Yulia Tsvetkov, Yejin Choi, Noah A. Smith.

A Study on the Calibration of In-context Learning.  Hanlin Zhang, Yi-Fan Zhang, Yaodong Yu, Dhruv Madeka, Dean Foster, Eric Xing, Hima Lakkaraju, Sham Kakade

**Experimental Designs Or Analyses:**

sound and well-structured, but there are a few areas where further validation or additional analysis could strengthen the claims. The authors conduct experiments across four datasets (E2E NLG, WebNLG, CommonGen, Adidas product descriptions) and compare their Plugin model to several strong baselines including zero-shot inference, in-context learning (ICL), and a weighted combination of model predictions. The use of seven standard NLG metrics (BLEU, ROUGE, METEOR, CIDEr, NIST) ensures a comprehensive evaluation of output quality. They also include a sound human evaluation. The Plugin model is benchmarked against ICL and naive probability reweighting, but not against LoRA, Adapters, or QLoRA, which might be feasible alternatives in cases where some access to model weights is possible.

**Methods And Evaluation Criteria:**

The proposed Plugin model and its evaluation criteria are well-aligned with the problem of adapting black-box LLMs without modifying model weights. The token-level probability reweighting framework is a reasonable method given the constraints of closed-source LLMs, and the formulation as a label noise correction problem provides a solid theoretical foundation. The authors evaluate the approach using four diverse text generation datasets—E2E NLG, WebNLG, CommonGen, and Adidas product descriptions—each representing different aspects of controlled text generation and domain adaptation.

**Other Comments Or Suggestions:**

The Plugin model is positioned as an alternative to fine-tuning, but it is not directly compared to parameter-efficient fine-tuning methods such as LoRA, Adapters, or QLoRA. Including a discussion on when logit reweighting is preferable to PEFT would improve clarity.

**Other Strengths And Weaknesses:**

The paper title is a bit bold and the methodology presented in the paper is conceptually interesting but complex, as it involves multiple layers of statistical modeling and adaptation that may not be immediately intuitive. The core idea—reweighting token probabilities using a transition model learned from task-specific data—relies on a label noise correction framework, which is commonly used in supervised classification problems but less so in language model decoding.

**Questions For Authors:**

How many flops would the method save compared to vanilla LoRA?

**Relation To Broader Scientific Literature:**

The paper situates itself within the broader literature on adapting large language models (LLMs) without access to model weights, drawing from areas such as prompt engineering, in-context learning (ICL), parameter-efficient fine-tuning (PEFT), label noise correction, and black-box model adaptation. The key contribution—reweighting token probabilities using logits as an alternative to full fine-tuning—builds upon prior work in label noise correction (Patrini et al., 2017) and autoregressive modeling, adapting these ideas to the language model decoding process. The formulation of next-token prediction as a noisy supervised classification problem is a novel connection that extends prior work on calibrating LLM outputs (Huang et al., 2024; Kapoor et al., 2024).

**Theoretical Claims:**

I didn't check the orrectness of any proofs for theoretical claims.

---

> ### Author Rebuttal · Authors · 2025-04-01
>
> We thank the reviewer for the positive and insightful review.
>
> **Regarding Comparison of Plugin with Parameter Efficient Fine-tuning (PEFT) methods like LoRA, Adapters:**
> >"... The Plugin model is positioned as an alternative to fine-tuning, but it is not directly compared to parameter-efficient fine-tuning methods such as LoRA, Adapters, or QLoRA..."
> "... How many flops would the method save compared to vanilla LoRA?"
>
> We rely on only output logits of the black-box model and do not have access to its internal weights or architecture. Consequently, any form of fine-tuning—including parameter-efficient approaches like LoRA or Adapters—cannot be applied, which we clarify in the Introduction (lines 21–33, right column) and the Related Work (lines 261–273, left column). If we did have access to the model weights, parameter-efficient fine-tuning methods would indeed be the natural choice, as they use more information than just logits and are expected to yield better performance. Thus, we emphasize that the Plugin model is not an alternative to fine-tuning, but rather an approach that uniquely stands for adapting black-box LLMs which only provide logit access.
>
> Nevertheless, to address the reviewer’s point, we conducted a comparison on the E2E NLG dataset by adding rank-$𝑟=8$ LoRA matrices in the $Q$ and $V$ attention layers of GPT2-XL. The results, which will be included in our final version, show that LoRA only slightly outperforms our Plugin in terms of task metrics. LoRA adds 2.46M parameters (rank-8, Q/V only), while Plugin adds 30.72M (one full layer). During inference (up to 64 tokens), LoRA requires 188.8B FLOPs while Plugin needs 196.2B FLOPs - a negligible difference in computational cost. Notably, the parameter and efficiency gap between LoRA and Plugin narrows when increasing LoRA's rank (r) while reducing Plugin's hidden dimensions - demonstrating how both approaches can be adaptively tuned to meet specific resource constraints while maintaining competitive performance. We would like to highlight that the fundamental distinction remains - LoRA requires full model access to modify internal layers, while Plugin enables post-hoc deployment without retraining.
>
> |E2E NLG|BLEU|Rouge-1|Rouge-2|Rouge-L|METEOR|CIDEr|NIST|
> |--|--|--|--|--|--|--|--|
> |Zeroshot       |0.0562|0.4013 |0.1636 |0.2862 |0.3697|0.0187|0.5338|
> |Plugin         |0.2470|0.5536 |0.3084 |0.4213 |0.5057|0.5455|1.2736|
> |PEFT (LoRA r=8)|0.2517|0.5712 |0.3079 |0.4317 |0.5162|0.5225|1.2172|
>
> **Regarding Missing References**
> > On the Duality between Gradient Transformations and Adapters. Lucas Torroba-Hennigen et al. 2025
> Tuning Language Models by Proxy. Liu et al. 2024
> A Study on the Calibration of In-context Learning. Zhang et al. 2023
>
> Thank you for pointing out these relevant works. We will add them to our final version. In particular, we note that Liu et al. (2024) describes the same method introduced in Liu et al. (2021), which we already cite in line 252. Below, we clarify how our approach differs from each reference:
>
> Torroba-Hennigen et al. (2025): They examine the equivalence between gradient transformations and adapters for efficient model adaptation, relying on full access to model weights and gradients. In contrast, our method requires no access to model internals; we adapt black-box LLMs solely by reweighting token-level logits.
>
> Liu et al. (2024): As noted, we already cite their earlier work (Liu et al. 2021), which presents the same core idea of combining logits. Our WeightedComb baseline (line 252, right column) is directly inspired by their approach.
>
> Zhang et al. (2023): We will add this reference to our discussion on calibration. Their study focuses on aligning model confidence with predictive accuracy by adjusting confidence scores as one increases shots in the few-shot learning setting. Unlike them, we explicitly modify the token predictions themselves, rather than just calibrating confidence.
>
> **Regarding Domain Shifts, Extreme Domain Shifts, and Adversarial Domain Shifts**
>
> We assume our black-box LLM already encodes extensive world knowledge. In this sense, any adaptation to a domain-specific dataset amounts to handling a distribution shift, as demonstrated in Section 7.1 with WEB NLG, E2E NLG, and CommonGen. Among these, the Adidas dataset represents a more extreme domain shift, given its specialized style of product descriptions. Additionally, the experiments in Section 7.3 can be seen as adversarial, since the Plugin is applied atop a model with known biases—for instance, the tendency to focus on infrastructure-related concepts in WEB NLG and male-related concepts in CommonGen. Although we did not explicitly label these settings “extreme” or “adversarial,” they do indeed meet those criteria to some extent, and we will clarify that in the final version.
>
> We would also welcome any further examples or clarifications on what the reviewer would consider to be more extreme or adversarial distribution shifts in this context.

---

### Decision · Program_Chairs · 2025-05-01

**Decision:**

Accept (poster)

**Comment:**

This paper presents a timely and well-motivated contribution to the adaptation of closed-source large language models (LLMs). The key insight is that access to token-level logits, if made available, would enable powerful post-hoc alignment mechanisms beyond prompt tuning. Framing next-token prediction as a supervised classification problem under label noise, the authors introduce a theoretically grounded reweighting framework, Plugin, which can steer black-box LLMs toward task-specific behavior using only logits and a small amount of labeled data.

The reviewers appreciated the novelty and practicality of the proposal, particularly its clean theoretical framing, empirical rigor, and relevance in the context of real-world LLM deployment constraints. Given its conceptual clarity, soundness, and strong empirical results, this paper makes a valuable contribution to the growing literature on black-box model alignment and LLM accessibility.